# Idealised simulations of cyclones with robust symmetrically-unstable sting jets

Ambrogio Volonté[1,2], Peter A. Clark[1], and Suzanne L. Gray[1]

[1]Department of Meteorology, University of Reading, Reading, RG6 6BB, UK
[2]National Centre for Atmospheric Science-Climate, University of Reading, Reading, RG6 6BB, UK

**Correspondence:** Ambrogio Volonté (a.volonte@reading.ac.uk)

**Abstract.** Idealised simulations of Shapiro-Keyser cyclones developing a sting jet (SJ) are presented. Thanks to an improved and accurate implementation of thermal wind balance in the initial state, it has been possible to use more realistic environments than in previous idealised studies. As a consequence, this study provides further insight in SJ evolution and dynamics and explores SJ robustness to different environmental conditions, assessed via a wide range of sensitivity experiments.

The control simulation contains a cyclone that fits the Shapiro-Keyser conceptual model and develops a SJ whose dynamics are associated with the evolution of mesoscale instabilities along the airstream, including symmetric instability (SI). The SJ undergoes a strong descent while leaving the cloud-head banded tip and markedly accelerating towards the frontal-fracture region, revealed as an area of buckling of the already-sloped moist isentropes. Dry instabilities, generated by vorticity tilting via slantwise frontal motions in the cloud head, exist in similar proportions to moist instabilities at the start of the SJ descent

and are then released along the SJ. The observed evolution supports the role of SI in the airstream's dynamics proposed in a conceptual model outlined in a previous study.

Sensitivity experiments illustrate that the SJ is a robust feature of intense Shapiro-Keyser cyclones, highlighting a range of different environmental conditions in which SI contributes to the evolution of this airstream, conditional on the model having adequate resolution. The results reveal that several environmental factors can modulate the strength of the SJ. However, a

positive relationship between the strength of the SJ, both in terms of peak speed and amount of descent, and the amount of instability occurring along it can still be identified.

In summary, the idealised simulations presented in this study show the robustness of SJ occurrence in intense Shapiro-Keyser cyclones and support and clarify the role of dry instabilities in SJ dynamics.

## 1   Introduction

The current state of knowledge about sting jets (SJs) was recently reviewed by Clark and Gray (2018) (CG18 hereafter). SJs are coherent air streams in extratropical cyclones that descend from mid-levels within the cloud head and accelerate into a frontal-fracture region. Frontal fracture is characteristic of cyclones that evolve according to the Shapiro-Keyser conceptual model (Shapiro and Keyser, 1990). SJs descend towards the top of the boundary layer (lying above the cold conveyor belt jet for at least part of the cyclone lifecycle) and can lead to localised transient strong, damaging, winds if their associated strong

momentum is mixed down to the ground. The SJ is distinct in location, lifespan and spatial scale from the synoptic-scale cold and warm conveyor belts that exist in most cyclones. However, it is difficult to distinguish the SJ from the cold conveyor belt using surface observations alone (e.g. see Martínez-Alvarado et al. (2014) in which numerical model simulations as well as chemical tracer observations were used to distinguish these two strong wind regions). As a result, there are relatively few in-depth studies of SJs in observed cyclones, and certainly not enough to allow more than qualitative conclusions regarding the relationship between SJ structure and precursor characteristics. Here we use idealised simulations of SJ-containing cyclones developing in a realistic atmospheric environment to explore the robustness of SJ generation and characteristics for a wide range of environmental background states and assess the importance of dry mesoscale instability release in the generation of the SJ.

Climatological analyses using the existence of atmospheric instability in the cloud head as a predictor of likelihood that a cyclone will produce a SJ have revealed that SJs are probably a common feature of cyclones: Hart et al. (2017) found 32% of all extended winter North Atlantic cyclones had the required instability, increasing to 42% in the 22% of those cyclones that developed explosively. The proportion of cyclones with SJs may also increase in a future warmer climate (Martínez-Alvarado et al., 2018). Although the proportion of cyclones with SJs that lead to strong surface winds is not known, the likely current and future prevalence of cyclones with SJs and diagnosed presence of SJs in case studies of damaging European cyclones such as the Great October storm (Browning, 2004) and the St. Jude's day storm (Browning et al., 2015) motivates further research into their characteristics.

In the time since SJs were first formally identified in Browning (2004) detailed case studies of about ten SJ-containing cyclones have been published (see list in Table 2 in CG18). These analyses have revealed many common features and led to the development of the commonly-accepted definition given above. More contentious has been attribution of SJs to a mechanism with large-scale dynamics, the release of mesoscale instabilities (conditional symmetric instability (CSI), symmetric instability (SI), inertial instability (II) and conditional instability (CI)), evaporative cooling and frontal dynamics all evidenced as being associated with SJ descent (see section 5 of CG18). Consequently, it is concluded in that review that it is likely that a continuum of behaviour occurs with release of mesoscale instability (if it occurs) enhancing the SJ strength beyond that which can be achieved through frontal dynamics in the frontolytic region at the tip of the cloud head.

Although mesoscale instability presence and release has been diagnosed in many case studies (e.g. the study of windstorm Friedhelm by Martínez-Alvarado et al. (2014)), the control of the atmospheric environment afforded by moist idealised simulations of SJ cyclones makes them an ideal tool for exploring the range of instabilities that may occur and their impact on SJ characteristics. Baker et al. (2014) produced the first analysis of idealised simulations of SJs in cyclones and found that, while the presence of a SJ was robust to the environmental state in which the cyclone developed, the existence of the different mesoscale instabilities varied: CSI release occurred along the SJ in the control simulation whereas II and CI release occurred in the simulation with the weakest static stability. In their idealised simulations Coronel et al. (2016) found spatially-localised regions of negative saturated equivalent moist potential vorticity (MPV*, for which negative values imply the presence of mesoscale instability) and near-zero MPV* values along diagnosed SJ trajectories. The authors interpreted this finding as implying that the environment was near-neutral to CSI. However, as explained in CG18, another interpretation is that the near-

neutral condition is evidence of CSI having been (or continuously being) released. Buckled absolute momentum surfaces were also generated in the Coronel et al. (2016) study implying that II existed.

  While both these idealised modelling studies produced cyclones with characteristics consistent with those of observed SJ cyclones, the atmospheric environments were cooler (and in Coronel et al. (2016) also drier) than typically observed, likely due to the constraints involved in designing idealised environments: Baker et al. (2014) used a near -freezing surface temperature,

though their initial relative humidity was 80%, and Coronel et al. (2016) also used cool temperatures together with a lower-tropospheric relative humidity of 60%. In this study we have overcome these constraints and have used environments that are more consistent with those observed. We also explore a different range of parameter values and parameters to these previous studies. Baker et al. (2014) only presented results of changing the initial tropospheric static stability of the environmental state in their moist simulation in their paper (as changing this parameter was found to have the greatest effect on the strength and

descent rate of the resulting sting jet), although the dependence on upper-tropospheric jet strength, stratospheric static stability and cyclonic barotropic shear were also explored by the authors and the results of these experiments are presented in the associated PhD thesis (Baker, 2011). Coronel et al. (2016) explored the sensitivity of the SJ cyclone and SJ to whether or not it is initialised to the warm south side, or on, the upper-tropospheric zonal jet axis and also to model resolution (considering three resolution configurations from the four possible with horizontal grid spacings of 20 km and 4 km and two different

vertical resolutions). In our study we explore the sensitivity of SJs to horizontal resolution, initial state relative humidity, upper-tropospheric jet strength, and surface temperature.

  In contrast to most previous studies we focus on the importance of mesoscale dry instabilities (II and SI), rather than moist instabilities (CI and CSI), for SJ enhancement, following analysis of the prevalence of all of the different mesoscale instabilities. This focus is motivated by our recent analysis of windstorm Tini (Volonté et al., 2018). In that study we demonstrated that

mesoscale instability release occurring in a higher-resolution simulation enhanced the strong winds in the frontal-fracture region occurring through synoptic-scale cyclone dynamics (from comparison with results from a coarser-resolution simulation that was not able to represent mesoscale instability release). We also found that the SJ first became largely unstable to CSI and then to SI and II. Most previous moist case study analyses that have assessed the role of mesoscale instability release in SJ generation have not explicitly considered SI though some have considered II, as now summarised. Martínez-Alvarado

et al. (2014) found that the SJ trajectories in windstorm Friedhelm (2011) followed a band of initially II and later CSI as they wrapped cyclonically within the cloud head. The recent Eisenstein et al. (2020) study of windstorm Egon (2017) found, for their control simulation, that the proportion of SJ trajectories that were unstable to II and SI increased to about 90% as the SJ descended and the previously-present CSI was released. They conclude that CSI appears crucial to trigger the descent when the SJ is still within the cloud head, while II and SI take over later during the descent with a similar contribution. Similarly

Brâncuş et al. (2019) find that II was present in the last part of the SJ descent in an explosive Mediterranean cyclone as the SJ entered into the boundary layer, although in contrast to the previous study they concluded that mesoscale instability release was not important during the initial SJ descent. Finally, although Smart and Browning (2014) do not explicitly discuss the possible presence of dry instabilities in their case analysis of windstorm Ulli (2012) they show the evolution of PV along three sting jet trajectories in their Figure 9. The PV decreases as all three sting jet trajectories ascend and along one of the trajectories

approaches very close to zero PVU at 01 UTC 3 January, as the trajectories reach their minimum pressure level and start to descend. Although none of this small sample of three trajectories acquire negative PV, the approach towards near neutrality to SI along one of the trajectories suggests that the possible presence of SI cannot be excluded.

As noted in CG18, as the environment in an intense extratropical cyclone is unlikely to be strictly barotropic, a diagnosis of II should probably be interpreted as implying the presence of SI. However, in Volonté et al. (2018) we distinguished between II
diagnosed through negative values of the vertical component absolute vorticity, $\zeta_z$, and SI diagnosed through negative potential vorticity (PV). Finally, we proposed that the generation of negative MPV*, and later negative PV, occurs through the tilting of horizontal $\zeta$ generated by frontal ascent and descent in the cloud head to produce negative values of the vertical component of $\zeta$ which is then amplified prior to the final stage of descent of the SJ by stretching. We further explore this proposed mechanism in the current study.

The structure of this paper is as follows. The methods are described in Sec. 2 starting with the idealised model configuration and setup of the initial environmental state (Sec. 2.1) and followed by the diagnostics used for mesoscale instability (Sec. 2.2), the trajectory method (Sec. 2.3) and the rationale for the sensitivity experiments (Sec. 2.4). The results section (Sec. 3) starts with an analysis of the SJ in the cyclone in the control simulation (Sec. 3.1) and is followed by analysis of the dependence of the SJ on the environmental initial state through the sensitivity experiments (Sec. 3.2). Section 4 contains a discussion of these
results and the overall conclusions.

## 2 Data and Methodology

### 2.1 Model configuration and chosen initial state

#### 2.1.1 Background

The initial base state used is generated using an algorithm based on, and ultimately very close to, that of Polvani and Esler
(2007), referred to as PE hereafter. However, two issues arise from their formulation. The first is that their derivation is based upon geostrophic thermal wind balance in hydrostatic primitive equation (HPE) set; as is well-known (e.g. White et al., 2005), in part this approximate set ignores various terms, including some Coriolis terms and some metric terms. As a result, a 'balanced' state in the HPEs is not perfectly balanced in a less approximate equation set (and *vice versa)*. The MetUM solves the non-hydrostatic deep atmosphere dynamical equations on a sphere, so the balanced initial state found by PE is only ap-
proximately balanced when used in the MetUM. This initial imbalance is quite unphysical and generates an initial transient flow (largely gravity waves) which, at the resolutions used, can result in an unstable response. Baker et al. (2014) followed the pragmatic approach of first running for a day or so in a lower resolution version of the model, which was found to sufficiently damp the transient response and settle to a balanced field that could be numerically interpolated to the required grid. However, we have derived an equivalent to geostrophic thermal wind balance in the non-hydrostatic deep atmosphere and hence removed
this initial imbalance. To our knowledge this has not been previously published, so we include the derivation in Appendix A1. Ultimately, the initial state we produce is essentially equivalent to that of PE but for a non-hydrostatic, deep atmosphere model.

A more significant issue with the initial conditions used by Baker et al. (2014) was their rather cold overall temperature, discussed above, which was necessary to produce numerically-stable model simulations. This temperature structure arises from the PE initial state; it is of only minor importance in PE's dry simulations, but restricts the magnitude of diabatic forcing by phase changes in Baker et al. (2014). In this study a method to adjust the entire vertical virtual potential temperature ($\theta_v$) profile to make that at the jet centre equal to the reference has been devised, the details of which are also given in Appendix A1. In any configuration, thermal wind balance tends to reduce the static stability in the upper troposphere to the south of the jet compared with that to the north—we have used a state which takes this to the extreme of being close to neutrality in the upper troposphere south of around 40°N. In fact, it is very weakly statically unstable, but not sufficiently to generate either a dynamical response or significant unstable mixing through the turbulence scheme, and, if unperturbed, it was found that the whole state remained essentially unchanged through a 10-day integration apart from some very minor wave activity in the upper stratosphere. As a result, this state represents probably the warmest we could achieve without taking a different approach from PE.

As a result of these improvements, the instability issues described in Baker et al. (2014) are no longer present and so it has been possible to use a more realistic temperature profile with a central value of potential temperature of 295 K at the surface, as opposed to the rather cold temperatures occurring in the that study. The SJ occurring in the control simulation has values of wet bulb potential temperature ($\theta_w$) around 285 K, ∼10 K warmer than the ones in the idealised simulation of Baker et al. (2014) and the high horizontal and vertical resolution idealised simulation of Coronel et al. (2016) (which were about 276 K and 274 K (for the warmest trajectory), respectively). The SJ analysed in a simulation of the Great Storm (Clark et al., 2005) had $\theta_w$ around 286 K while simulations of Gudrun (Baker, 2011), Ulli (Smart and Browning, 2014) and Tini (Volonté et al., 2018) produced SJs with $\theta_w$ around 279 K, 281 K and 278 K, respectively. Hence, the thermal properties of the SJs generated in the idealised simulations presented in this paper are consistent with those analysed in case studies, particularly warm ones like in the Great Storm.

The simulations described in this paper are performed in a spherical geometry, with the periodic channel extending from 12°N to 78.5°N in latitude and from 20°W to 25°E in longitude. The lower boundary condition is modelled as ocean and heat and moisture fluxes are set to zero. The initial state is designed to develop an LC1-type cyclone. Thorncroft et al. (1993) highlighted that LC1 cyclones develop similarly to the Shapiro-Keyser conceptual model with the occurrence of frontal fracture and, in later stages, a warm seclusion.

## 2.2 Mesoscale instability diagnostics

Mesoscale instabilities are detected following the method in Volonté et al. (2018), whose key points are repeated here. Diagnostics for each instability (labelling criteria in Table 1) are evaluated at each grid point and interpolated onto relevant trajectories (see Section 2.3). The two conditional instabilities (CI and CSI) contain an additional constraint of $\mathrm{RH}_{ice} > 80\%$ (where $\mathrm{RH}_{ice}$ is relative humidity calculated with respect to ice). This constraint is required because these instabilities can only be released if the air is saturated; the threshold of 80% is used because partial cloud formation occurs in the model when RH exceeds this threshold in the free troposphere. A grid point is labelled as stable only if none of the four diagnostic tests indicate instability. If instead diagnostic tests for CSI or CI indicate instability, but the saturation constraint is not met, the relevant grid point is

not labelled as stable (S) and does not belong to any of the categories in Table 1. Every point can be labelled with more than one instability at the same time if two or more conditions are met. Our approach is to take the criteria as indicators of the underlying atmospheric state to highlight the processes leading to instabilities that might drive the dynamics of the SJ. These criteria are not used to indicate which instability might take precedence if more than one is present, particularly as they rely on assumptions about the background state upon which perturbations grow that are very rarely met in a highly baroclinic environment such as an intense extratropical cyclone. The release of a diagnosed instability would be expected to be associated with air motion in the appropriate direction (i.e. vertical for CI, horizontal for II, and slantwise for CSI and SI). A more thorough discussion on the implications of this approach and on the relevant properties of PV (whose negative values are condition for symmetric instability) can be found in Section 2.3 of Volonté et al. (2018).

## 2.3 Trajectory analysis

The potential of Lagrangian trajectories to isolate airstreams and assess their properties and their time evolution has been extensively used in recent SJ research. Here, trajectories are computed using the LAGRANTO Lagrangian analysis tool (Wernli and Davies (1997); Sprenger and Wernli (2015)), as in Volonté et al. (2018). LAGRANTO uses an iterative Euler scheme with an iteration step equal to 1/12 of the time spacing of input data. Ideally, data every model time step would be used to compute the trajectories, but in practice some compromise is necessary to reduce the amount of required model output. For the idealised

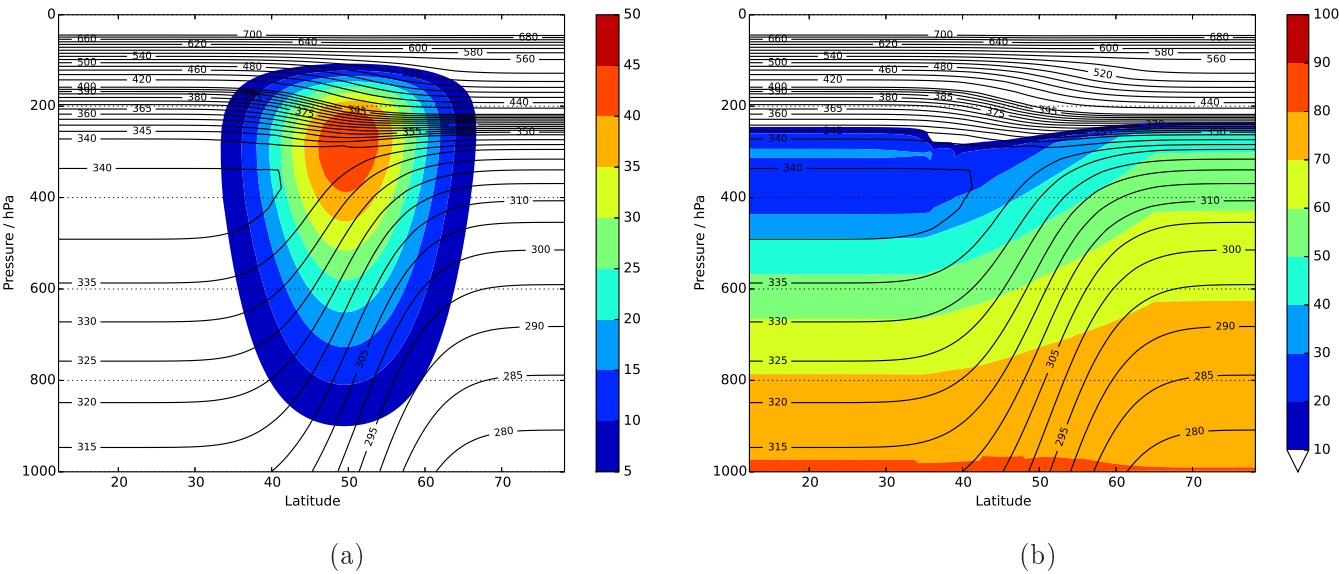

(a)  (b)

**Figure 1.** Initial potential temperature and (a) zonal wind, (b) relative humidity using setup according to Baker et al. (2014) but with $m_v = 10^{-8}$ in the stratosphere and $\Gamma_S = 0.025$, and our revised profile correction method. Note $\theta$ contour interval changes from 5 K to 20 K at 400 K.

| Label | $N_m^2$ | $RH_{ice}$ | $\zeta_z$ | PV | MPV* |
|---|---|---|---|---|---|
| Conditional Instability (CI) | $< 0$ | $> 80\%$ | | | |
| Inertial Instability (II) | | | $< 0$ | | |
| Symmetric Instability (SI) | | | | $< 0$ | |
| Conditional Symmetric Instability (CSI) | | $> 80\%$ | | | $< 0$ |
| Stable | $\geq 0$ | | $\geq 0$ | $\geq 0$ | $\geq 0$ |

**Table 1.** Criteria for trajectory instability and stability labels. $N_m^2$ is the moist Brunt-Väisälä frequency as defined by Durran and Klemp (1982). $\zeta_z$ is the vertical component of absolute vorticity (on pressure levels). PV is the potential vorticity and MPV* the moist saturated potential vorticity (Bennetts and Hoskins, 1979). Multiple entries in a row require all criteria to be satisfied (i.e. 'and' rather than 'or').

simulations analysed in this study it was found that trajectories computed with hourly input frequency of model data showed satisfactory conservation of relevant physical quantities; hence, all the results presented in this article use this input frequency.

### 2.4 Sensitivity experiments: motivation and configuration

The analysis of sensitivity experiments constitutes a substantial part of this study. As stated in the introduction, this analysis
has been performed on a different set of parameters and range of parameter values to previous literature (Baker et al., 2014; Coronel et al., 2016) and is aimed at assessing the robustness of the occurrence of the SJ in intense Shapiro-Keyser extratropical cyclones, along with its strength and connection with mesoscale instabilities. The impacts of variations in model configuration and environmental parameters have been explored: model resolution, upper-tropospheric jet strength, sea surface temperature and initial state relative humidity (see Table 2).

Previous literature has shown the importance of adequate model resolution for a correct simulation of SJ dynamics (see section 4.3 of CG18 for an overview and Volonté et al. (2018) for a detailed discussion on the effect of instability generation along the SJ). Therefore, two simulations have been run with horizontal spacing increased from the 0.11° of the control run and with accordingly coarser vertical resolution (as shown in Table 2). The control value of the maximum initial relative humidity, $RH_0 = 80\%$ (see Eq. (A36)), is the same as that chosen by Baker et al. (2014) to produce a profile similar to real
soundings. Experiments were performed with increased and decreased $RH_0$ to provide an assessment of the influence of moisture content on the evolution of the cyclone and the associated SJ. These experiments are particularly relevant because moist processes occurring in the cloud head have a primary role in the evolution of the cyclone in which the SJ occurs and are instrumental in the SJ generation mechanism proposed in Volonté et al. (2018). Additionally, four experiments have been performed varying the strength of the jet-stream speed by changing its initial central value $u_0$ (see Eq. (A31)) from the control
value of $u_0 = 45$ m s$^{-1}$. These experiments reveal the effects of different jet strength and, as a consequence, of different vertical wind shear and meridional temperature gradient.

A final set of experiments has been performed, but the results not included in this paper (although some results are presented in the PhD thesis by the lead author (Volonté, 2018)). This set consists of four experiments in which the physical latent heat

constants of condensation and freezing are scaled by 0.75, 0.875, 1.125 and 1.25, following Büeler and Pfahl (2017). Whilst these experiments can help in assessing the effect of a variation in the intensity of diabatic processes and, as a consequence the effect of a changed static stability and resultant PV distribution, without changing the initial wind and temperature profile, their results are not presented for essentially two reasons. Firstly, changing the values of thermodynamical constants such as these produces an unphysical situation with latent heat processes that are more or less intense than in the real world. Secondly, there is an inconsistency in the model as not all the variables can be updated according to the changes in these constants. In particular, the values of the specific humidity of saturation are taken from a look-up table, so they do not change if the value of the latent heat constants are changed, whereas all the thermodynamic quantities related directly or indirectly to those constants via an equation do. This generates a small inconsistency in the amount of cloud water, the effects of which are difficult to quantify. Having been unable to solve these problems and acknowledging that the results from this set of experiments do not describe the behaviour of a realistic cyclone, we decided to not include them in this paper. However, we mention the attempt as in our opinion this method could become useful in future analyses if the aforementioned issues can be solved.

## 2.5 Additional post-processing

The software NDdiag (Panagi, 2011) has been used to convert the model output to 15-hPa spaced pressure levels (30 hPa for the 04deg simulation) and to compute further diagnostic fields. All fields considered in this study, including conditions for instabilities (see Table 1), are thus on pressure levels. See Volonté et al. (2018) and Grams and Archambault (2016) for examples of the use of NDdiag in the literature.

System-relative speed has been computed by subtracting the average speed of the cyclone centre, i.e. its surface pressure minimum, from the Earth-relative wind speed of the trajectories. The average speed of the cyclone centre has in turn been computed after applying a 16-hour smoothing to the detected pressure minimum location. This choice is motivated by the need to remove the jumpiness of the precise location of the surface pressure minimum in each of the different simulations while retaining genuine variations in the SJ speed relative to the motion of the cyclone centre. At the same time, system-relative speed is strongly dependent on the relative directions of the SJ and of the cyclone movement, and these can vary between different simulations and on different timescales. The time value for the running mean has thus been chosen empirically to obtain a system speed that is as consistent as possible between different simulations. System speed values go from 8.3 to 13.5 m s$^{-1}$ in the simulations with modified initial jet-stream strength and stay within 10.9 and 12.4 m s$^{-1}$ in all other simulations. Most of this variation is due to changes in zonal system speed, while meridional system speed ranges from 2.4 to 3.2 m s$^{-1}$ in all simulations.

| Experiment | horiz. grid spacing (°) | vertical levels | vert. spacing at 850 hPa (m) | $u_0$ (m s$^{-1}$) | $RH_0$ (%) | $T_0$ (K) |
|---|---|---|---|---|---|---|
| control | 0.11 | 70 | 140 | 45 | 80 | 295 |
| 025deg | 0.25 | 63 | 200 | 45 | 80 | 295 |
| 04deg | 0.4 | 38 | 360 | 45 | 80 | 295 |
| rh90 | 0.11 | 70 | 140 | 45 | 90 | 295 |
| rh70 | 0.11 | 70 | 140 | 45 | 70 | 295 |
| jet35 | 0.11 | 70 | 140 | 35 | 80 | 295 |
| jet40 | 0.11 | 70 | 140 | 40 | 80 | 295 |
| jet50 | 0.11 | 70 | 140 | 50 | 80 | 295 |
| jet55 | 0.11 | 70 | 140 | 55 | 80 | 295 |
| t291 | 0.11 | 70 | 140 | 45 | 80 | 291 |
| t293 | 0.11 | 70 | 140 | 45 | 80 | 293 |
| t297 | 0.11 | 70 | 140 | 45 | 80 | 297 |
| t299 | 0.11 | 70 | 140 | 45 | 80 | 299 |

**Table 2.** Table showing the parameter values chosen for the different experiments. Model top height is 40 km for all simulations. $u_0$,$RH_0$ and $T_0$ indicate, respectively, the maximum initial values of wind speed and relative humidity and the central initial value of surface temperature, as defined in Section 2.1.

## 3   Results

### 3.1   Identification and characterisation of a SJ in the control simulation

#### 3.1.1   Overview of the life-cycle of the storm

230   As discussed in the introduction, a necessary condition for the occurrence of a SJ is that the extratropical cyclone containing it evolves according to the Shapiro-Keyser conceptual model. Figure 2 confirms that the idealised moist baroclinic wave evolves according to this model by showing the time evolution of surface pressure and potential temperature, $\theta$, at 850 hPa throughout the main stages in the control simulation. At 72 hours from the start of the run the cyclone is in its initial stage of development (Figure 2a), with meridional advection associated with the growing disturbance. At 84 hours (Figure 2b) the foremost part of

235   the warm advection is now westward relative to the cyclone centre as the cyclone evolves towards a frontal 'T-bone' structure

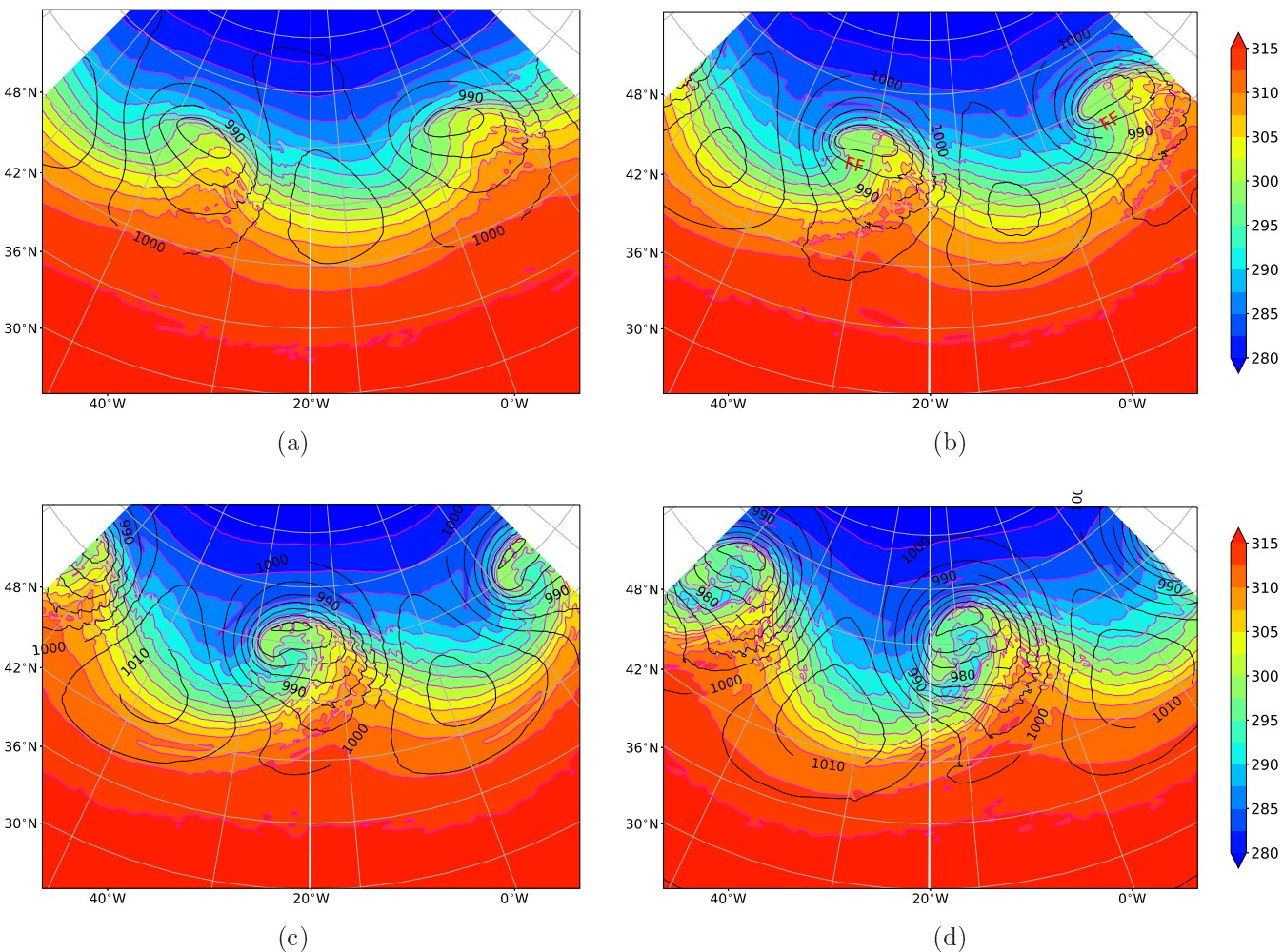

**Figure 2.** Surface pressure (black contours, hPa) and $\theta$ at 850 hPa (colours and magenta lines, K) after (a) 72, (b) 84, (c) 96 and (d) 108 hours from the start of the control simulation. The location of the opening frontal-fracture region is marked with 'FF' in panel (b). Note that the model domain (whose original zonal extension goes from 20°W to 25°E with periodic E-W boundary conditions) is repeated zonally to facilitate visualisation.

with the development of a bent-back front and the opening of a frontal-fracture region to the south of the cyclone centre, an area with intermediate $\theta$ values (between 295 K and 300 K) and weak gradients of $\theta$ (and of $\theta_w$, not shown). This opening frontal-fracture region is marked with 'FF' in Figure 2b. At 96 hours (Figure 2c) the bent-back front is fully developed and the associated cold advection is already starting to close the frontal-fracture region and isolate a warm seclusion at the centre 240 of the cyclone. The cyclone is now deeper, with the minimum surface pressure below 970 hPa, 12 hPa less than 12 hours before. At 108 hours (Figure 2d) the warm seclusion is almost completely encircled by the wrapped-up bent-back front, in what resembles a mature stage of evolution of a Shapiro-Keyser cyclone gradually losing its baroclinic nature. The evolution

just described follows the four stages of the Shapiro-Keyser model, correctly displaying its main features in time and space. In particular, it is between 84 and 96 hours from the start that the cyclone contains a widening frontal fracture, resembling a typical Shapiro-Keyser cyclone in Stage III of its development (Shapiro and Keyser, 1990). Hence, it is in that time range that the descent of a SJ can be expected to take place in the frontal-fracture region (CG18).

### 3.1.2 Analysis of strong winds in the frontal-fracture region

Figure 3 shows that strong low-level winds in the frontal-fracture region can indeed be clearly identified in the control simulation during stage III of the cyclone evolution. At 94 hours from the run start, Figure 3a displays a wind-speed maximum exceeding $36 \, \mathrm{m \, s^{-1}}$ at 850 hPa that is located in an area characterised by weak $\theta_w$ gradients. This region is just south of the cyclone centre, and lies behind an intense cold front ($\Delta \theta_w \sim 10$ K across the less-than-100 km width of the front) and ahead of the cloud-head tip and the associated bent-back front, partially wrapped around the cyclone centre. The cloud-head tip at 700 hPa displays some waviness, consistent with the occurrence of cloud banding and slantwise convection. It is clear from all the features just described that the location of this wind maximum is indeed the frontal-fracture region, which lies outside the tip of a bent-back banded cloud head visible at higher levels. This is consistent with the descent of a SJ.

Other low-level strong-wind regions of similar or even larger magnitude can be found elsewhere in the cyclone (see in the same figure the elongated strong-wind region running along the warm front, locally up to $34 \, \mathrm{m \, s^{-1}}$, associated with the warm conveyor belt) and/or at later times. The strongest wind speeds at 850 hPa in the whole lifetime of the storm, close to $40 \, \mathrm{m \, s^{-1}}$, occur at around 120 hours and are associated with the cold conveyor belt when the storm is in its mature stage (not shown). As this work is focused on the evolution of SJs in idealised extratropical cyclones, other low-level strong-wind features will not be examined.

The core of the SJ-associated wind maximum is located around 800 hPa, as shown in Figure 3b. The black dots overlaid on the wind-maximum region ($> 38 \, \mathrm{m \, s^{-1}}$) indicate the starting points of the Lagrangian trajectories selected to represent the core of this airstream. The selection has been performed by using only a constraint on Earth-relative wind speed at a specific time, i.e. by identifying the 100 grid points with highest wind speed at 94 hours from the run start located at 805 hPa or in contiguous levels (i.e. aligned vertically with the grid points selected at 805 hPa to form uninterrupted columns of points with speed exceeding the threshold). All these points have a wind speed exceeding $38.51 \, \mathrm{m \, s^{-1}}$ and they are all located between 760 and 820 hPa in a compact area in the core of the wind maximum visible in the frontal-fracture region, just ahead of the cloud-head tip.

A cross section taken along the frontal-fracture region (Figure 3c, transect AB in Figure 3a) shows an evident fold in $\theta_w$ embedded in an area of weak $\theta_w$ gradients. The downward orientation of this fold is a sign of the descent of an airstream (the SJ) that is distorting moist isentropes in an area where weak descent is already present. This fold is also located underneath a more stable layer of downward-sloped isentropes in the middle troposphere, which are consistent with the widespread presence of negative vertical velocity, particularly intense near to the cyclone centre where a tropopause fold can be expected and it is indeed hinted at by a bend in $\theta_w$ visible at around 400-500 hPa.

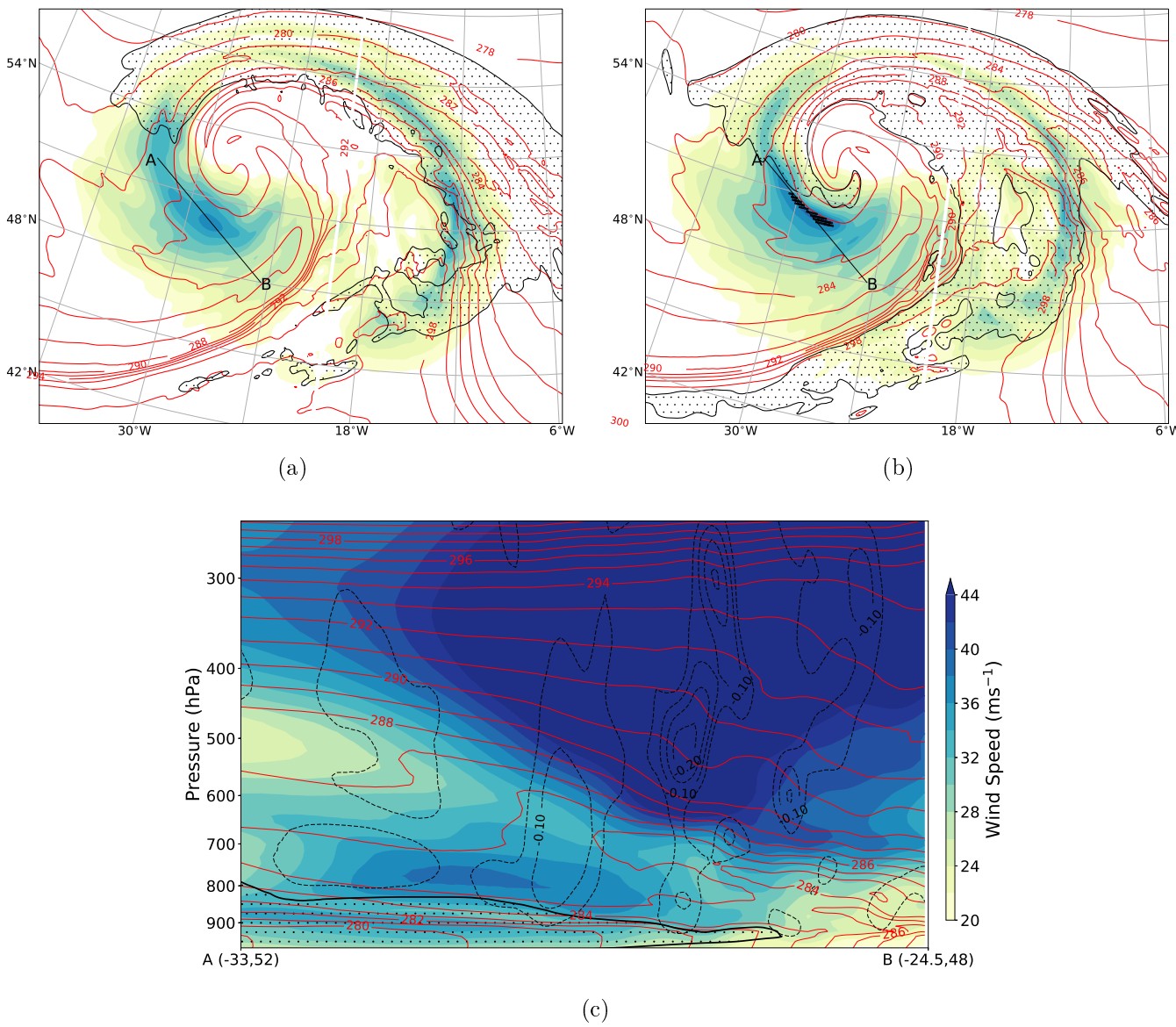

**Figure 3.** (a) Wind speed at 850 hPa (shading, m s$^{-1}$), $\theta_w$ at 850 hPa (K, red lines) and cloudy regions at 700 hPa (RH$_{ice}$ >80%, black lines and dotted regions) after 94 hours from the start of the control simulation (as all the other panels in this figure). Note that the model domain (whose original zonal extension goes from 20°W to 25°E with periodic E-W boundary conditions) is repeated zonally to facilitate visualisation but only data between 45°W and 5°W are shown here. (b) Same as (a) but at 805 hPa for all fields. Black dots show the horizontal locations of SJ trajectories. (c) Cross section (transect AB in panel (a)) of wind speed (green filled contours), negative vertical velocity (dashed black lines, m s$^{-1}$), $\theta_w$ (red lines, K) and cloudy regions (RH$_{ice}$ >80%, thick black lines).

The low-level SJ wind maximum described in the previous paragraphs can be clearly identified in this cross section, located along the base of the $\theta_w$-fold, surrounded by regions of negative vertical velocity and with speed values close to 40 m s$^{-1}$ at its core, at around 800 hPa. This wind maximum sits on top of the saturated boundary layer, neutral to slightly conditionally unstable in its lower part and capped by a strongly stable layer. The boundary layer also contains a low-level jet around 900 hPa, with wind speed exceeding 34 m s$^{-1}$, that is possibly associated with the front edge of the cold conveyor belt. The situation depicted by Figure 3 is consistent with the description of SJ-associated wind maxima in previous studies (listed in Table 3 in CG18). In particular, this can be compared with Figure 4 in Volonté et al. (2018); the structure is very similar, even though that SJ reaches higher wind speeds, up to 60 m s$^{-1}$ (to our knowledge, the highest SJ peak speed reported).

Lagrangian trajectories are used in the next sections to assess the characteristics of the identified SJ, following the steps outlined in Volonté et al. (2018).

### 3.1.3 Analysis of relevant quantities on the SJ airstream

Figure 4 shows the time evolution of various physical quantities along the trajectories identified as the SJ core, with dashed vertical lines highlighting key times in their evolution: 82, 88 and 94 hours from the start of the run. These times indicate respectively (and somewhat subjectively): (1) the start of the increase in SJ speed in both Earth-relative and system-relative reference frames; (2) the start of more rapid SJ descent and the end of the main increase in system-relative speed; and (3) the end of the increase in Earth-relative SJ speed, at the time of SJ detection. The steady increase in wind speed experienced by the airstream between (1) and (3) is displayed in Figure 4a, with maximum values close to 40 m s$^{-1}$. Before (1) the various trajectories span wind speeds between 5 and 30 m s$^{-1}$, the range at least partially a consequence of different paths taken by the air parcels as they travelled around the cyclone centre. Some markedly reduce their speed, by up to 20 m s$^{-1}$ before reaching their peak altitude and starting descent.

Conversely, during their increase of wind speed all trajectories behave coherently, with the median value of trajectories increasing in speed by almost 30 m s$^{-1}$ (Figure 4a), indicating that the acceleration of a single coherent airstream is occurring, i.e the SJ. Part of this acceleration is due to the SJ rotating around the cyclone centre and moving eventually in the same direction of the overall motion of the cyclone when on its southern side. This effect is not present in Figure 4b, which shows wind speed in a system-relative reference frame, i.e. having subtracted the motion of the storm (mainly zonal) from the Earth-relative wind. Nevertheless, this figure shows that system-relative wind speed increases by about 5 m s$^{-1}$ for the median of trajectories between (1) and (2). For approximately 1/3 of the trajectories the coherent increase in system-relative wind speed in the same period reaches 10 m s$^{-1}$. It must be acknowledged that a substantial minority of the trajectories undergo little or no speed increase or even a slow-down between (1) and (2). Between (2) and (3) the trajectories are a much more coherent bundle and generally experience further small but positive speed increment, the median increasing by 2-3 m s$^{-1}$. These values, although substantially smaller than the $\sim$15 m s$^{-1}$ calculated for windstorm Tini in Volonté et al. (2018), indicate a non-negligible local speed increase of the SJ that is not related to the overall motion of the storm.

Figure 4c shows that the trajectories form a coherent bundle after they start their descent (or, more accurately, as they are back trajectories, start to lose their coherency as they extend back in time from the start of their descent). Their increase in

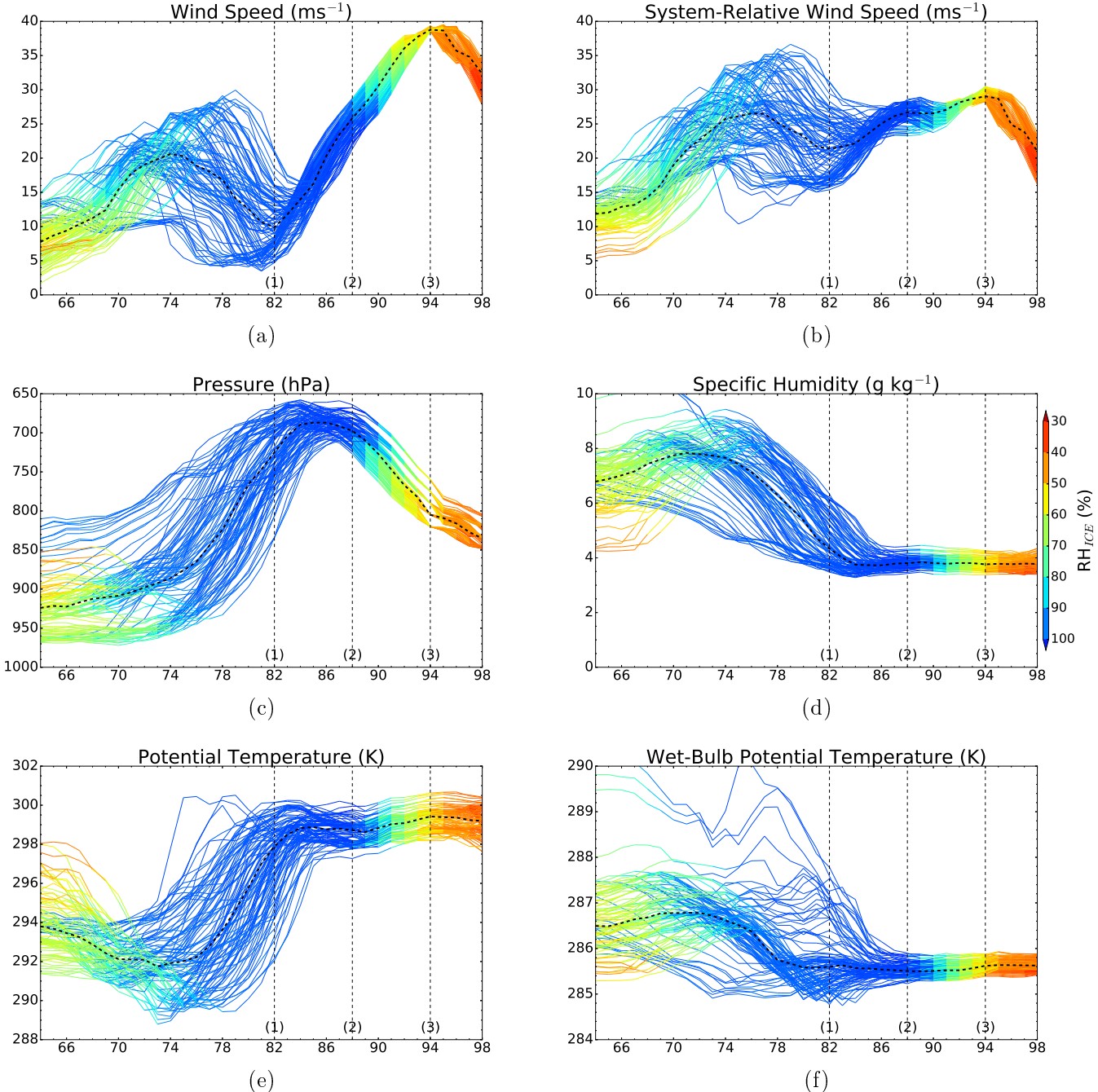

**Figure 4.** Timeseries (hours since start of the run) of (a) wind speed, (b) system-relative wind speed, (c) pressure, (d) specific humidity, (e) potential temperature and (f) $\theta_w$ along SJ trajectories. Colours indicate relative humidity with respect to ice along trajectories and the dashed line indicates the median of trajectories.

Earth-relative wind speed occurs at the same time as a steady descent. The median SJ trajectories increase their median pressure by 108 hPa in 6 hours, moving from 697 to 805 hPa between (2) and (3), during the most rapid stage of their descent. After (3) the descent gets weaker, as the SJ starts decelerating. This descent comes after an earlier ascent in which different populations of trajectories, mainly starting below the pressure level of 850 hPa, merge into a single airstream located around 700 hPa. Again, magnitude and rate of this descent are consistent with results from previous studies, although smaller than the largest

reported of 150 hPa in 3 hours, which was in windstorm Tini (see Figure 6 in Volonté et al. (2018)).

The descent of the SJ is also associated with a decrease in relative humidity, from close to saturation down to 40% in less than 10 hours. Figure 4d shows that during this drying stage specific humidity stays nearly constant at around $4 \, \text{g} \, \text{kg}^{-1}$ with only an almost negligible increase, of around $0.1 \, \text{g} \, \text{kg}^{-1}$ for most trajectories, in the first part of the descent up to (2). Throughout the ascent instead, all the trajectories stay close to saturation, with specific humidity substantially decreasing for all trajectories

(from nearly $8 \, \text{g} \, \text{kg}^{-1}$ to less than $4 \, \text{g} \, \text{kg}^{-1}$) after an initial increase.

Figures 4e and 4f confirm the absence of any clear evaporative — or sublimational — cooling signal during the descent of the SJ (in fact $\theta_w$ variations do not exceed 0.15 K for most trajectories in the whole descent) and the presence of condensational heating during the ascent, with a median increase in $\theta$ of around 7 K over 10 hours. This overall behaviour implies the occurrence of substantial condensation and precipitation while the SJ ascends within the cloud head whereas the amount

of evaporation during descent is at least an order of magnitude smaller. The role of evaporative cooling during descent is discussed in Sec. 5.6 of CG18. In common with our results, a number of papers cited therein suggest that evaporative cooling is negligible, as does recent work by Brâncuş et al. (2019). CG18 also highlight the possibility that evaporative cooling may have had an impact of some of the more extreme observed SJs, and this is further supported by the more recent work of Eisenstein et al. (2020). Our results support their statement that "while evaporative cooling may be occurring, it seems unlikely that this

additional latent cooling is *essential* for the formation of a SJ".

The limited variations in $\theta_w$ for most SJ trajectories from the final part of the ascent and throughout all the subsequent evolution indicate also the accuracy of the trajectories computed (as $\theta_w$ is expected to be conserved in absence of radiative and ice processes, and mixing), confirming the 'single coherent airstream' behaviour. This is opposed to the large variations, up to a few K, happening particularly by the beginning of the ascent where different populations of trajectories merge before starting

to travel together along a narrow frontal zone. Hence, these trajectories can be safely considered representative of the motion of the SJ as a coherent airstream only from halfway through the ascent, when the variations in $\theta_w$ to the end of the descent reduce below 1 K. This early lack of coherency does not affect the usefulness of the trajectories for this work as it is during the second part of the ascent and throughout the descent of the SJ that the dry mesoscale instabilities occur along the jet (see Section 3.1.4).

It must be remembered that these are back-trajectories computed from time (3); as already mentioned, they form a very well-defined and coherent structure almost to the start of their descent. We interpret the relative lack of coherency before this as arising from the fact that the air at the start of its descent is within the very tight gradient of the frontal zone in the cloud head. This tight gradient leads to mixing, which may be a combination of numerical artefact from the trajectory calculation and real shear-induced mixing. The broad picture of ascent and potential warming by condensation heating prior to descent is

likely to be reliable, but detail such as changes in horizontal speed less so. This is a limitation of the trajectory technique, not of the model, though of course the treatment of mixing at the frontal boundary in the model may also have some relevance.

In summary, the identified SJ shows an ascent-descent pattern that is consistent with most previous studies, although less intense than the one in windstorm Tini. The rate of the descent reaches values around $20\,\mathrm{hPa\,hr^{-1}}$ for 5 consecutive hours. This descent is associated with a substantial increase in wind speed with values up to $40\,\mathrm{m\,s^{-1}}$. The occurrence of condensation 350 during the ascent of the SJ airstream is evident while any contribution of evaporative/sublimational cooling at the start of the descent is small.

### 3.1.4 Evolution of mesoscale instabilities along the airstream

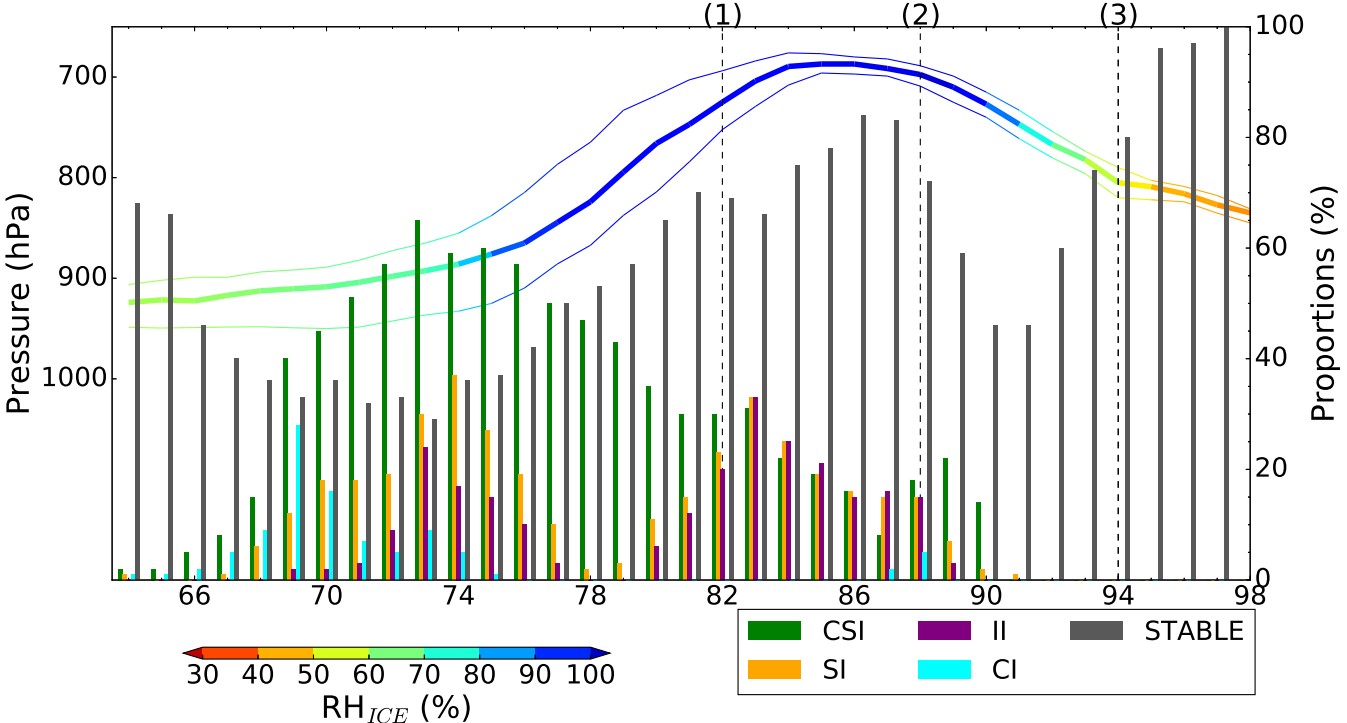

**Figure 5.** Timeseries (hours since start of run) of pressure (colours indicate relative humidity w.r.t. ice) and diagnosis of instability conditions along the trajectories in coloured bars. See section 2.2 for details on instability diagnostics.

Figure 5 shows the time-pressure profile of the SJ airstream overlaid on bars representing the percentage of trajectories unstable to different instabilities at each time, allowing us to assess the evolution of instabilities on the airstream. In the hours 355 preceding the ascent of the SJ the number of trajectories unstable to CSI increases up to a temporary maximum exceeding 60%. At the same time the number of trajectories unstable to SI gets close to 40%. As documented in Section 3.1.3, the evident non-conservation of $\theta_w$ during this period does not allow us to consider the bundle of SJ trajectories as a single coherent airstream. Nevertheless, the large numbers of points where instability to CSI and SI is detected indicates the occurrence of widespread

areas with negative MPV* and PV, respectively . The release of this instability between about 74 and 82 hours from the start is likely to have enhanced the ascent of the air as the trajectories converge to the north of the cyclone centre to form the coherent sting jet air stream that follows the bent-back front. In these early hours the number of trajectories unstable to II is smaller than to SI, see for example at 74 hours from the start. Moist static instability is almost negligible, as indicated by the lack of CI; consequently dry static instability is also negligible. Hence, a negative sign of PV in locations where $\zeta_z$ is not negative (i.e SI in the absence of II) does not come from a negative value of static stability but from negative values of the horizontal components of $\zeta \cdot \nabla \theta$, indicating a very tilted environment in terms of momentum— and $\theta$— surfaces. During the second part of the ascent of the SJ (where it can be considered a single coherent airstream) there is a second build-up of SI and II, both exceeding 30% of trajectories at 83 hours and just exceeding the value for CSI. As the SJ reaches the top of its ascent and then starts the subsequent descent, the values steadily decrease for all these instabilities until all the trajectories become stable. The maximum number of trajectories unstable to dry mesoscale instabilities in the ascended SJ is substantially smaller than in the results of Volonté et al. (2018), where up to 75% of the trajectories of the SJ in the simulation of windstorm Tini were labelled as unstable to SI and II. However, the evolution depicted does suggest that the release of dry mesoscale instabilities such as SI and II takes part in the dynamics of SJ speed increment and descent.

### 3.1.5 Evolution of potential vorticity in the cloud head

Figure 6 displays the evolution of PV at the tip of the cloud head and the associated location of trajectories between 83 and 91 hours from the run start, to investigate the location and extent of unstable regions and their relation to the SJ. The pressure level chosen for each panel is that closest to the median pressure of the SJ trajectories at that time. Figure 5 shows that up to ∼30% of the SJ trajectories become unstable to SI just before starting to descend, at around 83 hours, while Figure 6 displays the time evolution of SI along the trajectories from that time onwards. Elongated regions of negative PV travelling along the frontal zone, just on its outer side, associated with the warm - and then bent-back - front can be seen in all panels. The front is indicated by high PV values, a consequence of large positive values in $\zeta_z$, due to the across-front horizontal wind shear, and a tight horizontal across-front gradient in $\theta_w$. The SJ trajectories travel around the front within the cloud head, along the same path on which the localised regions of negative PV move. When the SJ then reaches the banded tip of the cloud head and exits from it starting to descend, the negative values of PV gradually disappear suggesting a release of SI via the slantwise descent experienced by the SJ trajectories.

Figure 7 shows two cross sections, one along-flow and one across-flow, taken at 83 hours from the run start, i.e. at the time during the SJ ascent when the number of SI-unstable trajectories is maximum. The transects of the sections are marked in Figure 6a. The along-flow section (Figure 7a) shows a band of negative PV centred at around 450 km from point A, located just below 700 hPa and slightly downward tilted. Some of the trajectories travel within this band, forming the 30% SI-unstable subset of the airstream already mentioned. Note that some other trajectories are located ahead of this band, somewhat closer to the tail of other bands of negative PV that extend to the tip of the cloud head (Figure 6a ). In general, the presence of several localised regions with values of PV close to zero or negative suggests that the area close to the bent-back front in the cloud head represents a favourable environment for the existence of SI (or at least reduced symmetric stability). The across-flow section

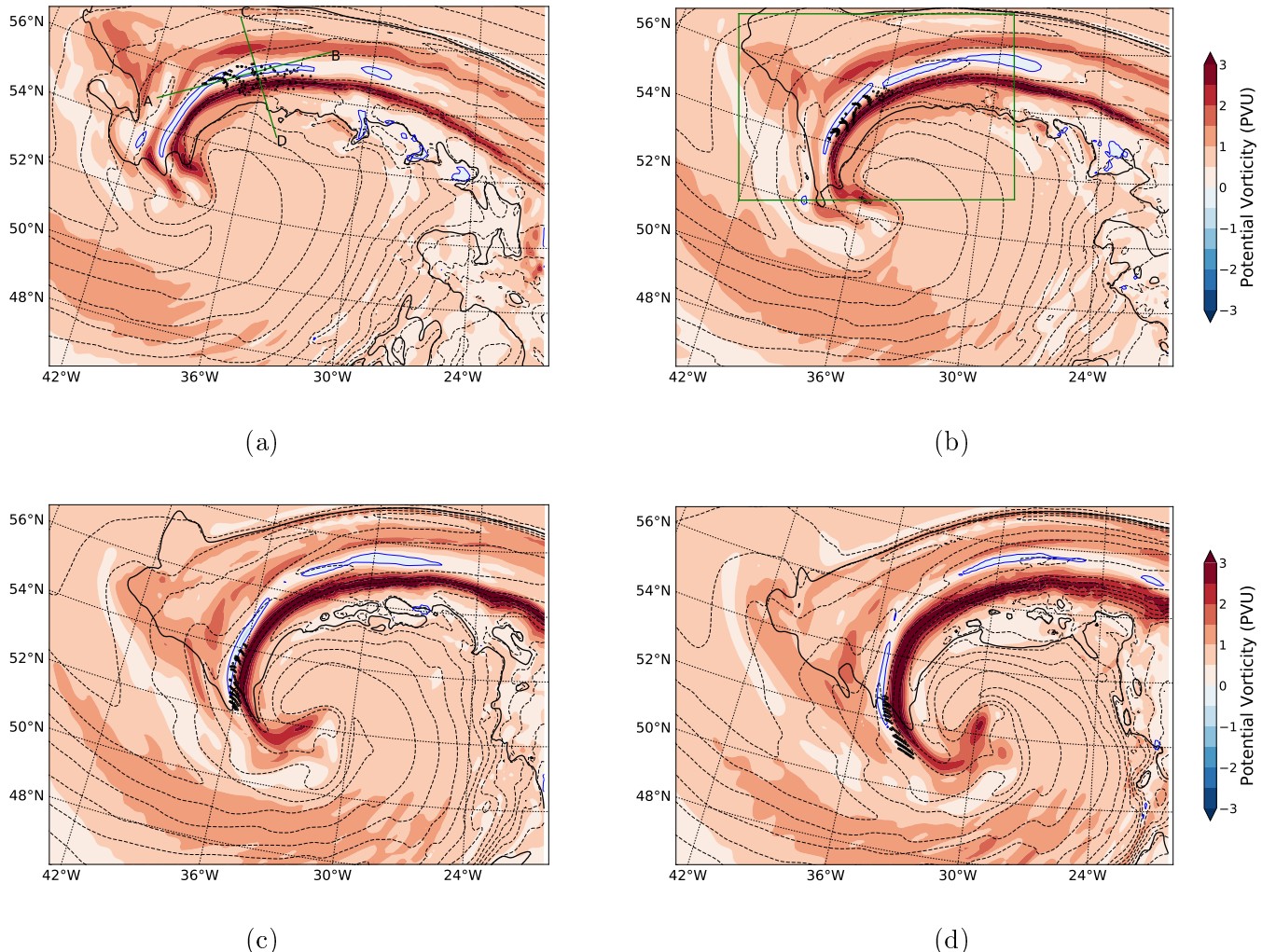

(a)  (b)

(c)  (d)

**Figure 6.** PV (shading, PVU), $\theta_w$ (thin dashed contours every 1 K) and cloudy regions (RH$_{ice}$=80%, black solid contours) at (a) 700 hPa at 83 hours from run start, (b) 685 hPa at 86 hours, (c) 715 hPa at 89 hours, (d) 745 hPa at 91 hours. Black dots show the locations of the trajectories. The two green transects, AB and CD, in panel (a) indicate the locations of the cross sections shown in Figures 7 and 8, while the green box in panel (b) indicates the domain plotted in Figure 9.

(Figure 7b) instead highlights the slantwise tilted dipole of PV located along the narrow frontal zone in which the SJ travels, at around 700 hPa and between 150 and 225 km from point C. The slanting nature of the dipole suggests the occurrence of
ascending and descending slantwise motions orientated with the slope of the frontal zone. In fact, the wind vectors highlight that this negative-PV region co-located with the SJ trajectories is in a narrow region just underneath slantwise ascent on the warm side of the bent-back front (Figure 7b) and which lies above the dominant nearly-horizontal low-level flow along the cold side of the bent-back front (Figure 7a). This pattern of "overall ascent and not much descent" is consistent with having

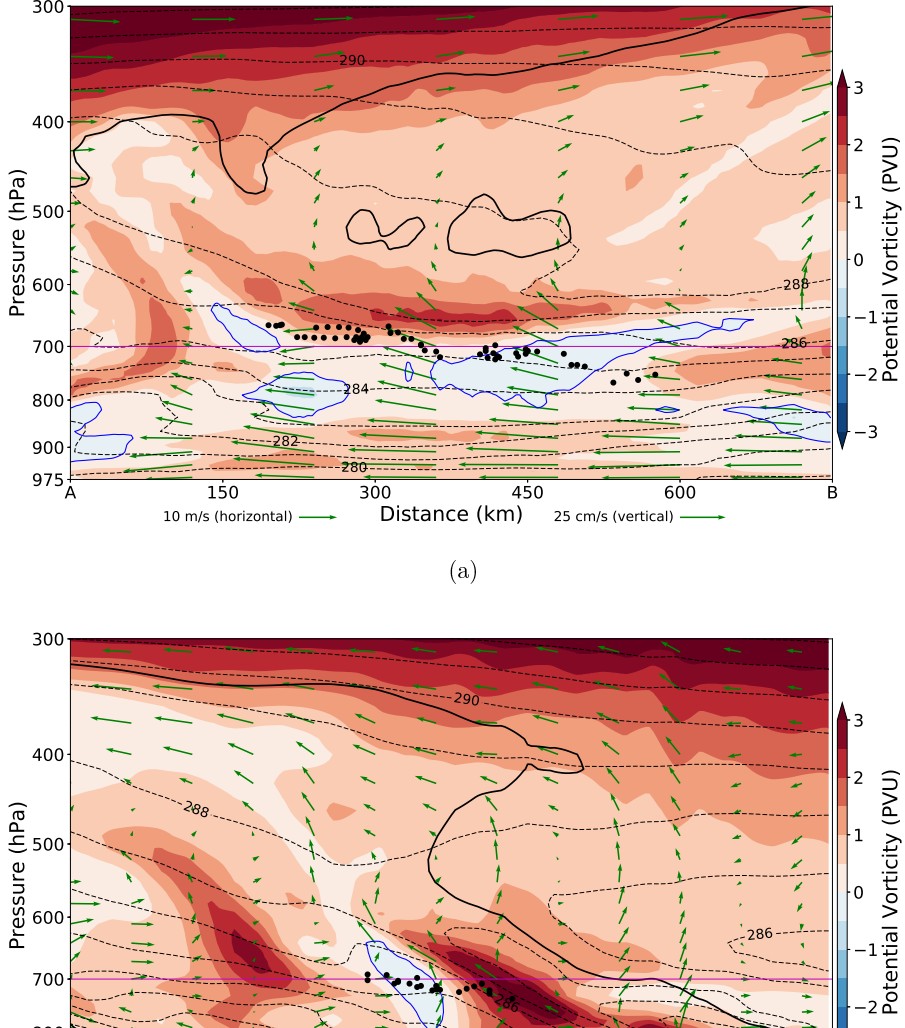

**Figure 7.** Cross sections on transect (a) AB and (b) CD of Figure 6a of PV (shading), $\theta_w$ (thin black dashed contours, K) and cloudy regions (RH$_{ice}$=80%, black bold contours) at 83 hours from run start. Green wind vectors are constructed using the horizontal wind parallel to the section as horizontal component and vertical velocity multiplied by (a) 48 or (b) 24 as vertical component (see scales beneath panels). This scaling has been done to preserve consistency with the section's aspect ratio. Black dots show the perpendicular projection on the transect of the trajectories that are less than 25 km distant from the relevant transect. The magenta line indicates the pressure of 700 hPa, to which Figure 6a refers.

slantwise circulations superimposed on the general ascent taking place in the cloud head at this time, in agreement with the

conceptual model of Browning (2004), see their Figure 14, and his observations of the Great Storm. At the tip of the cloud head this pattern would instead be reversed, with the slantwise circulations superimposed on overall descent (not shown).

Figure 8 shows the zonal and meridional components of relative vorticity along the cross-frontal section and indicates the presence of slanted vorticity dipoles and tripoles in the region of the SJ trajectories. These structures are a direct consequence of the slantwise ascent-descent pattern along the frontal zone just described and highlighted by the wind vectors (even though

the vectors indicate the flow parallel to the section, which is neither zonal nor meridional, and hence do not show the exact component of the wind field generating either of these vorticity components). In detail, the negative bands of the zonal and meridional components of relative vorticity co-located with the trajectory positions are sandwiched between positive bands. These negative regions are compatible with ascent on the southern and eastern side of the frontal surface, respectively, and the associated return circulation i.e. a slantwise direct frontal circulation, with ascent on the southeastern side of the front.

This overall situation is consistent with the mechanism, outlined in Volonté et al. (2018), of slantwise circulations generating negative horizontal relative vorticity, that can be eventually tilted into the vertical to trigger II (and SI too if, as a consequence and as is the case here, not only $\zeta_z$ but also PV decreases below zero).

Analysis of the tilting term of the vertical component of the vorticity equation (Figure 9) indeed confirms the mechanism just described. The vertical component of the vorticity equation (Holton, 2004) can be written as:

$$\frac{\partial \xi_z}{\partial t} + \mathbf{V} \cdot \nabla \zeta_z + \omega \frac{\partial \xi_z}{\partial p} = \mathbf{k} \cdot \left( \frac{\partial \mathbf{V}}{\partial p} \times \nabla \omega \right) - \zeta_z (\nabla \cdot \mathbf{V}) + \text{friction}, \tag{1}$$

where $\xi_z$ is the vertical component of relative vorticity. The first two terms on the right-hand side are the tilting and stretching terms, respectively, and here we consider the tilting term (TTz hereafter) that tilts negative horizontal relative vorticity into the vertical. Figure 9a shows that a narrow and elongated zone of negative TTz values exist on the cold side of the bent-back front at 81 hours when II and SI are increasing along SJ trajectories (see Figure 5). These values are of considerably larger magnitude

close to the cloud-head tip than where the majority of SJ trajectories are. This can be understood by considering that the SJ is still in an environment where ascent is predominant and hence the tilting of the negative vorticity vector from horizontal to vertical is smaller here than further ahead, where descent is prevailing. However, the negative values of TTz close to the SJ core are of similar magnitude to those along SJ trajectories analysed at an equivalent stage of evolution in the study of windstorm Tini by Volonté et al. (2018) (see their Figure 9). Figure 9b shows that at 86 hours the SJ is approaching the cloud-head tip

and the location of its core now corresponds with the sharp minimum in TTz, just on the cold side of the bent-back front. At this time the numbers of trajectories with SI and II are declining, indicating that instability is now being released as SJ descent begins despite still being generated by the tilting mechanism. Overall, the tilting contribution to generation of negative $\zeta_z$ is clear and its pattern consistent with SJ evolution.

In summary, the analysis presented in this section indicates the presence of localised regions of negative PV generated in the

cloud head, where slantwise motions occur along a narrow frontal zone. These regions are associated with a SJ that, travelling within that frontal zone, moves towards the cloud-head tip. During this evolution, negative horizontal relative vorticity is tilted into vertical leading to negative $\zeta_z$ (and PV) along the SJ; the SJ later descends and accelerates while the negative PV ceases

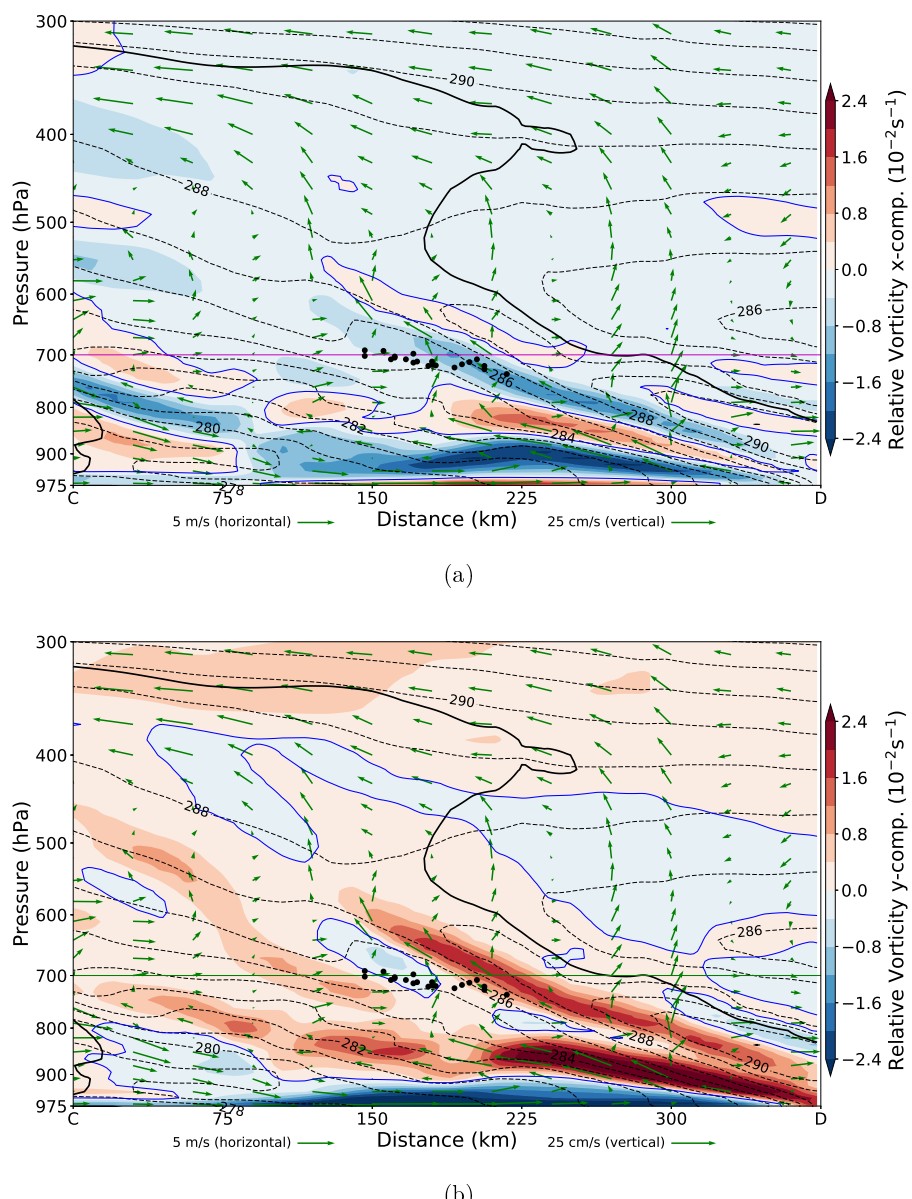

**Figure 8.** Cross sections on transect CD of Figure 6a of (a) zonal and (b) meridional components of relative vorticity (shading), $\theta_w$ (thin black dashed contours, K) and cloudy regions (RH$_{ice}$=80%, black bold contours) at 83 hours from run start. Green wind vectors are constructed using the horizontal wind parallel to the section as horizontal component and vertical velocity multiplied by 24 as vertical component, consistent with the section's aspect ratio (see scales beneath panels). Black dots show the perpendicular projection on the transect of the trajectories that are less than 25 km distant from the relevant transect. The magenta line indicates the pressure of 700 hPa, to which Figure 6a refers.

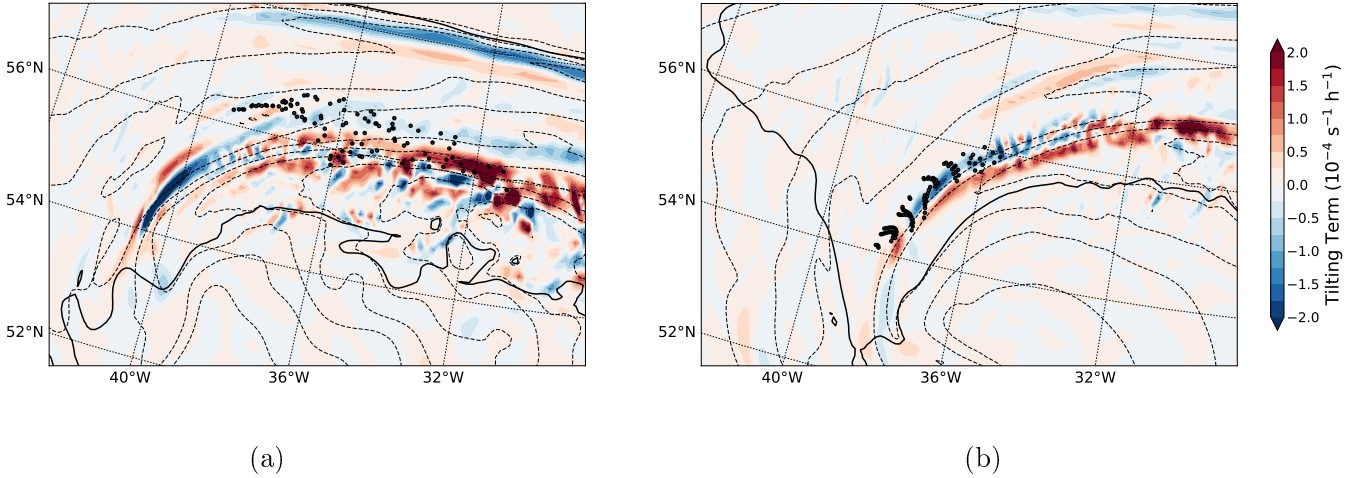

(a)  (b)

**Figure 9.** Tilting term of the vertical component of the vorticity equation (shading), $\theta_w$ (thin dashed contours every 1 K) and cloudy regions (RH$_{ice}$=80%, black solid contours) at (a) 745 hPa at 81 hours from run start, (b) 685 hPa at 86 hours. Black dots show the locations of the trajectories. The domain plotted is indicated in Figure 6b.

to be present. Therefore, the evolution of the SJ airstream identified in the control simulation and its association with the evolution of mesoscale instabilities such as II and SI follows the results and the conceptual model presented in Volonté et al.

(2018), despite the expected differences in magnitude and timing.

## 3.2 Sensitivity experiments

### 3.2.1 Aim and overview

In the previous section it was shown that mesoscale instabilities (conditional and dry) play an active role in the evolution of the SJ in the control simulation. We now assess the robustness of the occurrence of the SJ, along with its strength and connection

with mesoscale instabilities, through the sensitivity experiments described in Section 2.4.

All the simulations display a baroclinic wave evolving according to the Shapiro-Keyser conceptual model (not shown), similar to the evolution shown in Figure 2 for the control simulation. There is variability between experiments in the intensity of fronts and in the shape and size of cloud head and frontal-fracture region, particularly in runs where the initial temperature has been modified (not shown). However, in all simulations it has been possible to identify a low-level (i.e. between 700 and

850 hPa) wind maximum located in the frontal-fracture region (outside the cloud head and above the stable and moist boundary layer) at the time of its widening. Consequently, 100 contiguous grid points have been selected in each run as starting points for the SJ trajectories, following the same procedure used for the control simulation i.e. points contiguous with an Earth-relative wind-speed maximum located close to an area of $\theta_w$-folding in the frontal-fracture region and above the almost-saturated boundary layer; these trajectories tentatively represent the SJ cores. For most experiments the SJ is detected around 93-96

hours from the start of the run. The only simulations showing a markedly different speed of cyclone development and therefore

an earlier or later time of SJ identification are the ones with a modified strength of the upper-tropospheric jet: for example an increase of the initial jet speed of 5 m s$^{-1}$ is associated with an earlier onset of the SJ descent by about 10 hours, and vice versa.

### 3.2.2 SJ evolution

The airstreams tentatively identified as SJs in the different runs show several common characteristics. However, by dividing them in two subsets it is easier to detect noticeable differences in their behaviour. Time series of pressure, wind speed, PV and MPV* along the SJ trajectories for each run are shown in Figures 10 and 11 for the respective subsets of (i) experiments with SJs displaying most of the variability (control simulation plus the runs where jet-stream strength and model resolution have been modified) and (ii) the remaining experiments (control simulation plus the runs where initial surface temperature and 460 relative humidity have been modified).

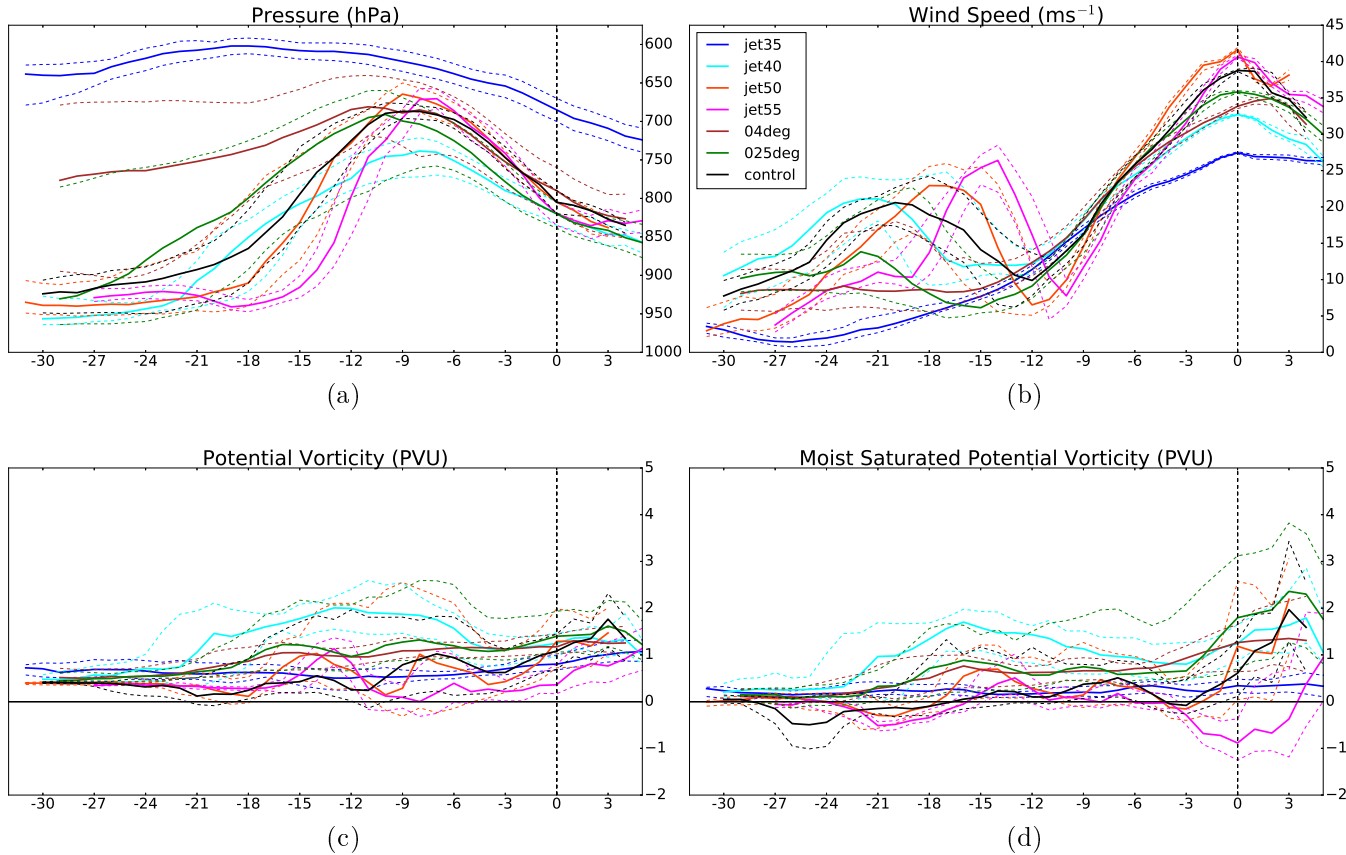

**Figure 10.** Timeseries (hours relative to the time of identification of each airstream) of (a) pressure, (b) wind speed, (c) PV, (d) MPV* along SJ trajectories for the control simulation and for the runs with modified jet stream (jet35, jet40, jet50, jet55) and model resolution (025deg, 04deg). Solid lines indicate the median values of each set of trajectories and dashed lines indicate 25$^{th}$ and 75$^{th}$ percentile values.

Figure 10a shows that in all but one of the simulations the median values of trajectories are located close to 800 hPa at the time of identification, towards the end of their strong descent. The exception is the jet35 run, in which the descent is much weaker and the median of trajectories is around 700 hPa at that time; we later conclude that the airstream identified in jet35 cannot be classified as a SJ. This airstream also does not display an initial ascent (although this is not a requirement for the classification as a SJ), whereas a marked ascent-descent pattern can be identified for the SJ airstreams in all other runs. Median values of trajectories generally rise from below 900 hPa to around 700–750 hPa before descending back to around 850 hPa. The ascent rate of the SJ prior to its descent decreases with decreasing model resolution, particularly noticeable for the 04deg run for which the median of trajectories never exceeds 800 hPa in its early stages and the spread of ascent rates (as shown by the quartile trajectories) is largest. The SJs showing the largest ascent and descent rates are those with the enhanced jet strength: those in the jet50 and jet55 runs. Their second part of ascent and first part of descent are faster than for other SJs, with a total descent around 30–40 hPa larger than for the other SJs. The analysis of the evolution of relative and specific humidity and of $\theta$ and $\theta_w$ (not shown) reveals that these SJs go through an evolution that is similar to that described in Section 3.1.3 for the control run. All selected airstreams (apart from that in jet35) show some initial evaporation/sublimation occurring on the trajectories, prior to the formation of coherent airstreams, followed by saturated ascent in the coherent airstream associated with condensation and precipitation and then an almost-adiabatic descent during which moist processes are absent or negligible (not shown).

The associated evolution of wind speed is shown in Figure 10b. Most of the SJs show a clear oscillation in wind speed centred at around 15 hours before the identification time. This oscillation is associated with the airstreams turning around the cyclone centre (and so changing the alignment of their direction with that of the environmental flow) while travelling in the bent-back cloud head. The magnitudes (and timing) of this oscillation vary between runs, indicating a difference in the SJ paths in their early stages of development. In the runs with coarser resolution this oscillation is much weaker (025deg) or even absent (04deg and jet35). This result suggests a different origin of the airstreams in these runs, and a more zonal path during their evolution, inconsistent with the exit of the airstream from a hooked cloud head (not shown). The oscillation is followed by an acceleration that starts around 10–12 hours before the identification time and ends around identification time, to reach wind speeds of $\sim$35–40 m s$^{-1}$. Although the maximum wind speed varies between runs, the SJs show a remarkable similarity in this strong acceleration (except in run jet35) which can thus be considered as one of the main characteristics of SJ evolution, along with the associated descent. The largest increase in speed occurs in the same experiments as those with the greatest total descent, exceeding 30 m s$^{-1}$ in the SJs in runs jet50 and jet55 compared to 25 m s$^{-1}$ in the control run. Conversely, the experiments with coarser resolution and reduced jet strength display a weaker acceleration.

The occurrence of SI on the SJ airstreams is revealed by the evolution of PV along the trajectories (Figure 10c). In the control run and in the runs with enhanced jet strength (particularly jet55) the 25$^{\text{th}}$ percentile and median values get close to or even below zero (the condition for the onset of SI) during the final part of the ascent and the beginning of the descent, in a time window centred around 10 hours before identification time. This suggests the occurrence and the subsequent release of SI in the SJ in these runs. Conversely, for the runs with coarser resolution and with reduced jet strength, PV values stay well above zero (particularly in run jet40) indicating the absence of substantial SI along the trajectories. The airstream in run jet35 displays

an almost-constant PV throughout all its evolution, indicating the absence of non-negligible diabatic processes throughout its evolution. This, along with the low moisture content of the airstream (not shown) and the absence of a substantial acceleration and a strong descent towards low levels, indicates that this airstream is simply adiabatically advected and weakly accelerated above the boundary layer towards the frontal-fracture region. Hence, it does not fulfil the SJ criteria according to the definition in CG18; instead it might be part of — or associated with — the intrusion of dry air from upper levels.

A similar ocurrence of instability, but at an earlier time, exists for CSI, with 25$^{th}$ percentile and even median values of MPV* in the control, jet50 and jet55 runs below zero at around 20 hours before identification time i.e. at the start of the coherent SJ airstream ascent into the cloud head (Figure 10d). These median values then stay close to zero throughout the time when SI develops. Again, in the other runs these values are well above zero. The drop in MPV* in the jet55 run during the final hours of SJ descent can be ignored as that airstream is far from saturated at that time and hence the conditions for CSI release do not hold. The occurrence of CI is also ruled out as values of moist static stability (not shown) stay well above zero during the whole evolution of the airstreams in all runs, going down to negative values only at the end of the descent for jet55 (when CI cannot be released as the air is unsaturated).

As stated earlier, the selected possible SJ airstreams are less distinct from each another in the other subset of sensitivity experiments (Figure 11). Time series of pressure and wind speed along the trajectories (Figures 11a and b, respectively) show for all runs a similar ascent-descent pattern associated with a strong speed increase following an early oscillation, also quantitatively akin to the evolution described for the first subset of experiments. Hence, the description of the SJ undergoing a coherent saturated ascent associated with condensation and precipitation followed by a nearly-adiabatic descent associated with strong acceleration can be extended to the whole set of sensitivity experiments performed, with the exception of jet35. For SI on the SJs in this experiment subset, the 25$^{th}$ percentile value of PV decreases to or below zero in the time window between 15 and 5 hours before identification time (i.e during the final part of the ascent and the start of the descent) in all runs apart from rh70 and t299 (Figure 11). Hence SI exists on at least a quarter of the SJ trajectories in five out of the seven runs in this subset. Considering CSI instead, Figure 11d shows that for all the SJs the 25$^{th}$ percentile (and for most of them also the median) decreases to negative values of MPV* at around 15–20 hours before identification time, even though it increases soon after in the t299 run. Hence, the occurrence of CSI along the SJ during its ascent is common in these sensitivity experiments.

In summary, the analysis of the evolution of selected physical quantities along the SJ trajectories in the different sensitivity experiments highlights a common behaviour: a saturated-ascent/adiabatic-descent pattern and strong increase in wind speed associated with the descent of the airstream. Additionally, the analysis reveals the existence of different environmental conditions for the occurrence of SI along the SJs during their evolution. In the first subset of experiments, simulations with different percentages of SI on the trajectories are associated with different values in the strength of descent and maximum wind speed. In the second subset of experiments, whilst the range in the percentage of SI on trajectories is still present, the values of wind speed and pressure are instead very similar for all SJs throughout their evolution. As a whole, this indicates that model resolution and jet strength can change the intensity of the SJ whereas other environmental changes (e.g to initial relative humidity or surface temperature), while influencing the dynamics of the SJ generated (as indicated by the variations in SI), do not seem to have a clear effect on its intensity both in terms of peak speed and descent. It is also revealed that lower-tropospheric wind

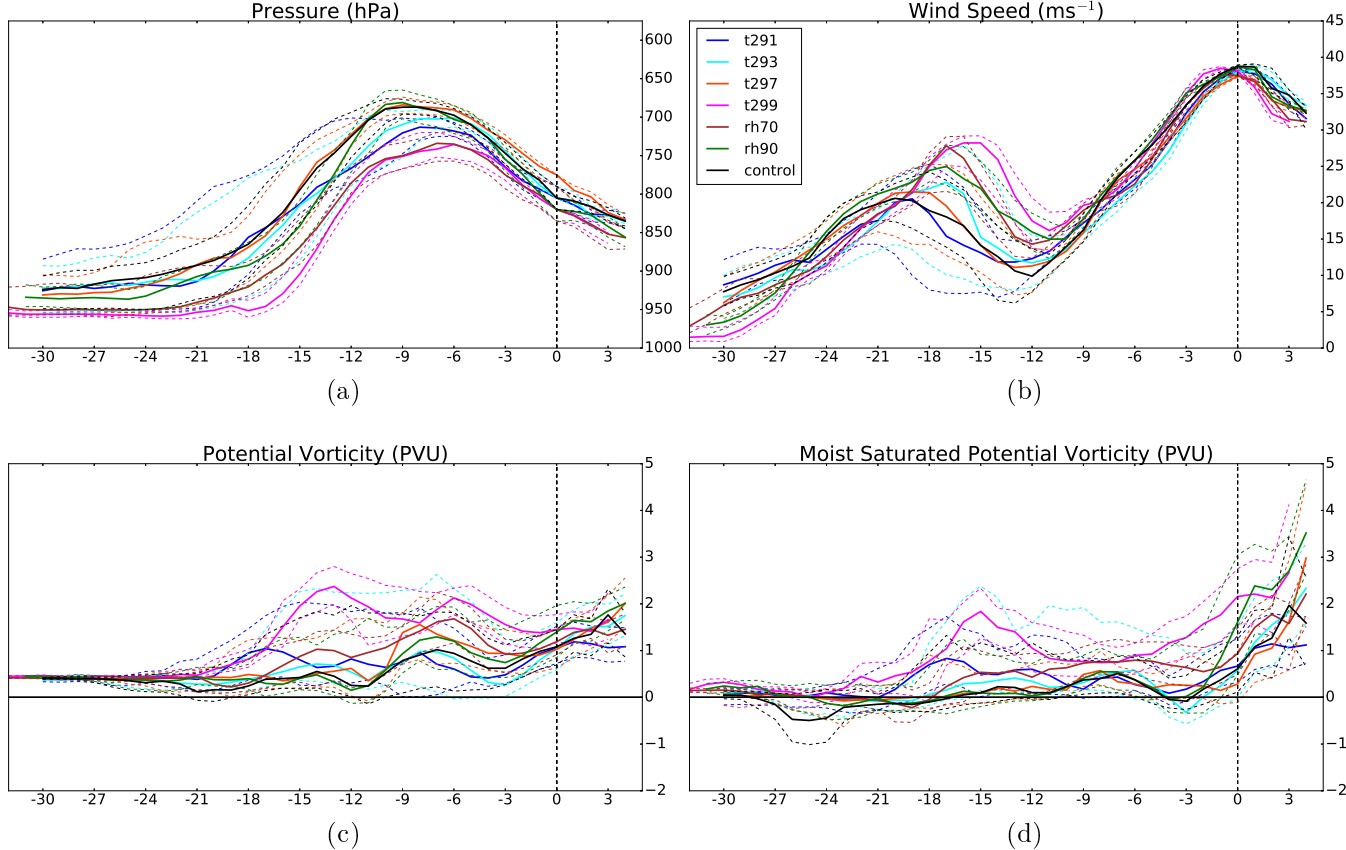

**Figure 11.** As in Figure 10 but for the control simulation and for the runs with modified initial relative humidity (rh70, rh90) and initial temperature (t291, t293, t297, t299).

maxima in the frontal-fracture region can occur from airstreams being zonally and adiabatically transported by a weakly descending and accelerating flow associated with the intrusion of dry air in the same frontal-fracture region (as occurs in run jet35). Hence, consistent with the first modelling study of a sting jet cyclone (Clark et al., 2005), SJ identification requires the wind maxima to be associated with an airstream that exits from the cloud head and undergoes a descent associated with marked increase in wind speed.

### 3.2.3 Synthesis of the results

The control simulation has demonstrated that the SJ is closely associated with a band of negative or zero PV. As discussed in CG18, separating cause and effect is very difficult in that frontal circulations are known to have a strong impact on stability while even balanced dynamics is strongly modified as neutral stability is approached. Nevertheless, it is worthwhile considering what the sensitivity studies can tell us about the relationship between mesoscale instability and SJ properties, here expressed by wind speed and descent and summarised in the panels of Figure 12.

The mesoscale instability metrics used are essentially measures of SI. $trajs_{PV<0}$ is the maximum number of trajectories of the 100 selected with negative PV at any instant within 15 hours from the time when the trajectories are selected (i.e. identification time) and $volume_{PV<0}$ is the maximum percentage of SI-unstable volume in the cloud head around the SJ detected in the same time interval (see caption for more details). These metrics are compared in Figure 12a and found to correlate reasonably well but with substantial variability between cases, so it is likely that there is a degree of arbitrariness in the choice of metric. The simulations where surface temperature has been changed the most display the largest deviations from this correlation. In the following we consider just $trajs_{PV<0}$. The wind speed and descent metrics used are: the maximum Earth-relative wind speed, $|\mathbf{U}|$; the system-relative wind speed at the time of $|\mathbf{U}|$, $|\mathbf{U}|_{sys}$; the maximum pressure increase over a period of five hours ending by the time of $|\mathbf{U}|$, $\Delta p_{5h}$ (i.e the steepest 5-hour descent); and the total pressure increase and speed increment during the whole SJ descent $\Delta p_{descent}$ and $\Delta|\mathbf{U}|_{descent}$, respectively.

Before considering the link between instability and SJ properties in all the simulations together, it is worth summarising results from each type of the experiments presented in the previous section. Most straightforward is the impact of horizontal resolution: lower resolution substantially reduces peak wind speed (both Earth-relative and system-relative, Figure 12b and c) and removes all SI. In the coarser resolution runs SI is absent with both $trajs_{PV<0}$ and $volume_{PV<0}$ equal to zero not only in the SJ but throughout the cloud head, as shown in Figure 12a. As discussed in Section 3.2.2, the impact of background jet strength is also very clear: higher jet strength leads to more unstable, stronger SJ winds (both Earth-relative and system-relative) and much more descent (Figure 12e). The three unstable cases have $|\mathbf{U}|$ 10 m s$^{-1}$ stronger and $|\mathbf{U}|_{sys}$ 5 m s$^{-1}$ stronger than the two stable ones with jet strength 40 m s$^{-1}$ and below; in fact case jet35 has already been noted not to have SJ characteristics at all. The relationship is not linear or even monotonic, as "saturation" appears to happen for the strongest background jet with the SJ in the jet50 experiment having stronger $|\mathbf{U}|$ than that in the jet55 experiment. This reduction is accompanied by a qualitative change in cyclone structure (not shown). Changes in relative humidity have little effect on $|\mathbf{U}|$, but an approximately linear impact on $\Delta p_{5h}$. Notably, the instability marginally increases over the control in case rh90, but decreases much more in rh70. It seems likely that this reflects the non-linearity of the sub-grid cloud condensation scheme. Surface temperature also has relatively small detectable impact: $|\mathbf{U}|$ and instability ($trajs_{PV<0}$) decrease slightly at lower temperatures, but the highest temperature (t299) has the smallest instability (essentially stable) and yet the largest $|\mathbf{U}|$ and $|\mathbf{U}|_{sys}$ values.

When grouped together, the first thing to note is the existence of two subsets of cases: SJs termed here as being either unstable or stable to SI release as determined from $trajs_{PV<0}$ (Figure 12a). The six stable cases comprise two with zero values of $trajs_{PV<0}$ (those in the runs with with coarser resolution than the control run) and four with $0 < trajs_{PV<0} \leq 10$ (those in the runs with reduced jet strength, including non-SJ case jet35, and cases t299 and rh70). In contrast, the seven unstable cases all have $trajs_{PV<0} \geq 25$.

Discounting the non-SJ jet35 case, the $|\mathbf{U}|$ values reached by different SJ airstreams are generally within 37 and 40 m s$^{-1}$ despite the range of environmental conditions (Figure 12b). It is particularly notable that simply increasing the background jet strength does not provide much scope for increase in SJ strength as it eventually leads to a change in cyclone structure. We have yet to find an idealised background state able to produce the more extreme SJs such as the Great Storm of 1987 and cyclone Tini. On average, unstable SJs have larger $|\mathbf{U}|$ values, with a weak tendency for this windspeed to increase with instability.

The emerging picture displays SJ strengths in the mid 30s m s$^{-1}$ in stable cases being enhanced by around 5 m s$^{-1}$ although the variability between experiments reduces the clarity of this effect. However, much of this signal comes from the jet strength and resolution experiments; the remaining two of the stable SJs and four of the unstable ones have very similar $|\mathbf{U}|$ with values between 37 and 39 m s$^{-1}$. We assume that one effect of changing the RH is to change the effective static stability and hence available potential energy of the background state; however, as cloud formation occurs it is difficult to quantify the magnitude of this and there is no obvious dependence of $|\mathbf{U}|$ on either the RH or sea surface temperature.

In Sec. 3.1.3 we showed that, during their descent, SJ trajectories in the control simulation increased their system-relative speed by around 5 m s$^{-1}$. The uncertainty in system speed discussed in Sec. 2.5 means that it is difficult to measure changes in system-relative speed of this magnitude. The relationship between system-relative speed and SI is displayed in Figure 12c. The impact of resolution and background jet strength is, perhaps, clearer than in Figure 12b, but cases rh70 and t299 show even more similarity in terms of $|\mathbf{U}|_{sys}$ to high instability cases. Overall, it is difficult to assert a strong relationship between either Earth-relative or system-relative SJ maximum speed and degree of SI. Some relationship between instability and the magnitude of steepest SJ descent is instead evident in Figure 12d; this would be further strengthened if the resolution experiments were ignored, as clearly resolution changes the overall behaviour of the cyclone in ways different from just changing the background state.

It is not obvious how to isolate the exact role of SI/II in generating the SJ; the basis of our analysis is essentially energetic. At the point where the SJ starts to descend, a substantial proportion of the air eventually leading to the SJ is unstable to SI/II (and so, since it is saturated, to CSI). This has an available potential energy (APE) associated with it, though there is likely to be release going on at the same time as generation so it is difficult to quantify the amount. We have not attempted to quantify the production rate, but have shown that production exists at the point where the SJ starts to descend (and before). As this instability is released, the SJ shows an increase in kinetic energy (KE). A strong correlation between descent and speed increment can be expected from dynamical arguments as the descent is largely radial with respect to the cyclone centre; the speed increment can then be associated with the work done by the horizontal pressure gradient on the ageostrophic motion. However, the ageostrophic acceleration by the horizontal pressure gradient is the only source of horizontal kinetic energy (friction is a sink) whether the flow is balanced or not so this tells us nothing about mechanism. The increase in final wind speed is of course the same whatever direction the wind enhancement is in and, in common with other studies, we have not decomposed the SJ wind speed into tangential and radial components. Assuming that the increase in KE is primarily due to acceleration in the radial direction, if this does contribute to the primarily tangential speed of the SJ then a mechanism must exist to turn the down-pressure gradient flow to the right. This turning may arise from anisotropy in the horizontal stability or due to the Coriolis acceleration. While the instability is fundamentally unbalanced, the descent occurs over a period longer than $f_z^{-1}$, where $f_z$ is the usual (vertical component of) Coriolis parameter so some geostrophic adjustment is to be expected. (It is also worth remembering that the MetUM, unlike many models, also includes the effect of Coriolis acceleration on the vertical motion.) Of course, exactly the same can be said of any mechanism promoting ageostropic flow towards the low centre.

The correlation between total descent and associated speed increment of SJs moving from the cloud head towards the frontal-fracture region is investigated in Figure 12e. Neglecting the zonally-moving non-SJ jet35 case, a somewhat linear relationship

is present overall, though still with substantial noise and no pattern in the surface temperature experiments when taken alone. The anomalously low value of $\Delta|\mathbf{U}|_{descent}$ for jet55 is a consequence of a late start of the SJ descent, possibly associated with a different cloud-head structure, meaning that some of the speed increment of the SJ trajectories is not considered. A clearer relationship is instead visible between $\Delta p_{5h}$ and $|\mathbf{U}|$ (the magnitude of steepest descent and the peak Earth-relative speed) (Figure 12f), although this comparison is less dynamically meaningful than the previous one. We have not attempted to refine these relationships using more objective statistical tests, as we do not feel the heterogenous nature of the data set justifies doing so. We conclude that, although not a strong relationship, there is evidence that weakly enhanced SJ strength is *associated* with increased SI. This relationship is evident despite the complicating effects of the likely changes in other processes such as large-scale and frontal dynamics and mixing in the different experiments. It is not possible to cleanly disentangle the role of frontogenesis/lysis in the SJ descent due to the fine-scale structures generated in the frontogenesis function in simulations with resolved mesoscale instability release (Volonté et al., 2018). The fine-scale structure we see is more complicated than the broader-scale pattern of frontogenesis along the bent-back front and frontolysis in the frontal fracture region found in cyclones without mesoscale instability release such as that simulated by Schultz and Sienkiewicz (2013) and the coarser resolution simulation of Volonté et al. (2018).

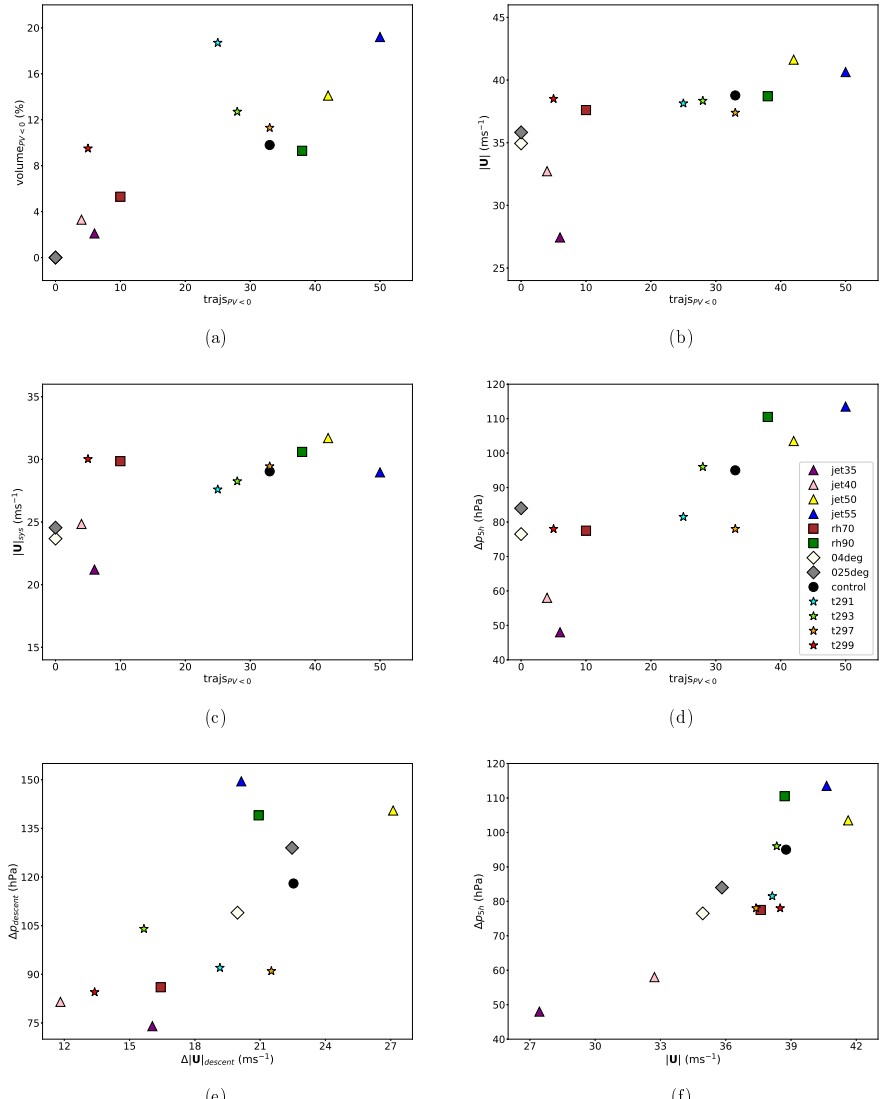

**Figure 12.** Scatterplots showing, for all the sensitivity experiments: (a) $volume_{PV<0}$ against $trajs_{PV<0}$, (b) $|\mathbf{U}|$ against $trajs_{PV<0}$, (c) $|\mathbf{U}|_{sys}$ against $trajs_{PV<0}$, (d) $\Delta p_{5h}$ against $trajs_{PV<0}$, (e) $\Delta p_{descent}$ against $\Delta|\mathbf{U}|_{descent}$ and (f) $\Delta p_{5h}$ against $|\mathbf{U}|$ . $trajs_{PV<0}$ is the maximum number of trajectories of the 100 selected with negative PV at any instant within 15 hours from the time when the trajectories are selected (i.e. identification time) and $volume_{PV<0}$ indicates the maximum percentage of grid points with negative PV, evaluated in the same time window as $trajs_{PV<0}$, in a volume centred on the location of the median of trajectories and extending for 20 grid points in latitude, 40 grid points in longitude and 3 vertical pressure levels (note that the symbols for the reduced resolution simulations overlie at point (0,0) in panel (a)). $|\mathbf{U}|$ is the maximum Earth-relative speed, $|\mathbf{U}|_{sys}$ is the system-relative speed at the time when $|\mathbf{U}|$ is reached, $\Delta p_{5h}$ is the maximum five-hour pressure increase in a period ending by identification time. $\Delta p_{descent}$ and $\Delta|\mathbf{U}|_{descent}$ indicate, respectively, total pressure increase and speed increment during the whole SJ descent, whose end is taken at identification time and whose start has to be not more than twelve hours before. All these kinematic quantities refer to the median of trajectories.

### 3.2.4 Comparison with previous moist idealised studies

A comparison of these results with previous idealised case studies reveals that the typical values of peak wind speed (37-40 m s$^{-1}$) are close to those in the high-high resolution simulation in Coronel et al. (2016) (whose SJ trajectories peak at 40-43 m s$^{-1}$) and more than 10 m s$^{-1}$ higher than in the control simulation in Baker et al. (2014). As the Baker et al. (2014) values are substantially lower than in observed events and simulated SJ case studies (listed in CG18), this is a further proof of the benefits of the improved initial balanced state in producing idealised extratropical cyclones containing a realistic SJ. The maximum values of 5-hour descent (typically around 80–110 hPa in the simulations presented in this study) are instead similar to those in both of these previously-published idealised simulations. Sensitivity runs in Baker et al. (2014) highlight that a reduction in tropospheric static stability substantially increases the strength of descent and acceleration of the SJ, and vice versa. This static stability reduction is associated with a marked decrease in inertial stability and thus indicates a possible role of dry mesoscale instabilities in SJ strengthening. This role is also highlighted here using a more extensive set of simulations, as is the general increase in SJ maximum wind speed and descent rate with increasing upper-level jet strength diagnosed in the additional experiments in Baker (2011).

Coronel et al. (2016) use sensitivity experiments to horizontal and vertical model grid spacing to highlight the model resolution constraints for correct simulation (and in some cases even generation) of SJs in numerical simulations. Our findings are also in agreement with theirs in that our coarser resolution simulations contain the weakest SJs with no sign of SI in their cloud-head environments. Coronel et al. (2016) found regions near-neutral to CSI close to the bent-back slanted frontal zone and revealed the possible occurrence of II in the same area. The work presented here confirms this finding by showing the presence of regions of negative PV and clarifies its generation mechanism, which is found to be consistent with the conceptual model detailed in Volonté et al. (2018). Therefore, while Coronel et al. (2016) emphasise dynamical geostrophic forcing as the main mechanism driving SJ evolution, our work reveals the generation of symmetrically unstable areas in the cloud head and along the SJ, the release of which strengthens the SJ and ultimately shapes its evolution.

In summary, the relevant findings from previous idealised works are consistent with the results presented here. As a consequence of using realistic environmental states and a wide and different set of sensitivity experiments, this study also constitutes a step forward in SJ research as it highlights the role of dry mesoscale instabilities in SJ evolution, clarifying the underlying dynamics while showing the robustness of SJ occurrence in intense Shapiro-Keyser extratropical cyclones.

### 4 Conclusions

Idealised simulations of Shapiro-Keyser cyclones developing a SJ are presented in this study. The setup and initial state of the simulations follow from Baker et al. (2014) which is in turn based on the LC1 baroclinic lifecycle in Thorncroft et al. (1993) (with the addition of moisture). Thanks to an improved and accurate implementation of thermal wind balance in the initial state, it has been possible to use a realistic temperature profile without suffering from the static stability issues that occurred in the Baker et al. (2014) study.

The control simulation produces a cyclone that fits the Shapiro-Keyser conceptual model and develops a SJ (control SJ hereafter) whose dynamics are associated with the evolution of symmetric instability along the airstream. The control SJ shows values of wind speed and descent comparable with previous case studies, including the idealised simulations in Coronel et al. (2016), although smaller than in the simulation of windstorm Tini (Volonté et al., 2018). The control SJ is an airstream that develops from air ascending up to 700 hPa into the cloud head before undergoing a descent of around 95 hPa in five hours while leaving the cloud-head banded tip and travelling towards the frontal-fracture region (a region of buckling of the already-sloped $\theta_w$ lines). While flowing into this region, the SJ accelerates strongly to produce a peak wind speed close to 40 m s$^{-1}$. The control SJ also increases its speed in a system-relative framework, although to a lesser extent than the SJ in the simulation of windstorm Tini. This acceleration suggests a link with mesoscale processes in addition to the synoptic evolution. The occurrence of condensation during the ascent of the airstream is evident while evaporative/sublimational cooling at the start of the descent is much less so.

The analysis of the evolution of mesoscale instabilities along the control SJ shows that, in addition to CSI, both SI and II are present in more than 30% of the trajectories just before the start of the SJ's descent, after which the proportion of unstable trajectories gradually decreases towards zero. This evolution indicates the possible role of dry mesoscale instabilities (such as SI and II) as well as moist instabilities in the dynamics of SJ acceleration and descent, although the maximum number of trajectories unstable to dry mesoscale instabilities is substantially fewer than in windstorm Tini. While Coronel et al. (2016) also reveal the possible occurrence of dry instabilities in the area of the bent-back front, here we were able to assess the underlying mechanism. Negative PV (condition for SI) is generated via vorticity tilting along the airstream flowing through the narrow and slanted frontal zone in the cloud head, before being released during the SJ descent. The evolution of SI thus follows the conceptual model outlined in Volonté et al. (2018).

Sensitivity experiments have been performed by varying model resolution and initial values of upper-tropospheric jet strength, sea surface temperature and relative humidity. All of the 13 simulations display a Shapiro-Keyser cyclone that evolves similarly to the control simulation and show a low-level wind maximum in the frontal-fracture region. In the simulation with the weakest jet stream this wind maximum is associated with a zonally and adiabatically transported airstream possibly associated with dry-air intrusion. In all other 12 simulations the frontal-fracture wind maxima are instead associated with SJs, i.e. airstream exiting from the cloud head and undergoing a descent while markedly increasing their wind speed (see definition in CG18). These experiments thus display the SJ as a robust feature of intense Shapiro-Keyser cyclones, across a wide range of environmental conditions.

Selected physical quantities along the SJ trajectories have been analysed in the different sensitivity experiments revealing a common SJ behaviour: a saturated-ascent/adiabatic-descent pattern and strong increase in wind speed (up to 37–40 m s$^{-1}$) associated with the descent (80–110 hPa/5h) of the airstream. This analysis also shows a range of different environmental conditions leading to the occurrence of SI along the SJs during their evolution. Whilst variations to model resolution and jet strength can change the intensity of the SJ, other environmental changes (e.g to initial moisture or temperature) do not seem to have a clear effect on its intensity both in terms of peak speed and descent while still influencing the dynamics of the SJ generated (as indicated by the variations in SI). Synthesis diagrams aimed at assessing the existence of a relationship between

dynamical drivers and kinematic effects of the SJ evolution in the different simulations demonstrate the existence of two subsets of airstreams: six 'stable SJs' and seven 'unstable SJs' based on the number of trajectories unstable to SI during their evolution. On average, unstable SJs show slightly faster wind speeds and larger descents although substantial noise is present.

In summary, this study uses idealised simulations to explore the evolution of SJs in extratropical cyclones and assess the robustness of their occurrence and dynamics with respect to different environmental conditions. Thanks to an improved initial balance, it has been possible to use more realistic environments than in previous idealised studies (Baker et al., 2014; Coronel et al., 2016). While confirming the relevant findings from previous idealised studies, the choice of a wide and a different range of sensitivity experiments allows further insight into the analysis of SJ evolution and dynamics in different environmental conditions. The conceptual model of SJ generation and strengthening outlined in Volonté et al. (2018) is reproduced by the control simulation, which shows a clear contribution of SI generated by slantwise frontal motions in the cloud head in the evolution of the SJ. The sensitivity experiments reveal the SJ as a robust feature of intense Shapiro-Keyser cyclones, highlighting a range of different environmental conditions (conditional on the model having adequate resolution) in which the release of SI and other mesoscale instabilities contribute to the evolution of this airstream. The airstream's strength is overall weakly enhanced in SJs with increased SI, both in terms of speed and descent. However, several environmental factors modulate this relationship, making it difficult to disentangle the net effect of instability release. While further dedicated studies are needed to fully quantify the effect of dry instabilities in SJ dynamics, this idealised work confirms and clarifies their role in the evolution of the SJ, here shown as a robust feature of intense Shapiro-Keyser extratropical cyclones.

*Code availability.* The source code for the MetUM is available to use. To apply for a license for the MetUM go to: https://www.metoffice.gov.uk/research/approach/collaboration/unified-model/partnership.

For more information on the exact model versions and branches applied please contact the authors

*Data availability.* Post-processed output data from the model simulations are currently stored in a local archive. Their transfer to an externally-accessible location could be a topic of future discussion.

**Appendix A: Model Configuration**

**A1  Model dynamical core**

The MetUM is a finite difference model that solves the non-hydrostatic deep atmosphere dynamical equations on a sphere (White et al., 2005). The equations of motion are based on those of the 'New Dynamics' (Davies et al., 2005), as used by Baker et al. (2014) and are stated here for further use below:

$$\frac{\mathrm{D}u}{\mathrm{D}t} = \frac{uv\tan\phi}{r} - \frac{uw}{r} + f_3 v - f_2 w - \frac{c_{pd}\theta_v}{r\cos\phi}\left(\frac{\partial\Pi}{\partial\lambda} - \frac{\partial\Pi}{\partial r}\frac{\partial r}{\partial\lambda}\right) + S^u, \tag{A1}$$

$$\frac{\mathrm{D}v}{\mathrm{D}t} = \frac{u^2 \tan\phi}{r} - \frac{vw}{r} + f_1 w - f_3 u - \frac{c_{pd}\theta_v}{r}\left(\frac{\partial\Pi}{\partial\phi} - \frac{\partial\Pi}{\partial r}\frac{\partial r}{\partial\phi}\right) + S^v, \tag{A2}$$

$$\frac{\mathrm{D}w}{\mathrm{D}t} = \frac{u^2 + v^2}{r} + f_2 u - f_1 v - g - c_{pd}\theta_v\frac{\partial\Pi}{\partial r} + S^w, \tag{A3}$$

725    where:

$$\frac{\mathrm{D}}{\mathrm{D}t} \equiv \frac{\partial}{\partial t} + \frac{u}{r\cos\phi}\frac{\partial}{\partial\lambda} + \frac{v}{r}\frac{\partial}{\partial\phi} + \dot\eta\frac{\partial}{\partial\eta}, \tag{A4}$$

$$\Pi = \left(\frac{p}{p_0}\right)^{\frac{R_d}{c_{pd}}}; \quad p_0 = 10^5 \text{ Pa}, \tag{A5}$$

and

730    $$\theta_v = \frac{T}{\Pi}\left(\frac{1+\frac{1}{\epsilon}m_v}{1+m_v+m_{cl}+m_{cf}}\right); \quad \epsilon = \frac{R_d}{R_v} = 0.622. \tag{A6}$$

Here, $(u, v, w)$ are the three components of vector wind, $p$ the pressure, $T$ the temperature, $m_v$, $m_{cl}$ and $m_{cf}$ mixing ratios of vapour, liquid water and ice cloud respectively and $c_{pd}$ is the specific heat capacity of dry air at constant pressure. $S^\chi$ represents a source term in $\chi$. Time is denoted $t$. The spherical polar spatial coordinates are $(\lambda, \phi, r)$, while $\eta$ is a generalised vertical coordinate. All derivatives with respect to $\lambda$ and $\phi$ (i.e. 'horizontal' derivatives) are along constant $\eta$ surfaces. The Coriolis 735  terms are given by

$$(f_1, f_2, f_3) = 2\Omega\left(0, \cos\phi, \sin\phi\right) \tag{A7}$$

on an unrotated grid.

The new dynamical core introduced operationally in 2014, ENDGAME, solves the same equations (with more accurate numerical methods) but does not use $\theta$ or $\theta_v$ as prognostic — all variables are related to *dry* density, $\rho_d$ (i.e. the density of the 740  air not including water vapour), so

$$\rho = \rho_d\left(1 + \sum m_X\right) = \rho_d\left(1 + m_v\right), \tag{A8}$$

where $\sum m_X$ is the sum of all water species. This reduces to $m_v$ in our case as we are considering a background state with no pre-existing condensate. Similarly,

$$\theta_{vd} = \theta\left(1 + \frac{1}{\epsilon}m_v\right) \tag{A9}$$

745    and

$$\theta_v = \frac{\theta_{vd}}{(1 + \sum m_X)}. \tag{A10}$$

The equation of state is thus

$$\rho_d = \left(\frac{p_0}{R_d}\right)\frac{\Pi^{\frac{1-\kappa_d}{\kappa_d}}}{\theta_{vd}} \tag{A11}$$

Alternative forms include:

$$p = R_d \rho_d T_{vd} = R_d \rho T_v. \tag{A12}$$

The integration scheme is semi-implicit and semi-Lagrangian (Wood et al., 2014) (though the inherently conservative version was not used as it is very computationally expensive). The model uses Arakawa C staggering in the horizontal (Arakawa and Lamb, 1977) and a terrain-following hybrid-height Charney-Phillips vertical coordinate (Charney and Phillips, 1953). Model parametrizations include longwave and shortwave radiation (Edwards and Slingo, 1996), boundary-layer mixing (Lock et al., 2000), sub-grid cloud condensation (Smith, 1990), cloud microphysics and large-scale precipitation (Wilson and Ballard, 1999) and convection (Gregory and Rowntree, 1990). The initial state described below was implemented in Version 10.5 of the MetUM.

## A2 Thermal wind balance for uniform zonal flow

Since MetUM solves non-hydrostatic deep atmosphere equations on a sphere, traditional derivations of thermal wind balance need some modification. Previous studies using the MetUM (Boutle et al., 2011; Baker et al., 2014) used an approximate solution based on hydrostatic balance in a shallow atmosphere and introduced a 'balancing step' to allow the model to adjust to a true balance. However, we have derived a more accurate initial state which avoids this requirement. For balanced zonal flow $v = w = 0$, $u \equiv u(\phi, \eta)$. With no surface orography $r \equiv r(\eta)$. Then, ignoring source terms, eqs. (A1) to (A3) become, after some rearrangement:

$$\frac{\partial \Pi}{\partial \lambda} = 0 \tag{A13}$$

$$c_{pd}\frac{\partial \Pi}{\partial \phi} = \frac{A}{\theta_v}; \quad A = u^2 \tan\phi - f_3 u r \tag{A14}$$

$$c_{pd}\frac{\partial \Pi}{\partial r} = \frac{B}{\theta_v}; \quad B = \frac{u^2}{r} + f_2 u - g \tag{A15}$$

Eq. (A15) expresses quasi-hydrostatic balance (White et al., 2005).

The next step is to eliminate the Exner pressure by differentiating eq. (A14) with respect to $r$ and eq. (A15) with respect to $\phi$ to obtain, after re-arrangement, the thermal wind relationship:

$$\frac{\partial \ln\theta_v}{\partial \phi} = \frac{\partial \ln B}{\partial \phi} - \frac{\theta_v}{B}\frac{\partial}{\partial r}\left(\frac{A}{\theta_v}\right) = \frac{\partial \ln B}{\partial \phi} + \frac{1}{B}\left[A\frac{\partial \ln\theta_v}{\partial r} - \frac{\partial A}{\partial r}\right] \tag{A16}$$

In Hydrostatic Primitive Equations, $A \equiv -f_3 u a$ (with $a$ the mean radius of the earth) and $B \equiv -g$. Thus the first term on the right hand side is zero. The remaining terms can be combined thus:

$$\frac{\partial \ln \theta_v}{\partial \phi} = -\frac{f_3 a}{g} \theta_v \frac{\partial}{\partial r} \left( \frac{u}{\theta_v} \right) \tag{A17}$$

which is a form of the standard thermal wind equation in height coordinates. Of course, this is more conveniently expressed in hydrostatic pressure coordinates, but this convenience no longer obtains in the deep atmosphere, non-hydrostatic case.

## A3  Balanced potential temperature and pressure profiles.

Our objective is to start with a specified uniform zonal flow and a reference virtual potential temperature profile and derive a full $\theta_v$ field by integrating eq. (A16), then integrating the hydrostatic relationship to get the Exner pressure. Finally we derive the water vapour mixing ratio from the relative humidity field.

Equation (A16) can be integrated to give $\theta_v(r, \phi)$ but we have a choice where $\theta_v(r, \phi)$ matches $\theta_v^{ref}(r)$; in principle, we can integrate both north and south from a reference latitude. In practice, it is easier to integrate just in one direction. We can formally integrate eq. (A16) from the southernmost latitude, $\phi_s$, to latitude $\phi$ thus:

$$\theta_v(r, \phi) = \theta_v(r, \phi_s) \frac{B(r, \phi)}{B(r, \phi_s)} \exp \left( -\int_{\phi_s}^{\phi} \frac{1}{B} \left[ \frac{\partial A}{\partial r} - A \frac{\partial \ln \theta_v}{\partial r} \right] d\phi' \right). \tag{A18}$$

This is an implicit equation for $\theta_v$ if we specify $u$. The first term in the integral is related to the thermal wind. The second is related to the stability. This suggests an iterative approach, using $\theta_v^{ref}$ in this second term as a starting estimate, then refining the estimate using the previous iteration in this term. In practice this iteration has proven unnecessary; using $\theta_v^{ref}$ leads to a well-balanced state and our approximate solution is:

$$\theta_v(r, \phi) = \theta_v(r, \phi_s) \frac{B(r, \phi)}{B(r, \phi_s)} \exp \left( -\int_{\phi_s}^{\phi} \frac{1}{B} \left[ \frac{\partial A}{\partial r} - A \frac{\partial \ln \theta_v^{ref}}{\partial r} \right] d\phi' \right) \equiv \theta_v(r, \phi_s) H(r, \phi) \tag{A19}$$

where we have defined the function $H(r, \phi)$ for convenience below. Clearly, $H(r, \phi_s) = 1$, and the resulting field matches the reference at the southern boundary. The lack of a need to iterate suggests that the change in virtual potential temperature due to the balanced thermal wind has a negligible impact on contribution from the stability term.

For a complete solution we need to know $\theta_v(r, \phi_s)$. Previous studies based on PE integrated from a reference profile on the southern boundary, so $\theta_v(r, \phi_s) = \theta_v^{ref}(r)$ in eq. (A19) and our solution is:

$$\theta_v^1(r, \phi) = \theta_v^{ref}(r) H(r, \phi) \tag{A20}$$

As discussed above, this led to rather cold initial states. We would like to choose a reference profile more representative of the domain centre. Clearly, we can add a constant (i.e. independent of all spatial coordinates), $\Delta \theta_v$ to any solution and still satisfy eq. (A16) if we are using the $\theta_v^{ref}$ on the right had side. Boutle et al. (2011) and Baker et al. (2014) generated a more

representative profile in the domain centre (to some extent) by adjusting the vertical $\theta_v$ profile everywhere using a constant value, $\Delta\theta_v = \theta_v^{ref}(a) - \theta_v^1(a, \phi_0)$, to make that at the surface at the jet centre equal to the reference. Thus:

$$\theta_v(r, \phi) = \theta_v^1(r, \phi) + \theta_v^{ref}(a) - \theta_v^1(a, \phi_0) \tag{A21}$$

where $\phi_0$ is the jet centre and $a$ is the value of $r$ at the surface. This goes some way towards warming the profile, but the result is not guaranteed to match the reference profile anywhere.

The algorithm used in this study seeks instead to adjust the entire vertical $\theta_v$ profile everywhere to make that at the jet centre equal to the reference. Thus, an adjustment is made which is a function of $r$ only. The factor multiplying $\theta_v^{ref}$ in eq. (A19) is just a function of $u$, and hence $r$ and $\phi$, say $H(r, \phi)$. Thus, eq. (A19) gives us:

$$\theta_v(r, \phi_0) = \theta_v(r, \phi_s) H(r, \phi_0) \equiv \theta_v^{ref}(r) \tag{A22}$$

leading to the required solution:

$$\theta_v(r, \phi) = \theta_v^{ref}(r) \frac{H(r, \phi)}{H(r, \phi_0)} \tag{A23}$$

In order to maintain an approach consistent with Boutle et al. (2011) and Baker et al. (2014) we can write this as a correction to the solution $\theta_v^1(r, \phi)$, eq. (A20), thus:

$$\theta_v(r, \phi) = \left(\theta_v^{ref}(r) + \theta_{corr}(r)\right) H(r, \phi). \tag{A24}$$

with

$$\theta_{corr}(r) = \frac{\left(\theta_v^{ref}(r)\right)^2}{\theta_v^1(r, \phi_0)} - \theta_v^{ref}(r). \tag{A25}$$

Again, if eq. (A18) were used, the small dependence of $H(r, \phi)$ on $\frac{\partial \ln \theta_v}{\partial r}$ could be dealt with by iteration but this proves unnecessary.

PE point out that the LC1 setup has zero wind at the surface, so it is sufficient to set the surface pressure to a uniform $p_s = 1000$ hPa. More generally, there is a non-zero zonal wind at the surface. A first estimate of Exner pressure, $\Pi^1$ is computed by integrating eq. (A15) down from the model top:

$$\Pi^1(r, \phi) = \Pi_t + \int_{r_t}^{r} \frac{B}{c_{pd}\theta_v(r, \phi)} dr \tag{A26}$$

where $r_t$ is the model top and $\Pi_t$ the Exner pressure there. This is then adjusted to make the surface pressure at the jet centre equal to the required surface pressure:

$$\Pi^1(r, \phi) = \Pi^1(r, \phi) + \Pi_s - \Pi^1(0, \phi_0) \tag{A27}$$

$\Pi_t$ is thus irrelevant and is set to zero in the code.

Overall, this procedure ensures that the reference surface pressure and $\theta_v$ profile result at the jet centre, while the $\theta_v$ and $\Pi$ fields satisfy quasi-hydrostatic and thermal wind balance.

## A4 Choice of Reference Profiles

PE use a reference profile given by

$$T_r(z) = T_0 + \frac{\Gamma_0}{\left(z_T^{-\alpha} + z^{-\alpha}\right)^{1/\alpha}}, \tag{A28}$$

with $T_0 = 300$ K, $\Gamma_0 = -6.5$ K/km, $z_T = 13$ km and $\alpha = 10$. This gives a constant dry lapse rate in the troposphere, transitioning smoothly to an isothermal layer in the stratosphere. Since Exner pressure is on a different level an iterative method is used to find a consistent integration of the hydrostatic relationship eq. (A15) (with zero wind).

We use the more straightforward piecewise linear profile introduced by Baker et al. (2014):

$$\begin{aligned}
\theta(z) &= \theta_0 + \Gamma_T z; & z &\leq z_T \\
&= \theta_0 + \Gamma_T z_T + \Gamma_S(z - z_T); & z &> z_T
\end{aligned} \tag{A29}$$

with $\Gamma_T = 0.004$ K m$^{-1}$, $\Gamma_S = 0.025$ K m$^{-1}$ and $z_T = 10^4$ m. Baker et al. (2014) use $\Gamma_S = 0.016$ K m$^{-1}$ but that value would result in an unstable profile in the balanced thermal wind setup of this study, as explained below when describing the moisture
profile.

The Baker et al. (2014) jet profile follows the approach of PE. We assume the jet is confined between $\phi_s$ and $\phi_e$. Define

$$\begin{aligned}
\phi^* &= 0 & &; \phi < \phi_s \\
&= \frac{\pi}{2} \frac{\phi - \phi_s}{\phi_e - \phi_s} & &; \phi_s \leq \phi \leq \phi_e \\
&= 0 & &; \phi_e < \phi
\end{aligned} \tag{A30}$$

Then

$$u(r, \phi) = u_0 F_\phi(\phi) F_r(r) + u_s G_\phi(\phi) G_r(r) \tag{A31}$$

where

$$F_\phi(\phi) = \sin^3\left(\pi \sin^2 \phi^*\right) \tag{A32}$$

$$F_r(r) = \left(\frac{z}{z_T}\right)^\gamma \exp\left\{\delta\left[1 - \left(\frac{z}{z_T}\right)^{\frac{\gamma}{\delta}}\right]\right\} \tag{A33}$$

$$G_\phi(\phi) = \sin^2(2\phi)\left(\frac{\phi - \phi_{sh}}{\Delta\phi}\right)\exp\left[-\left(\frac{\phi - \phi_{sh}}{\Delta\phi}\right)^2\right] \tag{A34}$$

$$G_r(r) = \frac{z}{z_s} \tag{A35}$$

$z = r - r_0$, $r_0$ is $r$ at the surface, $z_T$ is the height of the tropopause above the surface, $\phi_{sh}$ is the latitude of the centre of the

shear, $\Delta\phi$ the latitudinal width of the shear, $z_s$ is the scale height of the shear, $u_0$ is the strength of the jet and $u_s$ provides a

meridional shear and, thus, non-zero $u_s$ is used to setup an LC2 base state.

      PE use $\phi_s = 0°$N, $\phi_e = 90°$N (so the jet centre is at $\phi_c = 45°$N), $z_T = 1.3 \times 10^4$ m, $u_0 = 45$ m s$^{-1}$, $u_s = 0$ m s$^{-1}$, $\delta = 0.5$

and $\gamma = 1$. We follow Baker et al. (2014) and use $\phi_s = 15°$N, $\phi_e = 85°$N (so the jet centre is at $\phi_c = 50°$N), $z_T = 10^4$ m,

$u_0 = 45$ m s$^{-1}$ (for the control run), $u_s = 0$ m s$^{-1}$ (for an LC1 setup), $\delta = 0.2$ and $\gamma = 1$.

      The moisture profile is specified in terms of a specified relative humidity field $RH(r, \phi)$. A complicating factor is that the

calculation above has been performed in terms of $\theta_v$, without knowing $m_v$. We adopt an iterative procedure to ensure that the

correct RH profile is achieved while retaining the virtual temperature reference profile.

      The RH profile is specified thus:

$$RH(r, \phi) = RH_0 \left[ 1 - 0.9 R(\phi) \left( \frac{r - r_0}{z_T} \right)^{\alpha} \right] \qquad\qquad\qquad ; r - r_0 \leq z_T$$

$$= 0.0625 RH_0 \qquad\qquad\qquad\qquad\qquad\qquad\qquad ; z_T < r - r_0 \qquad\qquad \text{(A36)}$$

with

$$R(\phi) = 1.0 \qquad\qquad\qquad\qquad\qquad\qquad\qquad\qquad\quad ; \phi < \phi_s$$

$$= 1 - 0.5 \times \frac{\phi - \phi_s}{\phi_e - \phi_s} \qquad\qquad\qquad\qquad\qquad ; \phi_s \leq \phi \leq \phi_e$$

$$= 0.5 \qquad\qquad\qquad\qquad\qquad\qquad\qquad\qquad\qquad ; \phi_e < \phi \qquad\qquad\qquad \text{(A37)}$$

      The parameters chosen by Baker et al. (2014) produce a problem in the stratosphere; even though the RH is small, the tem-

perature in the stratosphere is high enough that the calculated water vapour pressure is high enough to contribute substantially

to the total pressure. The mixing ratio is thus very large (up to 0.4) resulting in low $\theta$ and, ultimately, a statically unstable pro-

file. The model fails after about a day. Setting the initial $m_v$ set to $10^{-8}$ above $z_T$ helps, but we still have a statically unstable

layer. Since we are not very concerned with the precise value of stratospheric stability, we have re-run with $\Gamma_S = 0.025$ K m$^{-1}$.

Using profile correction option 1 we obtain the initial state with a very weakly unstable layer in the stratosphere that runs stably

for (at least) 10 days. Using profile correction option 2 we obtain the initial state shown in Figure 1. This transfers the unstable

layer to the tropical troposphere, but also runs stably for 10 days. If unperturbed, the jet remains essentially unchanged through

this period, demonstrating that the initial setup is, indeed, well-balanced. The exception is the upper stratosphere, which does

develop some small perturbations. However, these have no noticeable affect on the tropospheric flow. We assume that these

perturbations arise from the approximation of eq. (A18) to eq. (A19), but have not verified this as it has no significance on the

results.

      To stimulate baroclinic growth, a small temperature perturbation is applied to the initial state following PE. The perturbation

is independent of height and defined as

$$T'(\lambda, \phi) = T_p \cos(m\lambda) \operatorname{sech}^2[m(\phi - \phi_c)] \qquad\qquad\qquad\qquad\qquad\qquad\qquad\qquad\qquad \text{(A38)}$$

where $m$ is the wavenumber of the perturbation, $\phi_c = 50$ °N is the latitude of the jet centre and $T_p = 1$ K. A wavenumber of $m = 8$ (consistent with the 45 °W–E extent of the model domain) is used in this study as in Baker et al. (2014), chosen to generate cyclones with a smaller length scale than those in Thorncroft et al. (1993) and PE (which used $m = 6$).

*Author contributions.* AV performed the simulations and the data analysis while thoroughly discussing experimental design and results with
PC and SG. PC developed the idealised model and prepared its description for the manuscript. SG's contribution was essential in improving the first version of the manuscript, prepared by AV and reviewed by both PC and SG. All authors were then fully involved in the revisions and the preparation of the final version of the manuscript.

*Competing interests.* The authors declare that they have no conflict of interest.

*Acknowledgements.* The authors wish to thank the Met Office for making the MetUM available and the National Centre for Atmospheric
Science Computational Modelling Services for supporting the academic use of the MetUM on the UK National Supercomputing Service, ARCHER. This work was supported by the AXA Research Fund project "Sting Jet Windstorms in Current and Future Climates." The authors would also like to thank Michael Sprenger (IAC-ETHZ) for his continuous support with the Lagranto code.

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
