# Peer review of "Idealised simulations of cyclones with robust symmetrically-unstable sting jets"

_Weather and Climate Dynamics, 2019_

## Referee Comment (RC1) · Gwendal Rivière (Referee) · 22 Oct 2019

The paper investigates the development of sting jets (SJs) within idealized cyclones simulated with the MetUM model. There are only very few studies on idealized sting jets and such a study is very welcome. The model set up is close to a previous study made by the same team a few years ago (Baker et al. 2014) but some improvements in the definition of the initial state have been accomplished by more accurately setting a thermal wind balance. The implementation of this new initialization procedure allowed the authors to perform a large set of sensitivity experiments by changing the jet strength, the surface temperature, the initial humidity, and the resolution. The description of the initialization procedure is however at times a bit hard to follow. Some steps should be more precisely described to be more readable (see my first main comment).

[Figure]

Before presenting the sensitivity experiments, the study is focused on one control experiment with a given set of parameters. The first objective of the analysis of the control experiment is to show the existence of a sting jet and to present its characteristics. Then, a subsection is dedicated to studying mesoscale instabilities like conditional symmetric instability (CSI), symmetric instability (SI), inertial instability (II) and conditional instability (CI). A deeper investigation of the dry instabilities (SI, II) is performed to show that the mechanism provided by Volonté et al (2018) based on the real storm Tini is herealso at play. This mechanism provides an explanation for the formation of negative regions for the vertical component of the absolute vorticity via the role played by the tilting term in the vorticity equation. I think this part could be improved in terms of the presentation and choice of the figures (see my second main comment). Finally, the analysis of the large set of sensitivity experiments constitutes the most original aspect of the paper. It helps improving the dependence of sting jet formation on large-scale environmental parameters (jet strength, humidity, surface temperature) and model resolution. The analysis is accurately made with a good choice of figures. The main result is that a more intense jet and a higher resolution help increasing both the SJ strength and the volume of unstable regions even though there is no one to one correspondence between the SJ strength and the level of instability. The results of this study certainly improve our understanding of sting jet formation. I consider the paper as worth for publication after considering the following comments, some of them being major. My main criticism is that some paragraphs/sections are tough to read and should be clarified.

Main comments:

1) Presentation of the initialization procedure.

a) Section 2.1.1. Is there a reason why a well-balanced baroclinic jet is more difficult to obtain in the MetUM model ? Since several pages are dedicated to this problem, it would be good to precise if it is a general problem. Please mention if it is a difficulty to be encountered when working with a non-hydrostatic model. I have some doubts because Polvani and Esler (2007) used a primitive equation model and the present

study is said to be inspired by Polvani and Esler (2007). Also, could you mention the type of instabilities met in Baker et al (2014) ? Another aim of the initialization procedure is to set a given surface temperature in the centre of the domain. Lines 100 to 105 are not fully clear to me.

b) Section 2.1.3. I was often confused in this section about what we are looking for. What are the known variables (u ?) and what are the unknown ones (theta_v(r,phi)?) ? Maybe after Eq. (19), some sentences would be useful to add to better understand the steps that are following Eq. (19). In my mind, once u and theta_vˆref(r) are prescribed, we have theta_v(r,phi), so we have everything we need. So I was really lost between lines 185 and 205. And then when the procedure is applied to the chosen case, does Eq. (27) correspond to the definition of theta_vˆref(r) ? Do Eqs. (28) to (33) correspond to the definition of u ? If yes, it means that theta_v(r,phi) is known from Eq. (19), isn't it ? Maybe the details of some steps could be put in an appendix and only the most important aspects of the procedure are kept in section 2.1.3.

2) Mechanism leading to the formation of II and SI unstable regions.

a) Lines 500 to 510 are tough to understand. This is another paragraph where I was lost. I think it would be useful to show the vertical velocity in Figs. 6 to 8.

b) I would encourage the authors to compute the tilting term and to show it in maps. If I well understood Fig.8a corresponds to -dv/dp and Fig.8b to du/dp and then the tilting term should be du/dp domega/dy - dv/dp domega/dx. So we need to visualize omega to understand how the tilting term is. Since it is the key finding of this section, I found a bit too bad not to spend more time to slowly present the whole detailed arguments to the reader.

3) The relationship between descent intensity, SJ intensity and mesoscale instabilities.

a) I agree that if there is a larger volume (or number of trajectories) satisfying the instability criteria, we should see stronger descents. This relationship is clearly seen

in Figure 11d. My concern is more about the Delta|U| during the descent. Why do we expect to get a stronger Delta|U| ? I understand that more unstable regions for SI should lead to more intense transverse circulations but what I do not understand is the acceleration of the along-flow wind speed in presence of such instability. Maybe my question is related to my lack of knowledge on SI in a 3D context, but I think some comments on that would be beneficial for the reader. In other words, the relationship between the slantwise instability across the flow and the along-flow wind speed is not clear to me.

b) Line 725: this is an example of straighforward remark that I do not understand "This acceleration suggests a link with mesoscale processes on top of the synoptic evolution".

Minor comments

1) Line 47: at this stage, it is not clear why a focus is made on dry instabilities rather than moist. Maybe you should mention that you want to check the relevance of a mechanism seen in the real storm Tini.

2) Lines 53-54: I think the end of the sentence has a strange structure and should be reworded

3) Line 82: "the strictly" –> "be strictly"

4) Line 87 "of descent"

5) Line 101: please explain a bit more the problem encountered in Baker et al. (2014). See my main comment above.

6) Line 176: $phi_s$ should be defined.

7) Lines 194-195: I do not fully understand the sentence. By constant value, do you mean independent of x and y ?

7) Line 195: Why is the result of Eq. (19) not enough to define theta_v(r,phi) ? See

main comment above

8) Lines 273-274: what kind of structural changes do we have inside the cyclone by changing the wavenumber ?

9) Line 276: I think the authors should mention they want to check the relevance of a mechanism for the formation of unstable regions for dry instabilities revealed in Volonté et al. Otherwise, it is not clear why a focus is made on dry instabilities

10) Figure 3: It is not clear why levels 850 hPa and 805 hPa are shown. I would prefer to see one level linked to the SJ (e.g., 805 hPa) and another in the boundary layer linked to the cold conveyor belt jet.

11) Line 389: The low-level wind maximum described in the previous paragraphs is the SJ wind maximum, isn't it ?

12) Line 402 and in the rest of the paper. At which level is the SJ wind speed defined ? Is it a given isobaric surface ? Or is it case-to-case dependent ?

13) Line 410: please insert (Fig. 4a) just before ", indicating"

14) Lines 457-458: why does this information important ?

15) Figure 5: At t= 70-75h, before the ascent, the percentage of CSI is quite high. What does that mean ? Is it relevant for the ascent ?

16) Line 504: I do not understand "the direct consequence"

17) Caption Figure 7: "Fig. 5a" –> "Fig. 6a"

18) Caption Figure 8: "Fig. 5a" –> "Fig. 6a"

19) Lines 505-510: paragraph to be reworded by diluting the information (see main comment 2) above).

20) Lines 527-531: how are all these conditions implemented in terms of maths ?

21) Line 662: please be more explicit to say why t299 case is an outlier.

22) Lines 671-673: A large part of the descent is azimuthal as well, isn't it ? These lines are difficult to follow

23) Line 715: I think the static stability issues are never described in the present paper. One or two sentences would be welcome.

24) Line 725: the suggested link is not clear to me (see my main comment above)

reviewed by Gwendal Rivière

---

## Referee Comment (RC2) · Anonymous Referee #2 · 30 Oct 2019

General

The paper presents an important investigation of sting jets (SJs) based on idealized simulations of Shapiro-Keyser cyclones that develop a SJ with the MetUM model. Compared to previous idealized studies, a more realistic representation of a SJ is obtained thanks to improvements of the model initialization. A range of sensitivity experiments is used to explore the environmental control of SJs, such as initial jet velocity, humidity and temperature profiles, and further study the relationship between dry mesoscale instabilities and SJ characteristics. This approach is crucial for understanding what generates the strongest winds, especially given the complexity and high case-to-case variability of previous real case studies. Indeed, SI volume is shown to be consistent with SJ descent through the parameter choices. The paper is overall well-written but

the presentation can be made clearer. The introduction in its present form does not set the scene with a clear story line for the current study, and lacks a clear motivation for specific choices made. These issues should be addressed, and I suggest in the following some ways to improve the clarity of the presentation.

Major comments

1. The background description starts with the cold bias of previous idealized studies. The continuation of Sections 2.1.2 and 2.1.3 drowns in details that divert the attention to configurations of the MetUM and the new thermal wind balance for deep flows, without keeping a line of thought of how this helps to solve the bias and different from the previous idealized simulations. The introduction already becomes very long before the mesoscale instabilities are introduced (which are the focus of this study), and is in many places tough to read, with long sentences and many equations that are not essential for understanding the dynamics in the Results. This part of the introduction should be rewritten and I suggest - to move substantial parts to an appendix and leave only concise information in the main text - especially the information that relates to correcting the temperature bias. - add the humidity profile to Fig. 1, Additionally it would be good to visualize the profiles of the present control to previous simulations and those in the sensitivity experiments. The figure(s) can replace some of the text or analytical expressions. - Break down long sentences (e.g., L51-54, L70-73).

2. It is not fully clear why you state that you focus on dry instabilities. Please clarify what you mean by dry instabilities on lines 75 and 276, which is currently confusing as you discuss also CI and CSI in the same paragraphs, and essentially through most of the results until the synthesis in Fig. 11. The results in Fig. 11 then raise the question of how CSI release in the ascent phase is related to the SJ characteristics. I therefore suggest to either justify the focus on the dry instabilities, or expand the analysis to encompass moist instabilities among the sensitivity experiments.

3. L. 278 and Section 2.3: Please explain the rationale behind attributing the instabilities to slantwise trajectories of the SJ. This is non trivial and especially confusing, considering that CI is defined by theoretical purely-vertical motion of an air parcel.

Minor comments

- You find a coherent saturated ascent - adiabatic descent pattern of SJ trajectories. Related to major comment 2, the reader is left wondering about the role of the initial ascent and its variability among the experiments, and how they relate to other SJ characteristics. Please elaborate on this.

- L4: "and different" is unclear and can be deleted.

- L 82: replace 'the' by 'be'.

- L103: replace "former study" by the reference, to avoid confusion.

- L141: why do you choose the reduction to m_v? how is this consistent with RH wrt ice that is used?

- L194, L198-199: these sentences are unclear. A figure (major comment 1) can help to clarify.

- L249: add 'd' to 'achieve'.

- L267: replace 'significant' by 'significance'.

- L310: reword to "... and with accordingly coarser vertical resolution (as shown..."

- Fig. 2: Mark the frontal fracture region onto the figure.

- Fig. 3: Mark 'SJ' on the wind maximum in 3c.

- L396: delete 'and see the list of other studies'.

- L401: add 'h' after 94.

- L427: replace 'magnitude' by 'rate'.

- L475: delete 'the' after 'until'.

- L501-502: This is not clear. The slope is directed across the slantwise descending flow. Please clarify.

- Fig. 8 caption: replace '5a' by '6a'.

- L505-508: These arguments are hard to follow. I suggest to add the vertical velocity onto the CD profiles.

- L569: replace 25 by 45.

- Fig. 11: The legend should have no lines. In caption to (a), note that the 04deg and 025deg overlap at (0,0).

---

## Short Comment (SC1) · 15 Nov 2019

**Review of 'Idealised simulations of cyclones with robust symmetrically-unstable sting jets' by Ambrogio Volonté, Peter A. Clark, and Suzanne L. Gray**

**Overview:**

Idealized simulations of cyclones are analyzed here with particular focus on the cause of the low-level wind jets. It is concluded that the strong winds resulted from a sting jet. The diagnostics used to analyze this case have been used previously in other published cases of sting-jet cyclones by these authors. There is a rather significant problem with the model simulation initial condition that will require all the simulations to be performed again. The interpretation of the results is a little superficial in places (see detailed comments) and it is unclear why the focus is on dry symmetric instability when moist instability, synoptic-scale forcing, and frontal forcing cannot be ruled out; further diagnosis will be required. The literature cited is incomplete, ignoring previous contributions by other authors, neglecting other cases from the literature that are inconsistent with their results, and not citing contradictory statements from the authors' own research.

Overall, therefore, I find the argument plausible but unconvincing. More precision is needed, or, at least, more caution. Further calculations of the other factors leading to sting jets is needed. While interesting model simulations undoubtedly exist here, I believe that the degree of revision needed constitutes at a minimum 'major corrections', if not 'reject and resubmit'.

**Major comments:**

**L47** Given that the authors had emphasized the importance of moist symmetric instability in sting jets in their previous publications (e.g., the sting-jet precursor depends strongly on the occurrence of CSI) and in L314–315 ("moist processes occurring in the cloud head have a primary role in the evolution of the cyclone in which the SJ occurs and are instrumental in the SJ generation mechanism"), this emphasis on the dry instabilities only is unclear to this reader. Figure 5 shows that between (1) and (2) that CSI, SI, and II are all equally important. Moist instabilities need to be considered equally, and the focus of the manuscript needs to be changed accordingly, requiring substantial rewriting.

**Figure 1** The layer between 400 and 300 hPa appears to be absolutely unstable (i.e., potential temperature decreasing with height). Abrupt and nonuniform gradients of static stability occur within the stratosphere, as well. With the emphasis on the role of instabilities in the cyclone, initializing a model with such a large region of instability should raise a concern. Other initial conditions for idealized cyclones do not show static instability in the upper troposphere (e.g., Fig. 3 of Thorncroft et al. 1993; Fig. 3 of Schultz and Zhang 2007, DOI: 10.1002/qj.87; Fig. 1 of Coronel et al. 2016), so why the authors chose such an unusual set of initial conditions is unclear. Even the set of initial conditions from Polvani and Esler (2007, their Fig. 2) – which the authors claim their "initial base state...is inspired by" and is in return inspired by

that of Thorncroft et al. (1993) – has smooth potential temperature gradients throughout the stratosphere and no instabilities in the troposphere. (With such large differences in the initial base states, in what way were your initial conditions "inspired"?) The initialization of such a deep layer of absolute instability then raises the question of whether the model is initialized with any moist instabilities, an analysis of which is lacking in the present manuscript. The model simulations should be redone with any dry or moist instabilities absent in the initial conditions.

**Figure 4 and its accompanying text** In Clark and Gray (2018, p. 954), the authors write about a modeled sting jet in which "the acceleration amounts to no more than about 2 m/s/hr, but acts over a very slow descent (over more than 12hr); so these trajectories only loosely resemble SJs in observed systems." The air in Figure 4 descends over a 12hr period (86–98 h) and accelerates from 20 m/s to a maximum of 38 m/s (1.5 m/s/hr). Therefore, to be consistent with the authors' previous publications, the trajectories within this simulation should be described within the present manuscript as "loosely resembling a SJ in observed systems".

**L341** Can the authors clarify within the text what they mean by "irregularities"?

**Section 2.1** Maybe I missed it, but can the authors state how the lower boundary condition was modeled? Is it flat land or ocean? How is the temperature of the surface specified, and is it fixed over time or allowed to vary? How are heat and moisture fluxes handled at the surface?

**L383** The trajectories for the sting jet are selected from a height of 805 hPa. Given that a sting jet is a surface expression of a region of strong winds (and hence the near-surface damage potential), it is unclear why the height for selecting sting-jet trajectories is so high. Should these not be selected much closer to the ground, or at least immediately above the boundary layer rather than a height of about 2 km above the surface?

**L434** The authors report that there is no clear cooling signal due to evaporation or sublimation. However, in Clark and Gray (2018, p. 948) they write that "it is not clear that this [sublimation of ice] differs dynamically from CSI." Given that they've chosen not to focus on CSI in the present paper, yet Figure 5 shows 20–30% of descending air parcels have CSI, how do the authors reconcile these apparent contradictory statements and model results? Is CSI the dynamical equivalent of the sublimation of ice, as they previously wrote? And, is CSI/sublimation present (or important) in the simulations described within this paper or not?

**L476–479 and throughout the manuscript** The authors state the number of trajectories unstable to dry mesoscale instabilities is "substantially smaller" than that of a previous case and therefore concludes that "the release of mesoscale instabilities such as SI and II takes part in the dynamics of SJ speed increment and descent". These two statements would seem to be contradictory. Moreover, these

statements also contradict, for example:

- L8–9: "A substantial amount of SI...is released along the SJ during its descent...."
- L519–520: "mesoscale instabilities...play an active role in the evolution of the SJ...."
- L667–668: "it is difficult to assert a strong relationship between...SJ maximum speed and degree of SI."
- L683–684: "there seems to be overall some evidence that weakly enhanced SJ strength is associated with increased SI".

Given the degree of inconsistency among these various statements within the manuscript, is the word "robust" in the title of this manuscript appropriate?

Thus, more clarity and consistency on the degree and importance of the dry (and moist) instabilities in relation to the acceleration of the SJ is needed throughout the manuscript.

**Section 3.2** Here it is concluded, "There seems to be overall some evidence that weakly enhanced SJ strength is *associated* with increased SI, but clearly other processes are occurring in the different cases to complicate behaviour." It's not clear to me how I've learned anything useful from this analysis. First, weak and vague words ("seems to be", "some evidence", "associated", "other processes", "complicate behavior") obscure the meaning of this sentence and the actual results. Greater precision is needed when writing such important conclusions.

Second, given that the authors admit that synoptic-scale and frontal-scale circulations are in part responsible for this descent (L618–619) and that the initial strength of the jet stream, and hence of the cyclone, has changed in these simulations, then the authors cannot rule out that the magnitude of the forcing has increased and is the major contributor to the differences in the accelerations of the jets across these sensitivity experiments. The analysis within the present manuscript does not tell us what is causing the acceleration to be strong in these regions. Schultz and Sienkiewicz (2013) discuss this issue in their paper (see p. 604). At a minimum, the authors would appear to need to calculate the synoptic and frontal forcing to determine if changes in these can explain their model results. Simply concluding that "several environmental factors modulate this relationship, making it difficult to disentangle the net effect of instability release" (L765–766) undermines the basis of their study. Such a sentence would appear to be a weak concluding statement when such factors could be calculated and examined. After all, the purpose of a scientific paper should be to shed light on these factors for the benefit of the readers, rather than be defeated and conclude such a problem is intractable.

**L16–17, but throughout the manuscript** Are you comparing with other analyzed cases of sting jet cyclones here when you say that the sting jet in your case is robust? If so, then the literature is not so clearly uniform on this issue of instabilities in cyclones. Some model simulations of real cyclones with sting jets (e.g., Smart and Browning 2014; Brâncuş et al. 2019) found little to no instability associated with the sting jet. What does the absence of instabilities in observed cases mean for the authors' conclusions? The authors should express that ambiguity more clearly throughout the manuscript because idealized simulations, as informative as they are and therefore commonly used, do not often represent reality. More care needs to be taken to avoid overgeneralizing the results of this study.

**Minor comments:**

**L25–26** Schultz and Browning (2017) (DOI: 10.1002/wea.2795) argue that one cannot identify a sting jet from the surface observations alone and should be cited here.

**Section 1** It would seem appropriate to cite the comprehensive review of conditional symmetric instability (as well as other instabilities) by Schultz and Schumacher (1999) (DOI: 10.1175/1520-0493(1999)127<2709:TUAMOC>2.0.CO;2) somewhere in the introduction.

**L46** I suggest that the paper by Schultz and Sienkiewicz (2013) (DOI: 10.1175/WAF-D-12-00126.1) is cited here as this is the first paper I know of that has discussed the importance of frontolysis.

**L81** Brâncuş et al. (2019) (DOI: 10.1175/MWR-D-19-0009.1) and Eisenstein et al. (2019) (DOI: 10.1002/qj.3666) also considered the importance of these instabilities and should be cited here.

**L427** There are many other papers on modeled sting jets that could be included here - all consistently showing that the descent here is consistent with that in previous studies. Thus, it is not correct to say that the results found in this case are the same as in other cases and cite only the Volonté et al. paper. Whether or not you do this, I suggest you reference the paper by Slater et al. (2017) (DOI: 10.1002/qj.2924). The Slater et al. paper considers the same case study as the Volonté et al. (2018) paper and has higher descent rates and accelerations to that being cited here. The authors would argue that the model used to analyse the Slater et al. cyclone has insufficient resolution to allow a sting jet to form if associated with any mesoscale instability, but it does have the resolution to produce descent and acceleration due to frontolysis.

**L434** There are many other papers on observed and modeled sting jets that could be included here - all consistently showing that cooling is minimal (e.g., Smart and Browning 2014; Coronel et al. 2016; Slater et al. 2017; Brâncuş et al. 2019). On the other hand, Eisenstein et al. (2019) found cooling was much more important in their

case. This diversity of results should be discussed.

**L614–615** Schultz and Browning (2017) (DOI: 10.1002/wea.2795) argued that the wind maximum of a SJ needed to exit the cloud head and accelerate, and should be cited here.

**L704–705** In addition to the citation to Coronel et al. for recognition of the importance of the synoptic-scale forcing, I suggest adding a sentence citing the paper by Schultz and Sienkiewicz (2013) (DOI: 10.1175/WAF-D-12-00126.1) as the first paper I know of that has discussed the importance of frontolysis as a forcing mechanism for sting jets.

**L720** There are many other papers on modeled sting jets that could be included here - all consistently showing that the descent here is consistent with that in previous studies. Whether or not you do this, I suggest you reference the paper by Slater et al. (2017) (DOI: 10.1002/qj.2924). The Slater et al. paper considers the same case study as the Volonté et al. (2018) paper and has higher descent rates and accelerations to that being cited here.

**L731–732 and throughout the manuscript** Not all sting jet cases are associated with dry instabilities. You should cite relevant literature by other authors that show other cases with negligible amounts of these instabilities (e.g., Smart and Browning 2014; Brâncuş et al. 2019). Consider other statements within the manuscript that should be similarly reworded with additional caveats and citations.

---

## Author Comment (AC1) · 17 Jan 2020

**Response to reviews of "Idealised simulations of cyclones with robust symmetrically-unstable sting jets" by Volonté et al. wcd-2019-8 - A. Volonté on behalf of all authors, 17 January 2020**

**Dear Editor,**

We thank the two reviewers (Gwendal Rivière and an anonymous reviewer) for their positive and constructive comments on our manuscript which have improved it. Below we give our responses to each of these reviewers in turn. We have also considered the additional "short comment" from Prof Schultz.

Our responses to the reviewers are given in *green italic* font, while new text in the manuscript is indicated in red, here and in the manuscript.

**Reviewer 1**

**Gwendal Rivière**

The paper investigates the development of sting jets (SJs) within idealized cyclones simulated with the MetUM model. There are only very few studies on idealized sting jets and such a study is very welcome. The model set up is close to a previous study made by the same team a few years ago (Baker et al. 2014) but some improvements in the definition of the initial state have been accomplished by more accurately setting a thermal wind balance. The implementation of this new initialization procedure allowed the authors to perform a large set of sensitivity experiments by changing the jet strength, the surface temperature, the initial humidity, and the resolution. The description of the initialization procedure is however at times a bit hard to follow. Some steps should be more precisely described to be more readable (see my first main comment). Before presenting the sensitivity experiments, the study is focused on one control experiment with a given set of parameters. The first objective of the analysis of the control experiment is to show the existence of a sting jet and to present its characteristics. Then, a subsection is dedicated to studying mesoscale instabilities like conditional symmetric instability (CSI), symmetric instability (SI), inertial instability (II) and conditional instability (CI). A deeper investigation of the dry instabilities (SI, II) is performed to show that the mechanism provided by Volonté et al (2018) based on the real storm Tini is here also at play. This mechanism provides an explanation for the formation of negative regions for the vertical component of the absolute vorticity via the role played by the tilting term in the vorticity equation. I think this part could be improved in terms of the presentation and choice of the figures (see my second main comment). Finally, the analysis of the large set of sensitivity experiments constitutes the most original aspect of the paper. It helps improving the dependence of sting jet formation on large-scale environmental parameters (jet strength, humidity, surface temperature) and model resolution. The analysis is accurately made with a good choice of figures. The main result is that a more intense jet and a higher resolution help increasing both the SJ strength and the volume of unstable regions even though there is no one to one correspondence between the SJ strength and the level of instability. The results of this study certainly improve our understanding of sting jet formation. I consider the paper as worth for publication after considering the following comments, some of them being major. My main criticism is that some paragraphs/sections are tough to read and should be clarified.

**Main comments:**

1) Presentation of the initialization procedure.

a) Section 2.1.1. Is there a reason why a well-balanced baroclinic jet is more difficult to obtain in the MetUM model? Since several pages are dedicated to this problem, it would be good to precise if it is a general problem. Please mention if it is a difficulty to be encountered when working with a non-hydrostatic model. I have some doubts because Polvani and Esler (2007) used a primitive equation model and the present study is said to be inspired by Polvani and Esler (2007). Also, could you mention the type of instabilities met in Baker et al (2014)? Another aim of the initialization procedure is to set a given surface temperature in the centre of the domain. Lines 100 to 105 are not fully clear to me.

As stated, the issue is that the MetUM uses a non-hydrostatic deep-atmosphere equation set. The deep-atmosphere aspect includes terms that are not present in the primitive equations (which are, of course, approximations!). Thus, 'balance' is simply different in the two models, and the state derived by PE is not a balanced state in the MetUM. It is perhaps an indication of why the MetUM does not make the primitive equation approximations that this imbalance is actually quite significant. Probably because the MetUM was the only operational model using the non-hydrostatic deep-atmosphere equation set when this work was started, we know of no published equivalent derivation of a balanced initial state, so have included it for completeness. However, this is not the key part of this paper, so for clarity we have moved the derivation to an Appendix and simplified it somewhat. We have also added the following to Sec 2.1.1:

"The initial base state used is generated using an algorithm based on, and ultimately very close to, that of Polvani and Esler (2007), referred to as PE hereafter. However, two issues arise from their formulation. The first is that their derivation is based upon geostrophic thermal wind balance in hydrostatic primitive equation (HPE) set; as is well-known (White et al, 2005), in part this approximate set ignores various terms, including some Coriolis terms and some metric terms. As a result, a 'balanced' state in the HPEs is not perfectly balanced in a less approximate equation set (and vice versa). The MetUM solves the non-hydrostatic deep atmosphere dynamical equations on a sphere, so the balanced initial state found by PE is only approximately balanced when used in the MetUM. This initial imbalance is quite unphysical and generates an initial transient flow (largely gravity waves) which, at the resolutions used, can result in an unstable response. Baker et al (2014) followed the pragmatic approach of first running for a day or so in a lower resolution version of the model, which was found to sufficiently damp the transient response and settle to a balanced field that could be numerically interpolated to the required grid. However, we have derived an equivalent to geostrophic thermal wind balance in the non-hydrostatic deep atmosphere and hence removed this initial imbalance. To our knowledge this has not been previously published, so we include the derivation in Appendix A. Ultimately, the initial state we produce is essentially equivalent to that of PE but for a non-hydrostatic, deep atmosphere model.

A more significant issue with the initial conditions used by Baker et al (2014) was their rather cold overall temperature, discussed above. This arises from the PE initial state; it is of only minor importance in PE's dry simulations but restricts the magnitude of diabatic forcing by phase changes in Baker et al (2014). In this study a method to adjust the entire vertical virtual potential temperature (\$\theta\_v\$) profile to make that at the jet centre equal to the reference has been devised, the details of which are also given in Appendix A. In any configuration, thermal wind balance tends to reduce the static stability in the upper troposphere to the south of the jet compared with that to the north - we have used a state which takes this to the extreme of being

close to neutrality in the upper troposphere south of around 40 N. In fact, it is very weakly statically unstable, but not sufficiently to generate either a dynamical response or significant unstable mixing through the turbulence scheme, and, if unperturbed, it was found that the whole state remained essentially unchanged through a 10-day integration apart from some very minor wave activity in the upper stratosphere. As a result, this state represents probably the warmest we could achieve without taking a different approach from PE."

b) Section 2.1.3. I was often confused in this section about what we are looking for. What are the known variables (u?) and what are the unknown ones (theta\_v(r,phi)?) ? Maybe after Eq. (19), some sentences would be useful to add to better understand the steps that are following Eq. (19). In my mind, once u and theta\_v^ref(r) are prescribed, we have theta\_v(r,phi), so we have everything we need. So I was really lost between lines 185 and 205. And then when the procedure is applied to the chosen case, does Eq. (27) correspond to the definition of theta\_v^ref(r) ? Do Eqs. (28) to (33) correspond to the definition of u ? If yes, it means that theta\_v(r,phi) is known from Eq. (19), isn't it ? Maybe the details of some steps could be put in an appendix and only the most important aspects of the procedure are kept in section 2.1.3.

As discussed above, we have moved this into an Appendix and restructured, and, hopefully, made the text clearer. Everything in the comment above is nearly correct; eq. (19) is key. This can be integrated to give theta\_v(r,phi) but we have a choice where (if anywhere) theta\_v(r,phi) matches theta\_v^ref(r). (Essentially, this is the choice of constant of integration.) In principle, we can integrate both north and south from a reference latitude. In practice, it is easier to integrate just in one direction and then correct that solution so that it matches the reference at the desired latitude.

2) Mechanism leading to the formation of II and SI unstable regions.

a) Lines 500 to 510 are tough to understand. This is another paragraph where I was lost. I think it would be useful to show the vertical velocity in Figs. 6 to 8.

b) I would encourage the authors to compute the tilting term and to show it in maps. If I well understood Fig.8a corresponds to -dv/dp and Fig.8b to du/dp and then the tilting term should be du/dp domega/dy - dv/dp domega/dx. So we need to visualize omega to understand how the tilting term is. Since it is the key finding of this section, I found a bit too bad not to spend more time to slowly present the whole detailed arguments to the reader.

We have included vertical velocity in the cross sections in Figs. 7 and 8. We have also added an analysis of the tilting term of the vertical component of the vorticity equation, shown in Fig.9. As a result, Section 3.1.5 has been substantially rewritten (from line 501 in the first version of the manuscript, i.e. from "... 225 km from point C" onwards, to its end). For the sake of brevity, the section is not copied here. Small edits have also been applied to abstract and conclusion to explicitly mention the tilting of vorticity as mechanism of instability generation.

3) The relationship between descent intensity, SJ intensity and mesoscale instabilities. a) I agree that if there is a larger volume (or number of trajectories) satisfying the instability criteria, we should see stronger descents. This relationship is clearly seen in Figure 11d. My concern is more about the Delta|U| during the descent. Why do we expect to get a stronger Delta|U| ? I understand that more unstable regions for SI should lead to more intense transverse circulations but what I do not understand is the acceleration of the along-flow wind speed in presence of such instability. Maybe my question is related to my lack of knowledge on SI in a 3D context, but I think some comments on that would be beneficial for the reader. In other words, the relationship between the slantwise instability across the flow and the along-flow wind speed is not clear to me.

We agree that this link is not fully understood, but we do believe that a relatively simple explanation exists, in that what is required is a mechanism to turn the ageostrophically accelerated radially descending flow to the right. We postulate that there is enough time for Coriolis acceleration to achieve at least some significant turning. We have modified the introduction to this analysis thus: It is not obvious how to isolate the exact role of SI/II in generating the SJ; the basis of our analysis is essentially energetic. At the point where the SJ starts to descend, a substantial proportion of the air eventually leading to the SJ is unstable to SI/II (and so, since it is saturated, to CSI).This has an available potential energy (APE) associated with it, though there is likely to be release going on at the same time as generation so it is difficult to quantify the amount. We have not attempted to quantify the production rate, but have shown that production exists at the point where the SJ starts to descend (and before). As this instability is released, the SJ shows an increase in kinetic energy (KE).

A strong correlation between descent and speed increment can be expected from dynamical arguments as the descent is largely radial with respect to the cyclone centre; the speed increment can then be associated with the work done by the horizontal pressure gradient on the ageostrophic motion. However, the ageostrophic acceleration by the horizontal pressure gradient is the only source of horizontal kinetic energy (friction is a sink) whether the flow is balanced or not so this tells us nothing about mechanism. The increase in final wind speed is of course the same whatever direction the wind enhancement is in and, in common with other studies, we have not decomposed the SJ wind speed into tangential and radial components. Assuming that the increase in KE is primarily due to acceleration in the radial direction, if this does contribute to the primarily tangential speed of the SJ then a mechanism must exist to turn the down-pressure gradient flow to the right. This turning may arise from anisotropy in the horizontal stability or due to the Coriolis acceleration. While the instability is fundamentally unbalanced, the descent occurs over a period longer than fz^{-1}, where  $f_z$  is the usual (z-component of) Coriolis parameter so some geostrophic adjustment is to be expected. (It is also worth remembering that the MetUM, unlike many models, also includes the effect of Coriolis acceleration on the vertical motion.) Of course, exactly the same can be said of any mechanism promoting ageostropic flow towards the low centre.

b) Line 725: this is an example of straightforward remark that I do not understand "This acceleration suggests a link with mesoscale processes on top of the synoptic evolution".

This has been edited to instead read "This acceleration suggests a link with mesoscale processes in addition to the synoptic evolution". A few other edits (highlighted in red) have been made to the text in Section 3.2.3 to improve its clarity.

Minor comments

1) Line 47: at this stage, it is not clear why a focus is made on dry instabilities rather than moist. Maybe you should mention that you want to check the relevance of a mechanism seen in the real storm Tini.

This sentence has removed from this part of the introduction and a related sentence has instead been inserted at the start of the second-to-last paragraph in the introduction where the results from our analysis of windstorm Tini are described.

2) Lines 53-54: I think the end of the sentence has a strange structure and should be reworded

*This sentence has been edited to read* "Baker et al. (2014) produced the first analysis of idealised simulations of SJs in cyclones and found that, while the presence of a SJ was robust to the environmental state in which the cyclone developed, the existence of the different mesoscale instabilities varied: CSI release occurred along the SJ in the control simulation whereas II and CI release occurred in the simulation with the weakest static stability."

3) Line 82: "the strictly" -> "be strictly" *Corrected.*

4) Line 87 "of descent" *Corrected.*

5) Line 101: please explain a bit more the problem encountered in Baker et al. (2014). See my main comment above. *This problem is now described at the start of section 2.1.1.*

6) Line 176: \$phi\_s\$ should be defined. *Done.*

7) Lines 194-195: I do not fully understand the sentence. By constant value, do you mean independent of x and y? *We mean independent of all spatial coordinates (and have now stated so in the text).*

7) Line 195: Why is the result of Eq. (19) not enough to define theta\_v(r,phi) ? See main comment above

As stated above, there remains a choice over precisely where theta\_v(r,phi) matches the input reference profile. (Essentially, this is the choice of constant of integration.)

8) Lines 273-274: what kind of structural changes do we have inside the cyclone by changing the wavenumber

Sting jets have been observed in compact and rapidly-developing extratropical cyclones evolving according to the Shapiro-Keyser conceptual model (Clark and Gray, 2018). Idealised cyclones with this type structure were generated in Baker et al. (2014) using a wavenumber of m=8, therefore we decided to keep the same value for this parameter. In any case, wavenumber is not commonly known as one of the main factors affecting cyclone structure, unless accompanied by change in moisture and/or jet configuration.

9) Line 276: I think the authors should mention they want to check the relevance of a mechanism for the formation of unstable regions for dry instabilities revealed in Volonté et al. Otherwise, it is not clear why a focus is made on dry instabilities *This sentence has been removed from here. This section contains definitions of all the mesoscale instabilities we diagnose, both moist and dry, and hence the sentence could be confusing.*

10) Figure 3: It is not clear why levels 850 hPa and 805 hPa are shown. I would prefer to see one level linked to the SJ (e.g., 805 hPa) and another in the boundary layer linked to the cold conveyor belt jet.

Figure 3a shows wind speed and wet-bulb potential temperature at 850 hPa and clouds at 700 hPa while Figure 3b shows all those fields at 805 hPa, i.e. the pressure level at which the SJ core lies at

that time. The purpose of Figure 3a is to show that the region in which the wind maximum is identified is indeed the frontal-fracture region, located outside the banded tip of a bent-back cloud head at a higher level, consistent with the descent of a SJ. This has been clarified in the text. As we are more interested in the evolution of SJ occurrence than of CCB and other boundary layer features, we think that is best to use Figure 3a to show the levels at which the frontal-fracture region is most visible. The aim is to allow the reader to identify the SJ as a descending airstream before focusing on the evolution of its core (visible in Figure 3b).

11) Line 389: The low-level wind maximum described in the previous paragraphs is the SJ wind maximum, isn't it? *Yes, added* "SJ" to this sentence for clarity.

12) Line 402 and in the rest of the paper. At which level is the SJ wind speed defined? Is it a given isobaric surface? Or is it case-to-case dependent?

The SJ wind is defined at the simulation-specific isobaric surface at which there is a maximum wind speed within the frontal fracture region (see 2nd paragraph in section 3.2.1. where the calculation of the SJ trajectories in the sensitivity experiments is described).

13) Line 410: please insert (Fig. 4a) just before ", indicating" *Inserted.*

14) Lines 457-458: why does this information important?

We agree that this information isn't particularly useful and so the second half of the sentence has been removed.

15) Figure 5: At t= 70-75h, before the ascent, the percentage of CSI is quite high. What does that mean? Is it relevant for the ascent?

Yes, the release of the CSI (which can be seen to occur in Fig 5 following the peak CSI percentages) is likely to have enhanced the ascent of the air prior to its subsequent decent. We have added the following sentence to the paper.

"The release of this instability between about 74 and 82 hours from the start is likely to have enhanced the ascent of the air as the trajectories converge to the north of the cyclone centre to form the coherent sting jet air stream that follows the bent-back front."

16) Line 504: I do not understand "the direct consequence"

This section of the text has been edited to address comment 2a above and we hope that it is now clearer.

17) Caption Figure 7: "Fig. 5a" -> "Fig. 6a" *Corrected.*

18) Caption Figure 8: "Fig. 5a" -> "Fig. 6a" *Corrected.*

19) Lines 505-510: paragraph to be reworded by diluting the information (see main comment 2) above). *Please see the response to comment 2a above.*

20) Lines 527-531: how are all these conditions implemented in terms of maths? *As explained in the text, the selection of the starting points for SJ trajectories for each run (including the control run, as described in Section 3.1.2) has been performed by identifying the 100 points with the maximum wind speed at low levels that are in the frontal fracture at the time of its widening.*

Those features have been identified manually in each of the runs by analysing  $\vartheta_w$  and wind speed maps at a variety of levels and times. We appreciate that this approach is only feasible when dealing with a small sample of storms and that it might appear subjective. However, we would argue that by the very nature of this Lagrangian analysis of the core of an airstream immersed in a continuum of air, it is the one that allows us to best identify the core of the SJ and highlight its evolution without cancelling out results by including air parcels travelling at its edges. The same methodology was followed in Volonté et al. (2018) for the analysis of windstorm Tini.

21) Line 662: please be more explicit to say why t299 case is an outlier.

We agree that the t299 case is not an obvious outlier in the figure panel this text relates to (figure 11b) and so have edited the sentence to read

"We assume that one effect of changing the RH is to change the effective static stability and hence available potential energy of the background state; however, as cloud formation occurs it is difficult to quantify the magnitude of this and there is no obvious dependence of **|U|** on either the RH or sea surface temperature."

22) Lines 671-673: A large part of the descent is azimuthal as well, isn't it? These lines are difficult to follow *Yes. Please see response to 3(a) above.*

23) Line 715: I think the static stability issues are never described in the present paper. One or two sentences would be welcome. *This is now mentioned in section 2.2.1.*

24) Line 725: the suggested link is not clear to me (see my main comment above) *Please see response to main comment 3b above.*

**Reviewer 2 Anonymous**

**General**

The paper presents an important investigation of sting jets (SJs) based on idealized simulations of Shapiro-Keyser cyclones that develop a SJ with the MetUM model. Compared to previous idealized studies, a more realistic representation of a SJ is obtained thanks to improvements of the model initialization. A range of sensitivity experiments is used to explore the environmental control of SJs, such as initial jet velocity, humidity and temperature profiles, and further study the relationship between dry mesoscale instabilities and SJ characteristics. This approach is crucial for understanding what generates the strongest winds, especially given the complexity and high case-to-case variability of previous real case studies. Indeed, SI volume is shown to be consistent with SJ descent through the parameter choices. The paper is overall well-written but the presentation can be made clearer. The introduction in its present form does not set the scene with a clear story line for the current study, and lacks a clear motivation for specific choices made. These issues should be addressed, and I suggest in the following some ways to improve the clarity of the presentation.

Major comments

1. The background description starts with the cold bias of previous idealized studies. The continuation of Sections 2.1.2 and 2.1.3 drowns in details that divert the attention to configurations of the MetUM and the new thermal wind balance for deep flows, without keeping a line of thought of how this helps to solve the bias and different from the previous idealized simulations. The introduction already becomes very long before the mesoscale instabilities are introduced (which are the focus of this study), and is in many places tough to read, with long sentences and many equations that are not essential for understanding the dynamics in the Results. This part of the introduction should be rewritten and I suggest - to move substantial parts to an appendix and leave only concise information in the main text - especially the information that relates to correcting the temperature bias. - add the humidity profile to Fig. 1, Additionally it would be good to visualize the profiles of the present control to previous simulations and those in the sensitivity experiments. The figure(s) can replace some of the text or analytical expressions. - Break down long sentences (e.g., L51-54, L70-73).

**Please see more detailed response to Reviewer 1, but in brief we entirely agree and have moved these sections to an Appendix and simplified them. We have also added the relative humidity profile to Figure 1.**

2. It is not fully clear why you state that you focus on dry instabilities. Please clarify what you mean by dry instabilities on lines 75 and 276, which is currently confusing as you discuss also CI and CSI in the same paragraphs, and essentially through most of the results until the synthesis in Fig. 11. The results in Fig. 11 then raise the question of how CSI release in the ascent phase is related to the SJ characteristics. I therefore suggest to either justify the focus on the dry instabilities, or expand the analysis to encompass moist instabilities among the sensitivity experiments.

The sentence at line 75 now includes the names of the dry (and moist) mesoscale instabilities we are considering as well as clarifying that our focus on the dry instabilities comes after an analysis of the prevalence of all the different mesoscale instabilities in the control idealised simulation. We also emphasize here that the motivation for our focus on dry instabilities comes from our published analysis of windstorm Tini in which we found dry instability release to be important in the sting jet generation. The sentence stating our focus on dry instabilities has been removed from line 276 as this section contains definitions of all the mesoscale instabilities we diagnose, both moist and dry, and hence the sentence could be confusing.

3. L. 278 and Section 2.3: Please explain the rationale behind attributing the instabilities to slantwise trajectories of the SJ. This is non trivial and especially confusing, considering that CI is defined by theoretical purely-vertical motion of an air parcel.

We are not attributing the instabilities to slantwise trajectories of the SJ. Instead we are diagnosing CI (and the other mesoscale instabilities) at points that lie along the SJ trajectories. If CI is released we would expect that release to occur through vertical motions in accordance with theory. For clarity the following sentence has been added in Section 2.2 where the mesoscale instabilities diagnostics are defined: "The release of a diagnosed instability would be expected to be associated with air motion in the appropriate direction (i.e. vertical for CI, horizontal for II, and slantwise for CSI and SI)."

**Minor comments**

- You find a coherent saturated ascent - adiabatic descent pattern of SJ trajectories.

Related to major comment 2, the reader is left wondering about the role of the initial ascent and its variability among the experiments, and how they relate to other SJ characteristics. Please elaborate on this.

The different initial ascent rates (and pressures reached) in the different runs are very likely associated with different generation of moist and dry instabilities (amounts and relative proportions). Different instability generation will affect the SJ descents. In the paper we focus on the evolution of the instabilities during the descent phase of the SJ, but the history of the SJ prior to its descent (particularly during the coherent part of its ascent) is vital for the SJ behaviour. The variability of the mesoscale instabilities throughout the entire SJ evolution (typically ascent followed by descent) across the different experiments is already illustrated by the time series of PV and moist saturated PV shown in figures 9 and 10 and discussed in the corresponding text.

- L4: "and different" is unclear and can be deleted. *Agree, removed.*

- L 82: replace 'the' by 'be'. *Corrected.*

- L103: replace "former study" by the reference, to avoid confusion. The reference referred to is already cited earlier in the same sentence. Hence, we have replaced "the former study" by "that study" to make the link to that reference clearer.

- L141: why do you choose the reduction to m\_v? how is this consistent with RH wrt ice that is used?

Here we are just considering the background state; it has no condensate (liquid or frozen) so the only remaining term is m\_v. This is perfectly consistent with the use of RH wrt ice. We have added a comment to the text to clarify.

- L194, L198-199: these sentences are unclear. A figure (major comment 1) can help to clarify. *See response to major comment 1.*

- L249: add 'd' to 'achieve'. *Corrected.*

- L267: replace 'significant' by 'significance'. *Corrected.*

- L310: reword to ": : : and with accordingly coarser vertical resolution (as shown: : :" Done

- Fig. 2: Mark the frontal fracture region onto the figure. The location of the opening frontal-fracture region is marked with 'FF' in Figure 2b. This has been specified in text and figure caption.

- Fig. 3: Mark 'SJ' on the wind maximum in 3c. Figure 3c has been replotted with a new colour scale that is the same of Figs. 3a and 3b. Hopefully, this makes the identification of the SJ easier in all panels. The description of Fig 3c has also been completed (new text in red). At this point, we believe that a 'SJ' mark is not needed in the figure.

- L396: delete 'and see the list of other studies'.

Done.

- L401: add 'h' after 94. *'hours' added.*

- L427: replace 'magnitude' by 'rate'. Changed to "...magnitude and rate of this descent are consistent with..."

- L475: delete 'the' after 'until'. Done

- L501-502: This is not clear. The slope is directed across the slantwise descending flow. Please clarify.

The slope of the bent-back front tilts outwards from the centre of the cyclone following the contours of equivalent potential temperature in Fig 7b. The slantwise motions ascend and descend the frontal slope as the air travels to the north of the low centre, along the bent-back front. Reviewer 1 also found this section of the text confusing and we have edited to improve clarity.

- Fig. 8 caption: replace '5a' by '6a'. *Corrected.*

- L505-508: These arguments are hard to follow. I suggest to add the vertical velocity onto the CD profiles.

As suggested, the vertical velocity is now included in the cross-sections plotted in Figs 7-8. Please response to main comment 2a from Reviewer 1.

- L569: replace 25 by 45.

25 m/s is indeed the correct value for the increase in wind speed during the descent of the SJ in the control run, which is the subject of the sentence.

- Fig. 11: The legend should have no lines. In caption to (a), note that the 04deg and 025deg overlap at (0,0). *Lines removed from legend. Added note that the symbols for the 04deg and 025deg overlap at (0,0)*

lines removed from legend. Added note that the symbols for the 04deg and 025deg overlap at (0,0) in panel (a) in the caption.

**Short Comment**

David Schultz

**Overview:**

Idealized simulations of cyclones are analyzed here with particular focus on the cause of the lowlevel wind jets. It is concluded that the strong winds resulted from a sting jet. The diagnostics used to analyze this case have been used previously in other published cases of sting-jet cyclones by these authors. There is a rather significant problem with the model simulation initial condition that will require all the simulations to be performed again. The interpretation of the results is a little superficial in places (see detailed comments) and it is unclear why the focus is on dry symmetric instability when moist instability, synoptic-scale forcing, and frontal forcing cannot be ruled out; further diagnosis will be required. The literature cited is incomplete, ignoring previous contributions by other authors, neglecting other cases from the literature that are inconsistent with their results, and not citing contradictory statements from the authors' own research.

Overall, therefore, I find the argument plausible but unconvincing. More precision is needed, or, at least, more caution. Further calculations of the other factors leading to sting jets is needed. While interesting model simulations undoubtedly exist here, I believe that the degree of revision needed constitutes at a minimum 'major corrections', if not 'reject and resubmit'.

We thank Prof. Schultz for his extensive short comment which has enabled us to clarify some aspects of our paper. We shall address individual points below, but would like first to respond to the few more general points raised here.

Regarding the model initial condition, we believe that, in part, Prof. Schultz has simply misinterpreted the contouring in the stratosphere. Regarding the stability to the south of the jet (which was discussed in the paper), as we explained, our objective was to increase the temperature of the overall simulation compared with previous idealised studies which are unrealistically cold (including that by Prof. Schultz), in order to realistically represent the role of moist diabatic processes. Thermal wind balance of a jet inevitably results in lower stability to the S compared with the N. We accept that we pushed this as far as practicable, but the result is a region in the upper troposphere to the S of the jet which is essentially neutrally stratified (yes, there is very slight instability in a small region but this is rather exaggerated by the contouring and has no significant effect on the flow); as stated in the paper, unperturbed the jet persists for 10 days, and this very weak instability erodes without any significant impact.

The reason to focus on dry symmetric instability is precisely because there has been very little discussion of it in the literature regarding SJs, and, in a sense, it is more fundamental than its moist counterpart. CSI may be diagnosed in an airmass but it requires saturation (or, at least, condensation, which may not require saturation in a model with a sub-grid cloud scheme) to be released. This is not ignored in our paper, but we have taken care to only consider it where relative humidity is sufficiently high. Thus, by our diagnosis, CSI can 'disappear' due to adiabatic warming in the descending SJ. In cloud, an airmass unstable to SI is also unstable to CSI, but the SI is not stabilised by decrease in RH. The presence of SI at the start of SJ descent and its decrease during that descent is therefore a much clearer indication that the APE associated with the instability is being converted to KE (and, eventually, heat). We believe that revisions to the text make this clearer now.

Prof. Schultz says the literature cited is incomplete. This is perhaps a question for the editor, but it is the case that, eventually, a topic becomes sufficiently mature that it is impractical to cite all of the literature published so far. This was, in part, our motivation for writing the review that was published a year or so ago (CG18). The majority of the issues arising from past work, and all of the papers cited by Prof. Schultz, are extensively discussed in that review. The only exceptions are a couple of papers published since (including two published after we submitted this one) which we have now cited in appropriate places. We really do want to avoid repeating every discussion within that review.

In particular, a conclusion of that review is that the evidence suggests that a combination of mechanisms are likely to be operating on different space and time-scales and to different degrees in

different systems. In particular, the frontal dynamics has always been regarded as having a role, going back to Browning's first (2004) paper, where he clearly emphasised that the SJ descends out of the cloud head into the frontal fracture region, and that at least some of the descent and acceleration was associated with frontal fracture. Far from ignoring frontal forcing, we have assumed that this is already a well-understood feature of Shapiro-Keyser cyclones.

Prof. Schultz claims that we do not cite 'contradictory statements from the authors' own research.' As will be stated on a case-by-case basis below, we do not recognise these statements as contradictory. Unfortunately, Prof. Schultz seems to be starting with the position that every part of every sentence in a paper must be able to stand alone, without the context in which it is written and, as a result. We believe that this is unrealistic. In each case, sentences have been taken out of context and used to apparently contradict our conclusions.

Finally, Prof. Schultz requires 'more caution' in the conclusions, though later describes some of the conclusions as using 'weak and vague words' when we have attempted to express such caution. Thanks to the comments from the referees, we do recognise and agree that more precision is needed in some places and we believe that the revisions made have addressed this appropriately.

However, Prof. Schultz has suggested that analysis is 'a little superficial in places' and that 'Further calculations of the other factors leading to sting jets is needed' which relate to his later comments on the role of processes other than mesoscale instability release in SJ development. We believe that this highlights a problem common in meteorology but particularly relevant here that warrants a broader discussion in the community. It is often straightforward (in principle) to analyse a mechanism on its own (such as frontogenesis or frontolysis, CSI), though, of course, the papers that first do so are often regarded as classics in the field. If we restrict ourselves to simulations where some phenomena are prevented from manifesting, e.g. by choosing only to run at low resolution, then we can calculate diagnostics to quantify those that do occur, provided they do not overly interfere with one another. Alternatively, where we are confident that a clear separation of scales exists, we may be able to diagnose the role of one mechanism by filtering fields, though, in doing so, we must take account of the larger-scale impact of the filtered phenomena. However, we believe (based on the evidence in CG18) that a continuum of SJ behaviours exists. In an environment where synoptically-forced frontolysis is occurring, undoubtedly modified by being close to neutral w.r.t. CSI and, at least in some cases, actually unstable to CSI and (in our case) SI, what calculations can be done to isolate the relative contributions of each mechanism to the flow? In fact, we agree with the comments of Schultz and Schumacher regarding the co-existence of CSI (or PSI) and frontogenesis: "We argue in this section that the atmospheric response in an environment characterized by MSI is typically closely related to the frontogenetic forcing, making this separation intractable" We know of no examples in the literature where such a separation methodology has been proposed for SJs beyond the trajectorystability method used in this paper and others. However, as discussed below, even the calculation of frontogenesis function becomes an issue due to the extensive folding resulting from the unstable flow (as shown in Volonté et al, 2018), and this method tells us nothing about the role of frontogenesis unless it occurs on its own. We contend that the strength of idealised simulations is that genuine sensitivity analysis can be performed to provide insight into contributing factors and regimes of behaviour, which is the essence of our paper.

**Major comments:**

**L47** Given that the authors had emphasized the importance of moist symmetric instability in sting jets in their previous publications (e.g., the sting-jet precursor depends strongly on the occurrence of CSI) and in L314–315 ("moist processes occurring in the cloud head have a primary role in the evolution of the cyclone in which the SJ occurs and are instrumental in the SJ generation

mechanism"), this emphasis on the dry instabilities only is unclear to this reader. Figure 5 shows that between (1) and (2) that CSI, SI, and II are all equally important. Moist instabilities need to be considered equally, and the focus of the manuscript needs to be changed accordingly, requiring substantial rewriting.

In response to the comments from the two reviewers above we have edited the text to more clearly state that our focus on dry instabilities is motivated by our previous analysis of windstorm Tini in which dry instability release was shown to be important in the sting jet descent. As reviewed in the introduction, several previous papers (including ones we authored) have diagnosed the importance of moist instabilities in the sting jet descent, whereas the role of dry instability release has been little studied and so is a more appropriate topic for a research study. As Prof. Schultz states, we have clearly shown that moist instabilities are present in the control simulation. We have also shown evidence of moist instabilities in the sensitivity experiments (Figs. 9d and 10d). Hence the potential role of moist instabilities in the SJ generation is clear. In response to this comment we have added "as well as moist instabilities" in the conclusions to remind the reader that the noted presence of CSI implies that moist instability release is likely also be involved in the dynamics of the sting jet. We disagree with Prof Schultz's assertion that having shown that moist instabilities as well as dry instabilities exist we should consider them equally in our paper.

**Figure 1** The layer between 400 and 300 hPa appears to be absolutely unstable (i.e., potential temperature decreasing with height). Abrupt and nonuniform gradients of static stability occur within the stratosphere, as well. With the emphasis on the role of instabilities in the cyclone, initializing a model with such a large region of instability should raise a concern. Other initial conditions for idealized cyclones do not show static instability in the upper troposphere (e.g., Fig. 3 of Thorncroft et al. 1993; Fig. 3 of Schultz and Zhang 2007, DOI: 10.1002/qj.87; Fig. 1 of Coronel et al. 2016), so why the authors chose such an unusual set of initial conditions is unclear. Even the set of initial conditions from Polvani and Esler (2007, their Fig. 2) – which the authors claim their "initial base state...is inspired by" and is in return inspired by that of Thorncroft et al. (1993) – has smooth potential temperature gradients throughout the stratosphere and no instabilities in the troposphere. (With such large differences in the initial base states, in what way were your initial conditions "inspired"?) The initialized with any moist instabilities, an analysis of which is lacking in the present manuscript. The model simulations should be redone with any dry or moist instabilities absent in the initial conditions.

We regret that the discussion of the features highlighted above was not clear enough. We find it strange that Prof Schultz should doubt our 'claim' that the initial state was 'inspired by' Polvani and Esler (PE). We are using the word 'inspired' in one of its dictionary definitions: "to give someone an idea for a book, film, product, etc.". Our initial state is a direct modification of PE and we believe it is normal practice to cite the origin of ideas. To avoid this apparently contentious term we have removed it from the text and stated more clearly that the background state follows their methodology closely, but has been modified for very specific and necessary reasons. The issue, as stated in the paper, is that the PE initial state, those in Thorncroft et al. 1993 and of Schultz and Zhang 2007, are quite unrealistically cold in a similar fashion to Baker et al 2014. This does not matter for those papers, as they are studying dry dynamics. That of Coronel et al. (2016) is similarly cold (with sub-zero surface temperatures beneath the jet axis). A strong emphasis of our paper is the role of moist processes in generating instability; to study this it seems essential to have temperatures similar to those of the real-world cyclones in which SJs have been observed. The derivation of our initial state does illustrate the difficulty of generating idealised zonal jets that resemble the real world, and we agree that other solutions to this initial state problem are possible. Thermal wind balance has the effect of reducing stability to the south relative to the north of the jet, and a strong jet can lead to unrealistic static stabilities. However, we wished to retain as much of the PE setup methodology as possible. We believe that we made it clear that our methodology was 'inspired by' PE in the sense that the jet formulation is essentially the same, but modifications were made a) to retain balance in a non HPE model and b) to achieve a warmer, moist, balanced state and c), as per Baker et al 2014, study a smaller and more rapidly developing (n=8 rather than n=6) cyclone. The second goal is achieved by the, in retrospect, simple change of making the virtual potential temperature profile at the jet centre agree with the reference profile, rather than that at the southern end of the domain. While we certainly pushed the PE formulation to its limits, as we clearly state in the paper, we believe that there are no major problems.

The layer that Prof. Schultz says 'appears to be absolutely unstable' between 400 and 300 hPa is, in fact, essentially neutral – we stated 'weakly unstable' in the text, but not to the degree that there is any significant response from either model dynamics or turbulence scheme. As stated, this initial state ran for 10 days with no significant change. The relative humidity is such that moist convection cannot occur until some dynamical forcing is generated. Once disturbances occur in the model, moist convection can and does happen, of course.

Regarding the suggestion that 'Abrupt and nonuniform gradients of static stability occur within the stratosphere, as well', we fear that Prof Schultz is compounding two features. The first is the less stable layer to the N of the jet with somewhat reduced stability around 100 hPa. This is discussed in the text and we see no evidence that it has major bearing on the dynamics of the cyclone. Many other studies do not even extend their domain so high. Our lid is around 40 km. Schultz and Zhang 2007, for example, have a model lid at only 16 km (or around 100 hPa); if a small change in stability here has such an impact, presumably having a model lid at a similar level across the whole domain renders the results of that paper unusable. Whatever the impact, such small changes in static stability are an inevitable consequence of having the top of the jet extend into the lower stratosphere and be in thermal wind balance. However, we suspect that Prof Schultz is actually referring to a second feature: the change in contour spacing from 5 to 20 K at 400 K explained in the figure caption. Of course, there is no 'abrupt gradient of static stability' there. Given the static stability in the stratosphere, there was very little option but to plot this way.

We have attempted to make these points more clearly in the text.

**Figure 4 and its accompanying text** In Clark and Gray (2018, p. 954), the authors write about a modeled sting jet in which "the acceleration amounts to no more than about 2 m/s/hr, but acts over a very slow descent (over more than 12hr); so these trajectories only loosely resemble SJs in observed systems." The air in Figure 4 descends over a 12hr period (86–98 h) and accelerates from 20 m/s to a maximum of 38 m/s (1.5 m/s/hr). Therefore, to be consistent with the authors' previous publications, the trajectories within this simulation should be described within the present manuscript as "loosely resembling a SJ in observed systems".

As is clearly marked in Figure 4 (and stated in the text), the rapid descent period in the control simulation lasts from 88 to 94 hours (6, not 12, hours) during which time the wind speed increases from about 25 to 38 m/s yielding an acceleration of 2.2 m/s/hr. As indicated in figure 11e the total speed increment during the sting jet descent is about 22 m/s. However, clearly different thresholds could be used to arrive at a range of numerical values and we agree that our acceleration rates are similar to those found in the study by Slater, Schultz and Vaughan referred to in the Clark and Gray

review. However, as also explained in that review (if the full text was quoted), the transitional nature (as described by Slater et al) of the just 9 trajectories found in that dry idealised simulation also casts some doubt on their relationship to observed sting jets. We have already included a subsection in the paper (section 3.2.4) in which our simulations are compared to other published idealised simulations of sting jet cyclones. Furthermore, throughout section 3.1, it is made clear that, while showing the properties of a SJ, significantly more extreme (in terms of descent rate, acceleration and final wind speed) SJs have been reported in real cyclones.

**L341 Can the authors clarify within the text what they mean by "irregularities"?**

We recognise that we could have been clearer here. The precise position of the surface pressure minimum at the centre of a Shapiro-Keyser extratropical cyclone has a jumpiness that is due, in addition to the intrinsic noise of this measure, to the elongation of the cyclone-centre region that normally takes place during the development of such a cyclone. The smoothing has therefore been applied to reduce possible errors in the computation of the average speed of the cyclone centre and hence of the system-relative speed of the SJ. The word "irregularities" has been replaced in the text with a mention to the jumpiness just described.

**Section 2.1** Maybe I missed it, but can the authors state how the lower boundary condition was modeled? Is it flat land or ocean? How is the temperature of the surface specified, and is it fixed over time or allowed to vary? How are heat and moisture fluxes handled at the surface?

**Thank you for pointing out that this information was omitted from the paper. The lower boundary condition of the model is ocean and heat and moisture fluxes are set to zero. This has been added to the text.**

**L383** The trajectories for the sting jet are selected from a height of 805 hPa. Given that a sting jet is a surface expression of a region of strong winds (and hence the nearsurface damage potential), it is unclear why the height for selecting sting-jet trajectories is so high. Should these not be selected much closer to the ground, or at least immediately above the boundary layer rather than a height of about 2 km above the surface?

We would like to point out that the pressure level of 805 hPa in the frontal-fracture region at the time of SJ selection is not equivalent to a height of about 2 km above the surface, but rather 1.6 - 1.7 km. This is not surprising as this region is not far from the centre of a rapidly-deepening cyclone, whose surface pressure minimum at that time is at around 970 hPa.

In any case, trajectory analysis uses the resolved model winds and so trajectories are not (particularly) representative of air motions in the possibly turbulent boundary layer where parametrized mixing occurs (see for example discussion on p2274 of Clark et al (2005)). Although some progress has been made since Clark et al. (2005) speculated that kilometre-scale eddies were responsible for the vertical transport of mass and momentum from the SJ through the boundary layer to the surface (see section 6 of the review by Clark and Gray (2018)), more work is needed to determine how and where this transport occurs. Hence, we have followed well-established approach of considering back trajectories from the strong wind SJ region above the boundary layer rather than from the related near-surface strong wind regions.

**L434** The authors report that there is no clear cooling signal due to evaporation or sublimation. However, in Clark and Gray (2018, p. 948) they write that "it is not clear that this [sublimation of ice] differs dynamically from CSI." Given that they've chosen not to focus on CSI in the present paper, yet Figure 5 shows 20–30% of descending air parcels have CSI, how do the authors reconcile these apparent contradictory statements and model results? Is CSI the dynamical equivalent of the sublimation of ice, as they previously wrote? And, is CSI/sublimation present (or important) in the simulations described within this paper or not?

Again, we would encourage other readers of this discussion to read the whole of the relevant section in Clark and Gray (2018). We recognise no contradictions. First, we did not write that CSI is "the dynamical equivalent of the sublimation of ice". The key point here is that, to manifest itself as an actual instability, the condition that the C in CSI refers to has to actually be met. i.e. the air has to be saturated. When considering upright convection, we often ignore this requirement in the descending part of the flow, on the grounds that it is sufficient that the ascending air is unstable. However, when we are considering descent driven by CI (or CSI) no such assumption can be made.

To save time, we refer to Prof's Schultz and Schumacher's excellent 1999 review. We agree that 'PSI' might, in general, be the better term, but for CSI to continue to be a real instability in our descending air, the air must remain saturated. As noted by Schultz and Schumacher, the original study of CSI by Bennetts and Hoskins (1979) is based on "two-dimensional flow in an atmosphere which is assumed to be saturated everywhere". Furthermore, any measure based upon a PV, since it is based upon local gradients, is only valid as a statement about actual instability in saturated air. In descending air, some mechanism is implicitly invoked to maintain this saturation. Either sufficient liquid water is present initially, or the supply is replenished by evaporation of precipitation falling from above. While, in reality, subtle differences due to water loading might distinguish the two cases, we assert that they are dynamically equivalent insofar as both maintain instability due to the continued evaporation while maintaining saturation. We extend this equivalence to ice processes – clearly the thermodynamic variables considered will be different, due to the additional latent heat of freezing, but the dynamical processes are the same.

A parcel of air may start descending from the saturated cloud head unstable to both SI and CSI (and/or its ice-analogue). However, it may have insufficient condensate present or being supplied by precipitation from above to maintain saturation. While still possessing CSI (or, perhaps more correctly, PSI), the parcel is no longer unstable due to it. In our trajectory analysis, once parcel RH has fallen low enough for cloud processes not to happen, we regard it as no longer unstable w.r.t. to CSI. Thus, very little evaporation may have occurred during descent despite the presence of CSI. (Again, this is extensively discussed in Clark and Gray, 2018.)

**L476–479 and throughout the manuscript** The authors state the number of trajectories unstable to dry mesoscale instabilities is "substantially smaller" than that of a previous case and therefore concludes that "the release of mesoscale instabilities such as SI and II takes part in the dynamics of SJ speed increment and descent". These two statements would seem to be contradictory.

All statements within our paper should, as with all papers, be read in the context of the text in which they are found. We do not, anywhere, make the direct link implied by Prof Schultz, but we also make it clear that the two statements are not contradictory. The proportion of trajectories unstable to dry mesoscale instabilities is exceeding 30% just before descent occurs in the idealised control simulation; in contrast this proportion exceeds 70% in windstorm Tini, which is the strongest, in terms of final windspeed, sting jet so far studied in the literature. Hence, the proportion is smaller in the idealised simulation than in the case study, but we argue that it is still "substantial". To some extent, the proportion of trajectories with instability depends upon how narrowly we define the final SJ core; it is clear in other figures that the trajectories consistently emerge from a coherent region of instability, and we could have, post facto, narrowed the selection criteria to that a much larger percentage of

**the trajectories had dry mesoscale instabilities. A substantial volume of air would have remained, but we felt that step to be unjustified. Hence the above statements are not inconsistent.**

Moreover, these statements also contradict, for example:

- L8–9: "A substantial amount of SI...is released along the SJ during its descent...."
- L519–520: "mesoscale instabilities...play an active role in the evolution of the SJ...."

• L667–668: "it is difficult to assert a strong relationship between...SJ maximum speed and degree of SI."

• L683–684: "there seems to be overall some evidence that weakly enhanced SJ strength is associated with increased SI".

Given the degree of inconsistency among these various statements within the manuscript, is the word "robust" in the title of this manuscript appropriate?

Thus, more clarity and consistency on the degree and importance of the dry (and moist) instabilities in relation to the acceleration of the SJ is needed throughout the manuscript.

L8-9: In response to this comment we have rewritten the sentence in the abstract to instead focus on the more important point that the proportion of trajectories with dry instabilities is similar to that with moist instabilities at the start of the sting jet descent and avoid the use of the subjective word "substantial".

The paper title states that symmetrically-unstable sting jets robustly occur in the simulations. As shown in the paper and summarised in the conclusions, 12 of the 13 simulations contain airstreams analysed as sting jets and more than 25% of trajectories have SI in 7 of these 12 sting jets. Hence symmetrically-unstable sting jets occur in more than half of the simulations and are thus robust despite the wide range of changes in the initial conditions applied. The simulations that do not yield symmetrically-unstable sting jets are mainly those where less instability would be expected: those with reduced resolution, jet strength and relative humidity.

Section 3.2 Here it is concluded, "There seems to be overall some evidence that weakly enhanced SJ strength is associated with increased SI, but clearly other processes are occurring in the different cases to complicate behaviour." It's not clear to me how I've learned anything useful from this analysis. First, weak and vague words ("seems to be", "some evidence", "associated", "other processes", "complicate behaviour") obscure the meaning of this sentence and the actual results. Greater precision is needed when writing such important conclusions. Second, given that the authors admit that synoptic-scale and frontal-scale circulations are in part responsible for this descent (L618–619) and that the initial strength of the jet stream, and hence of the cyclone, has changed in these simulations, then the authors cannot rule out that the magnitude of the forcing has increased and is the major contributor to the differences in the accelerations of the jets across these sensitivity experiments. The analysis within the present manuscript does not tell us what is causing the acceleration to be strong in these regions. Schultz and Sienkiewicz (2013) discuss this issue in their paper (see p. 604). At a minimum, the authors would appear to need to calculate the synoptic and frontal forcing to determine if changes in these can explain their model results. Simply concluding that "several environmental factors modulate this relationship, making it difficult to disentangle the net effect of instability release" (L765–766) undermines the basis of their study. Such a sentence would appear to be a weak concluding statement when such factors could be calculated and examined. After all, the purpose of a scientific paper should be to shed light on these

factors for the benefit of the readers, rather than be defeated and conclude such a problem is intractable.

**Thank you for pointing out that we should have expressed this final sentence of this section more strongly. We have now rewritten the sentence to read**

"We conclude that, although not a strong relationship, there is evidence that weakly enhanced SJ strength is *associated* with increased SI. This relationship is evident despite the complicating effects of the likely changes in other processes such as large-scale and frontal dynamics and mixing in the different experiments."

Prof Schultz criticizes our use of the word "associated". However, we have retained this word as we understand that it is appropriate to use it when causality hasn't been explicitly demonstrated, as is the case here. As already discussed in the introduction to the paper, particularly in reference to our review paper (Clark and Gray, 2018), we recognise that it is likely that a continuum of behaviour occurs in SJs from large-scale dynamics, frontal dynamics (including the frontolysis resulting in frontal fracture in Shapiro-Keyser cyclone), mesoscale instability release, and evaporative cooling. This continuum of behaviour will necessarily complicate the analysis of the relationship between SJ characteristics and mesoscale instability in the experiments. The interesting result from section 3.2.3 is that **despite** these other processes occurring, a relationship, albeit not a strong one, is found.

Prof. Schultz suggests that we also calculate the synoptic and frontal forcing to see whether these factors are also related to the SJ characteristics. We did not show these because of the experience gained with the Tini study. It would not be sufficient to simply calculate the frontogenesis function in the frontal-fracture/cloud head tip region of the cyclones as in Schultz and Sienkiewicz (2013) because, as Volonté et al. (2018) demonstrate for cyclone Tini, the release of mesoscale instability in simulations with sufficient resolution for this release to occur, leads to fine-scale banding in the frontogenesis function and so a more complicated pattern than the broader-scale frontogenesis region in the bent-back front and frontolysis region in the frontal-fracture region found in the cyclone studied by Schultz and Sienkiewicz (2013). Tracking the frontolysis function along trajectories showed us that, because of the complex nature of the folds generated by the unstable flow, no clear relationship between the SJ and frontolysis is apparent – in fact, much of the SJ is associated with very local frontogenesis. To our knowledge, only in calculations performed on fields where an unstable SJ did NOT form, such as the lower resolution simulations of Tini in Slater et al. (2017) and in Volonté et al. (2018), is a clear frontolysis pattern apparent, and tracking the evolution of the frontogenesis function along trajectories in simulations that resolve mesoscale instability release may give the, probably misleading, impression that frontal and synoptic dynamics have no role in the evolution of the SJ. We have added some comments to this effect (and referred to Schultz and Sienkiewicz (2013)) at the end of section 3.2.3.

It is not obvious to us what alternative calculations would be useful, as much depends on precisely what question is asked. It might be possible to generate a spatially smooth field of frontogenesis function by filtering or by calculating it from low resolution versions of the simulations which are unable to release instabilities, but it is not obvious how valid these simulations would be.

**L16–17, but throughout the manuscript** Are you comparing with other analyzed cases of sting jet cyclones here when you say that the sting jet in your case is robust? If so, then the literature is not so clearly uniform on this issue of instabilities in cyclones. Some model simulations of real cyclones with sting jets (e.g., Smart and Browning 2014; Brâncuş et al. 2019) found little to no instability associated with the sting jet. What does the absence of instabilities in observed cases mean for the

authors' conclusions? The authors should express that ambiguity more clearly throughout the manuscript because idealized simulations, as informative as they are and therefore commonly used, do not often represent reality. More care needs to be taken to avoid overgeneralizing the results of this study.

The title of our paper is "Idealised simulations of cyclones with robust symmetrically-instable sting jets". Hence it is clear that the "robustness" is between the different idealised simulations. Of course, an idealised simulation does not cover all possible real-world scenarios. The other issues raised here have already been discussed in Clark and Gray and do not seem relevant to this paper.

**Minor comments:**

**L25–26** Schultz and Browning (2017) (DOI: 10.1002/wea.2795) argue that one cannot identify a sting jet from the surface observations alone and should be cited here.

Thank you for the suggestion. We instead chose to cite an earlier paper, Martínez-Alvarado, O., Baker, L. H., Gray, S. L., Methven, J., and Plant, R. S.: Distinguishing the cold conveyor belt and sting jet air streams in an intense extratropical cyclone, Mon. Weather Rev., 142, 2571–2595, doi:10.1175/MWR-D-13-00348.1, 2014, which is precisely related to our actual statement which is that it is difficult to distinguish the SJ from the cold conveyor belt using surface observations alone.

**Section 1** It would seem appropriate to cite the comprehensive review of conditional symmetric instability (as well as other instabilities) by Schultz and Schumacher (1999) (DOI:10.1175/1520-0493(1999)127<2709:TUAMOC>2.0.CO;2) somewhere in the introduction.

We already cite the extensive description of the different mesoscale instabilities provided in a section of the review paper on sting jets by Clark and Gray (2018). The excellent review led by Prof Schultz is cited in that paper.

**L46** I suggest that the paper by Schultz and Sienkiewicz (2013) (DOI: 10.1175/WAF-D- 12-00126.1) is cited here as this is the first paper I know of that has discussed the importance of frontolysis.

In this sentence we are referring to frontal dynamics in general rather than frontolysis specifically. Hence, we have chosen to cite the Clark and Gray (2018) review paper in which all of the studies attributing sting jet flows to frontal dynamics are cited (including that led by Prof. Schultz) rather than list all of those studies here. However, note the discussion of 3.2.3 above.

**L81** Brâncuş et al. (2019) (DOI: 10.1175/MWR-D-19-0009.1 and Eisenstein et al. (2019) (DOI: 10.1002/qj.3666) also considered the importance of these instabilities and should be cited here.

**Thank you for reminding us to include reference to these two studies which have both been accepted for publication since we submitted our paper. Discussion of them is now included here.**

**L427** There are many other papers on modeled sting jets that could be included here - all consistently showing that the descent here is consistent with that in previous studies. Thus, it is not correct to say that the results found in this case are the same as in other cases and cite only the Volonté et al. paper. Whether or not you do this, I suggest you reference the paper by Slater et al. (2017) (DOI: 10.1002/qj.2924).

The Slater et al. paper considers the same case study as the Volonté et al. (2018) paper and has higher descent rates and accelerations to that being cited here. The authors would argue that the model used to analyse the Slater et al. cyclone has insufficient resolution to allow a sting jet to form

if associated with any mesoscale instability, but it does have the resolution to produce descent and acceleration due to frontolysis.

The results from Volonté et al. (2018) are the largest reported to our knowledge for a SJ in terms of peak wind speed and descent. Therefore, they are compared with the ones in this study, as it has now been clarified in the text. Regarding Slater et al. (2017), while it is true that some trajectories show a similar descent rate of around 150 hPa / 3 hrs, those trajectories are in fact only three out of the already very small sample of 5 selected. In Volonté et al. (2018) instead, the whole core of the SJ, indicated by 100 trajectories, descends and accelerates coherently. Furthermore, while the SJ in Volonté et al. (2017), even for those three carefully-selected trajectories.

Prof Schultz is right in thinking that we would argue that the simulation in Slater et al. (2017) does not have the necessary resolution to allow the formation of a SJ whose evolution is driven by the release of mesoscale instabilities (again, see Volonté et al., 2018 for a more comprehensive discussion). It is unfortunate that the authors did not test this with their model. For the same case, but a different model, Volonté et al. (2018) did, and showed a much stronger SJ in the higherresolution case (with mesoscale instabilities) than lower. As such, and given that its result cannot be considered of similar or larger magnitude than in Volonté et al. (2018), we do not see its relevance in the context of the sentence mentioned in the comment. If Prof. Schultz is suggesting that, in the most extreme case, frontolysis alone can produce as strong a SJ as produced in this idealised simulation, we believe this has already been said in Volonté et al. (2018), but we have already stated that our idealised simulations do not match the most extreme case - here the relevant comparison is with the lower resolution idealised simulation.

**L434** There are many other papers on observed and modeled sting jets that could be included here - all consistently showing that cooling is minimal (e.g., Smart and Browning 2014; Coronel et al. 2016; Slater et al. 2017; Brâncuş et al. 2019). On the other hand, Eisenstein et al. (2019) found cooling was much more important in their case. This diversity of results should be discussed.

Again, this is an issue that is discussed in the Clark and Gray (2018). We agree this warrants comment and that the two papers above published since then deserve mention. We have added the following:

The role of evaporative cooling during descent is discussed in Sec. 5.6 of CG18. In common with our results, a number of papers cited therein suggest that evaporative cooling is negligible, as does recent work by Brancus et al. (2019). CG18 also highlight the possibility that evaporative cooling may have had an impact of some of the more extreme observed SJs, and this is further supported by the more recent work of Eisenstein et al. (2019). Our results support their statement that "while evaporative cooling may be occurring, it seems unlikely that this additional latent cooling is *essential* for the formation of a SJ".

**L614–615** Schultz and Browning (2017) (DOI: 10.1002/wea.2795) argued that the wind maximum of a SJ needed to exit the cloud head and accelerate, and should be cited here.

**Thank you for the suggestion. We have instead chosen to cite to the first modelling study that identified these characteristics of sting jets (Clark et al. 2005).**

**L704–705** In addition to the citation to Coronel et al. for recognition of the importance of the synoptic-scale forcing, I suggest adding a sentence citing the paper by Schultz and Sienkiewicz (2013)

(DOI: 10.1175/WAF-D-12-00126.1) as the first paper I know of that has discussed the importance of frontolysis as a forcing mechanism for sting jets.

**As is stated in the first sentence of this section, we are here comparing against previous **idealised** case studies and so unfortunately it would not be appropriate to mention the case study paper led by Prof. Schultz here.**

**L720** There are many other papers on modeled sting jets that could be included here - all consistently showing that the descent here is consistent with that in previous studies. Whether or not you do this, I suggest you reference the paper by Slater et al. (2017) (DOI: 10.1002/qj.2924). The Slater et al. paper considers the same case study as the Volonté et al. (2018) paper and has higher descent rates and accelerations to that being cited here.

**Please see the response to previous comment.**

**L731–732 and throughout the manuscript** Not all sting jet cases are associated with dry instabilities. You should cite relevant literature by other authors that show other cases with negligible amounts of these instabilities (e.g., Smart and Browning 2014; Brâncuş et al. 2019). Consider other statements within the manuscript that should be similarly reworded with additional caveats and citations.

The first sentence in this comment seems to be suggesting that other cases have been reported in the literature that do not demonstrate dry instabilities. If this is the case, then we would agree, and have said so in the introduction.

Although the potential role of dry instabilities has simply not been considered in many studies there are some papers that have explicitly considered it in addition to our previously published one on the case study of windstorm Tini. However, we disagree that the two studies mentioned here show negligible amounts of these instabilities. The Smart and Browning (2014) paper does not include any discussion of the presence of dry instabilities in the text, but does show in their Fig 9 the evolution of PV along three sting jet trajectories. The PV of all 3 trajectories decreases as the three sting jet trajectories ascend and one of the trajectories approaches very close to zero PVU at 01 UTC, as the trajectories reach their minimum pressure level and start to descend. Although none of this extremely small sample of three trajectories (compared with the 100 used in this study) acquire negative PV, the reducing symmetric stability in all three trajectories during the sting jet ascent followed by increasing symmetric stability as the sting jet descends closely follows the results in our paper (compare to our Fig 9c) and the approach towards near neutrality to SI along one of the trajectories suggests that the possible presence of SI cannot be excluded in the Smart and Browning study. As a co-author of the Brâncuş et al. (2019) study, Prof. Schultz will be aware that although its abstract includes the sentence "Mesoscale instabilities appeared to be unimportant in the sting-jet formation.", section 8 of that paper includes the statements "inertial instability was present in the last part of the sting jet's descent when the sting jet entered into the boundary layer" and "inertial instability is present in narrow bands along the bent-back front" with the latter described as "consistent with the findings of Volonté et al. (2018)". The findings in these two papers related to dry instabilities are now both summarised in the introduction to our paper.

---

## Author Response (AR2)

**Response to editor comments on "Idealised simulations of cyclones with robust symmetrically-unstable sting jets" by Volonté et al.**

wcd-2019-8 - A. Volonté on behalf of all authors, 28 January 2020

Dear Editor,
Many thanks for taking time to assess our manuscript and for your comments which have improved its style and the presentation of its content. Below we respond to each of these comments.
Our responses are given in *green italic* font, while new text in the manuscript is indicated in red, here and in the manuscript.

0) Maybe a matter of taste, but I find that overall you use too few commas for a good reading flow. Instead sometimes you use semi-colons to concatenate sentences that could easily "survive" on their own.

*We have reread the manuscript looking for places to add additional commas and remove semicolons. We have added 4 commas to the introduction and removed 9 semi-colons from the text (either changing them to full stops or colons). Note that many of the semi-colons we use act as separators for lists and so it is not appropriate to remove them.*

1) L94–96: Sentence is hard to read, please reword.

*The sentence has been rephrased and hopefully clarified. It now reads:*
PV decreases along these three trajectories as they ascend and gets very close to zero on one of them at 01 UTC 3 January, the time at which the trajectories reach their minimum pressure level and start to descend.

2) L97: remove "possible"

*Removed.*

3) L114–137, new section on initial conditions: While the content is fine, I find the quality of writing inferior to the rest of the paper. Some expressions are somewhat colloquial and some sentences are a bit too long and unnecessarily complicated. I invite the authors to go over this part once again and sharpen the message.

*We have reworded these lines as follows:*
*The initial base state used is generated using an algorithm based on, and ultimately very close to, that of Polvani and Esler (2007), referred to as PE hereafter. However, two issues arise from their formulation. The first is that their derivation is based upon geostrophic thermal wind balance in the hydrostatic primitive equation (HPE) set. This approximate set ignores various terms, including some Coriolis terms and some metric terms (e.g. White et al (2005)). As a result, a 'balanced' state in the HPEs is not perfectly balanced in a less approximate equation set (and vice versa). The MetUM solves the non-hydrostatic deep atmosphere dynamical equations on a sphere, so the balanced initial state found by PE is only approximately balanced when used in the MetUM. This initial imbalance is quite physically unrealistic and generates an initial transient flow (largely gravity waves) that, at the*

*resolutions used, can result in an unstable response. Baker et al (2014) followed the pragmatic approach of first running for a "number of timesteps" in a lower-resolution, larger-area version of the model with fixed lateral boundary conditions on all four boundaries. This was found to sufficiently damp the transient response and settle to a balanced field that could be numerically interpolated to the required grid. However, we have derived an equivalent to geostrophic thermal wind balance in the non-hydrostatic deep atmosphere and hence removed this initial imbalance. To our knowledge this result has not been previously published, so it is included in Appendix A1. The initial state produced is equivalent to that of PE but for a non-hydrostatic, deep atmosphere model.*

*A more significant issue with the initial conditions used by Baker et al (2014) was their rather cold overall temperature, discussed above. This temperature structure arises from the PE initial state; it is of only minor importance in PE's dry simulations, but restricts the magnitude of diabatic forcing by phase changes. In this study a simple method to adjust the entire vertical virtual potential temperature (theta_v) profile to make that at the jet centre equal to the reference has been devised, the details of which are also given in Appendix A1.*

*Any upper-level jet in thermal wind balance must have lower static stability in the upper troposphere to the south of the jet compared with that to the north. The state used in this paper was chosen to have a realistic static stability in the middle of the jet. The profile is close to neutrality in the upper troposphere south of around 40° N. In fact, it is very weakly statically unstable, but not sufficiently to generate either a dynamical response or significant unstable mixing through the turbulence scheme, and, if unperturbed, it was found that the whole state remained essentially unchanged through a 10-day integration apart from some very minor wave activity in the upper stratosphere. We thus judged that the tiny adjustment required to remove this very small static instability was not necessary. Furthermore, a larger adjustment to increase the static stability in this region would have led to a reduced baroclinic growth rate, and the aim was to produce condition believed more conducive to SJ development.*

4) L361: use abbreviation "SJ"

*"sting jet" replaced with "SJ" here and for two other occurrences in the text, thanks.*

5) Caption Fig. 6: transects -> lines

*Done.*

6) L400: missing brackets?

*Brackets added (and moved to the end of the sentence), thanks for spotting this.*

7) L410-412: split into two sentences

*Sentence split in two, see below:*
This overall situation is consistent with the mechanism, outlined in Volonté et al. (2018), of slantwise circulations generating negative horizontal relative vorticity that can be eventually tilted into the vertical to trigger II and SI too. In fact, SI is generated if, as a consequence and as is the case here, not only ζz but also PV decreases below zero.

8) L594–595: sentence does not read well, either add commas or break in two

*The sentence now reads:*

This has an available potential energy (APE) associated with it. However, there is likely to be release going on at the same time as generation, so it is difficult to quantify the amount.

9) L600: you have now introduced the abbreviation "KE" and should use that consistently.

*We agree, corrected.*

10) L604: comma after "SJ"

*We disagree on adding that comma because that would imply that "if this does contribute to the primarily tangential speed of the SJ" could be removed from the sentence without affecting its meaning, which is not true.*

11) L607: comma before "so". I would refrain from including entire sentences in brackets.

*The comma has been added and the brackets removed from the following sentence.*

12) L609: I am not sure I really understand that last sentence. What do you lose if you omit it?

*Agreed. Sentence removed.*

13) Acknowledgements: I think the reviewers deserve a mentioning here.

*We agree, acknowledgements added.*